# TEAD2 initiates ground-state pluripotency by mediating chromatin looping

Rong Guo[1,2,7], Xiaotao Dong[2,3,7], Feng Chen[1], Tianrong Ji[2], Qiannan He[1,2], Jie Zhang [1,2], Yingliang Sheng [1,2,4], Yanjiang Liu[1,2], Shengxiong Yang[1,2], Weifang Liang[5], Yawei Song[2], Ke Fang [2], Lingling Zhang[6], Gongcheng Hu [1,2] & Hongjie Yao [1,2]✉

## Abstract

The transition of mouse embryonic stem cells (ESCs) between serum/LIF and 2i(MEK and GSK3 kinase inhibitor)/LIF culture conditions serves as a valuable model for exploring the mechanisms underlying ground and confused pluripotent states. Regulatory networks comprising core and ancillary pluripotency factors drive the gene expression programs defining stable naïve pluripotency. In our study, we systematically screened factors essential for ESC pluripotency, identifying TEAD2 as an ancillary factor maintaining ground-state pluripotency in 2i/LIF ESCs and facilitating the transition from serum/LIF to 2i/LIF ESCs. TEAD2 exhibits increased binding to chromatin in 2i/LIF ESCs, targeting active chromatin regions to regulate the expression of 2i-specific genes. In addition, TEAD2 facilitates the expression of 2i-specific genes by mediating enhancer-promoter interactions during the serum/LIF to 2i/LIF transition. Notably, deletion of *Tead2* results in reduction of a specific set of enhancer-promoter interactions without significantly affecting binding of chromatin architecture proteins, CCCTC-binding factor (CTCF), and Yin Yang 1 (YY1). In summary, our findings highlight a novel prominent role of TEAD2 in orchestrating higher-order chromatin structures of 2i-specific genes to sustain ground-state pluripotency.

**Keywords** Ancillary Factor; Chromatin Looping; Embryonic Stem Cells; Ground-state Pluripotency; TEAD2
**Subject Categories** Chromatin, Transcription & Genomics; Stem Cells & Regenerative Medicine

## Introduction

Embryonic stem cells (ESCs) are karyotypically normal cells derived from the inner cell mass (ICM) or epiblast of peri-implantation embryos. Mouse ESCs cultured in serum and leukemia inhibitory factor (serum/LIF; SL) display a metastable state, expressing various lineage-specific genes and being prone to differentiation (Chambers et al, 2007; Evans and Kaufman, 1981; Hayashi et al, 2008). In contrast, ESCs cultured in serum-free medium with LIF and two inhibitors (PD0325901 and CHIR99021) (2i/LIF; 2iL) exhibit a more homogeneous phenotype, resembling the inner cell mass of the preimplantation epiblast and reflecting a "ground-state" pluripotency (Boroviak et al, 2014; Marks et al, 2012; Marks and Stunnenberg, 2014; Ying et al, 2008). Despite both 2iL- and SL-ESCs representing "naïve" pluripotency (Hackett and Surani, 2014; Nichols and Smith, 2009) serving similar functions, they significantly differ in cell-cycle, metabolic, transcriptional, translational, and epigenetic profiles (Atlasi et al, 2020; Atlasi et al, 2019; Habibi et al, 2013; Joshi et al, 2015; Marks et al, 2012; Peng et al, 2020; Ter Huurne et al, 2017; van Mierlo et al, 2019). Importantly, the transition between 2iL- and SL-ESCs, achieved by altering the culture medium, provides a valuable system for investigating factors involved in ground-state pluripotency and studying gene regulation mechanisms (Atlasi and Stunnenberg, 2017; Habibi and Stunnenberg, 2017; Peng et al, 2020).

The regulation of pluripotency, self-renewal, and differentiation in stem cells involves core transcription factors (TFs) and ancillary factors (Hackett and Surani, 2014). Ancillary factors, including KLF2, ESRRB, PRDM14, SALL4, and TCFCP2L1, play crucial roles in stabilizing the pluripotency regulatory network and preventing loss of self-renewal (Hackett and Surani, 2014). Many ancillary factors are expressed differentially in 2iL-ESCs and SL-ESCs (Marks et al, 2012). Among these factors, PRDM14 and ESRRB, contribute to ground-state pluripotency in 2iL-ESCs by promoting active DNA demethylation and activating specific enhancers, respectively (Atlasi et al, 2019; Okashita et al, 2015; Yamaji et al, 2013).

Transcriptional enhanced associate domain (TEAD) transcription factors play pivotal roles in development, cell proliferation, regeneration, and tissue homeostasis (Huh et al, 2019). TEAD proteins encompass a DNA-binding domain known as the TEA domain and a protein binding domain that facilitates interaction with transcriptional co-activators (Landin-Malt et al, 2016). TEAD2 specifically interacts with Yes-associated protein (YAP)

[1]State Key Laboratory of Respiratory Disease, The First Affiliated Hospital of Guangzhou Medical University, Guangzhou National Laboratory, Guangzhou Medical University, Guangzhou, China. [2]Center for Health Research, Guangzhou Institutes of Biomedicine and Health, Chinese Academy of Sciences, Guangzhou, China. [3]School of Basic Medical Science, Henan University, Kaifeng, China. [4]Division of Life Sciences and Medicine, University of Science and Technology of China, Hefei, China. [5]College of Veterinary Medicine, Shanxi Agricultural University, Jinzhong, China. [6]Institute of Clinical Pharmacology, Anhui Medical University, Hefei, China. [7]These authors contributed equally: Rong Guo, Xiaotao Dong. ✉E-mail: yao_hongjie@gzlab.ac.cn

and PDZ-binding motif (TAZ), orchestrating the delicate balance between stem cell self-renewal and differentiation across various developmental stages by modulating Hippo pathway activity (Currey et al, 2021; Tian et al, 2010). While the functions of YAP and TAZ in ESCs have been extensively scrutinized (Passaro et al, 2021; Sun et al, 2020), the role of TEAD proteins remains less clear. Mammals express four TEAD proteins (TEAD1-4), each exhibiting distinct expression patterns during development (Yasunami et al, 1996). TEAD2, expressed early in mammalian development, emerges as an essential player in neural development (Kaneko et al, 2007; Landin-Malt et al, 2016; Sawada et al, 2005). However, the influence of TEAD2 on the regulation of pluripotency in mouse ESCs remains elusive.

In the current study, we elucidated the factors crucial for ground-state pluripotency and observed that knockdown of *Tead2* did not impact the expression of core pluripotency factors *Oct4* and *Sox2* in both 2iL- and SL-ESCs. However, it induced a shift in the morphology and gene expression of cells cultured in 2iL-ESCs, making them more akin to SL-ESCs. *Tead2* knockout resulted in altered cell morphology, disrupted self-renewal, and diminished expression of 2i-specific genes during the SL-to-2iL transition. Although *Tead2* overexpression failed to enhance SL-to-2iL transition, it conferred upon SL-ESCs the expression of partial 2i-specific genes. Notably, TEAD2 exhibited specific binding to more active chromatin regions in 2iL-ESCs compared to SL-ESCs. *Tead2* knockout led to the loss of TEAD2-mediated EP interactions on 2i-specific genes, culminating in the downregulation of these genes.

# Results

## Dynamic changes in chromatin accessibility during SL-to-2iL and 2iL-to-SL conversion

To decipher the factors governing the transition of mESCs between SL-to-2iL or 2iL-to-SL conditions, we established protocols for the interconversion of SL-ESCs and 2iL-ESCs, faithfully reproducing key features such as cell morphology and DNA methylation reported previously (Habibi et al, 2013; Marks and Stunnenberg, 2014). Our data revealed increased morphological heterogeneity in cell colonies during 2iL-to-SL transition, contrasting with the more homogeneous and domed morphology observed during SL-to-2iL transition (Fig. EV1A). Global DNA methylation, assessed through *HpaII* (unmethylated DNA) and *McrBC* (methylated DNA) restriction enzymes, exhibited a gradual decrease and increase during transitions from SL-to-2iL and 2iL-to-SL, respectively (Fig. EV1B,C). Subsequently, we conducted RNA sequencing (RNA-seq) and assay for transposase-accessible chromatin with high-throughput sequencing (ATAC-seq) at various time points (day 0, 3, 6, 9, 12, 15 of SL-to-2iL and 2iL-to-SL) throughout the interconversion of 2iL-ESCs and SL-ESCs. Principal component analysis (PCA) illustrated gradual changes in both gene expression and chromatin accessibility landscape trajectories during transitions between 2iL-ESCs and SL-ESCs (Fig. EV1D,E), signifying these systems as ideal for studying cell fate conversion.

To explore the dynamics of chromatin accessibility during the interconversion of 2iL-ESCs and SL-ESCs, we categorized the peaks at each time point into three groups: permanently open (PO) in both 2iL-ESCs and SL-ESCs, open-to-closed (OC), and closed-to-

open (CO). The CO and OC peaks were further classified into five subgroups (CO1-5; OC1-5) based on the timing of opening and closing, capturing the alterations in accessible chromatin between 2iL-ESCs and SL-ESCs (Fig. 1A,B). In addition, we defined the serum-specific subgroups OC1-5 of SL-to-2iL as Region 1 and CO1-5 of 2iL-to-SL as Region 4. The 2i-specific subgroups CO1-5 of SL-to-2iL were designated as Region 2, and OC1-5 of 2iL-to-SL as Region 3 (Fig. 1A,B). Notably, peaks in Region 2/3 were predominantly situated in the initial stages of the transition (Fig. 1A,B). Counting these peaks revealed that PO peaks (38,013) were the most abundant, and Region 2/3 (20,508) surpassed Region 1/4 (13,341) during the transition (Fig. 1C). These findings suggest that changes in chromatin accessibility at 2i-specific peaks may play a crucial role in the transition between 2iL-ESCs and SL-ESCs.

Next, we assessed the overlapping peaks of POs, OCs, and COs between these transition processes. We observed that 28,652 of the PO peaks between these two processes were identical. Region 1 shared 17.8% (1566/8819) with Region 4, and Region 2 shared about 34.3% (6430/18,721) with Region 3 (Fig. 1D). Genes associated with PO peaks, such as *Ctcf* and *Sox2*, maintained open chromatin accessibility, and their expression remained relatively stable during the transitions (Fig. EV2A,B). Furthermore, genes in overlapping peaks in Region 1/4 and Region 2/3 were predominantly present in the initial stages of transition (day 0–6), constituting 57.5% (481/837) and 32.5% (766/2357), respectively (Fig. 1E). Gene Ontology (GO) analysis revealed that the 481 genes in Region 1/4 were implicated in lineage-specific terms, including neuronal system, epithelial cell differentiation, and muscle structure development. The 766 genes in Region 2/3 were associated with metabolic terms, including glycerophospholipid metabolic process and phospholipid metabolic process, consistent with previous descriptions of SL-ESCs or 2iL-ESCs (Marks et al, 2012; Marks and Stunnenberg, 2014) (Fig. 1F; Dataset EV1). Both chromatin accessibility and expression levels of genes in Region 1/4 (such as *Mmp2* and *Krt19*) and genes in Region 2/3 (such as *B4galt6* and *Lpin1*) changed at day 3 during SL-to-2iL or 2iL-to-SL (Figs. 1G,H and EV2C,D). These results suggest that the overlapping accessible peaks between these two processes are functionally related to 2iL- and SL-ESCs and begin to change at the initiation stage of the transition, representing a key region for the transition.

## TEAD2, TEAD4, TCFCP2L1, ESRRB, and NR5A2 have been identified as potential regulators of ground-state pluripotency

To further investigate the TF networks involved in the interconversion of 2iL-ESCs and SL-ESCs, we conducted motif analysis on the PO, CO, OC peaks. Approximately 60 TFs were identified, with their enrichment primarily observed in PO or the initial stages (day 0–6) of Region 2/3, showing less enrichment in Region 1/4 (Fig. 2A,B). This implies that 2i-specific peaks might play a crucial role in the interconversion of 2iL-ESCs and SL-ESCs. Notably, PO loci were predominantly enriched with motifs for CTCF, ZF, and POU families, while Region 2/3 loci specifically displayed enrichment with TEA, CP2, and NR motifs (Fig. 2A,B). Subsequently, we explored the dynamic expression of TEA, CP2, and NR family transcription factors (*Tead2, Tead4, Tcfcp2l1, Esrrb*, and *Nr5a2*) during the transition between 2iL- and SL-ESCs. Integrating

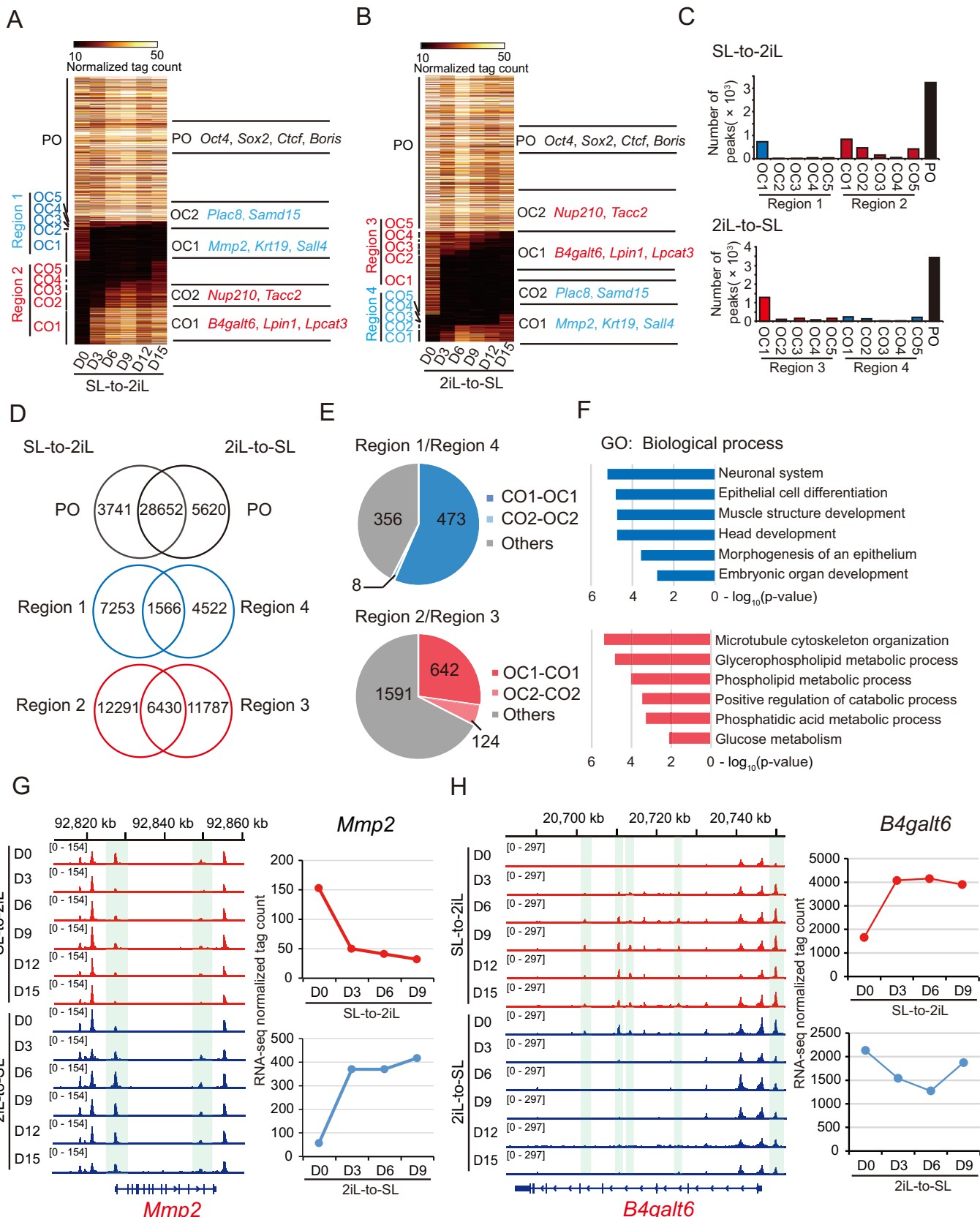

◀ **Figure 1.** Dynamics of chromatin accessibility during the interconversion between SL-ESCs and 2iL-ESCs.

(A, B) Chromatin loci arranged into groups according to time and status of being closed or opened, closed to open (CO) or open-to-closed (OC), or permanently open (PO) during the transition from SL-to-2iL (A) and 2iL-to-SL (B). Representative genes are noted for each subgroup on the right. (C) Number of peaks defined in CO/OC and PO for (A, B). (D) Venn diagrams of CO/OC and PO peaks during interconversion between SL-ESCs and 2iL-ESCs. (E) Statistics of the number of genes that were switched at different time points of interconversion between SL-ESCs and 2iL-ESCs on the loci of Region 1/4 and Region 2/3. (F) GO analysis of 481 genes in Region 1/4-CO1/OC1 and CO2/OC2, and 766 genes in Region 2/3-OC1/CO1 and OC2/CO2 in (E). The enrichment p-value was calculated by Metascape software. (G, H) Representative loci of Mmp2 and B4galt6 within Region 1/4 (G) and Region 2/3 (H) defined by ATAC-seq during the transition between SL-ESCs and 2iL-ESCs, respectively (left). Expression values of Mmp2 and B4galt6 from RNA-seq data (right).

RNA-seq data and RT-qPCR results revealed a general upregulation of these genes during the SL-to-2iL transition and a downregulation during the 2iL-to-SL transition (Fig. EV3A,B). In conclusion, we identified these five potential regulators as key players in controlling ground-state pluripotency.

## TEAD2 assumes pivotal role in establishing the ground-state pluripotency

To scrutinize the roles of these five factors in cell fate transitions, we designed three small interfering RNAs (siRNAs) for each factor targeting *Tead2*, *Tead4*, *Esrrb*, *Nr5a2*, and *Tcfcp2l1*, respectively. These siRNAs were transfected into mESCs every 3 days during SL-to-2iL transition, and knockdown efficiencies were examined by RT-qPCR (Fig. 2C). Our findings revealed that knocking down either *Tead2* (si*Tead2*) or *Nr5a2* (si*Nr5a2*) impeded domed colony formation during SL-to-2iL transition, with *Tead2* knockdown exhibiting a more pronounced effect than *Nr5a2* knockdown (Fig. 2D–F). Conversely, knockdown of the other three factors (si*Tead4*, si*Esrrb*, and si*Tcfcp2l1*) had minimal effects on cell morphology (Fig. 2D–F). With the exception of si*Tead4*, knockdown of any of the other four factors diminished self-renewal and proliferation capacity to varying degrees, with si*Tead2* and si*Tcfcp2l1* also impacting day 0 during the transition (Fig. 2G). ESRRB and TCFCP2L1 play crucial roles in stabilizing the regulatory network of naïve pluripotency and preventing the loss of self-renewal (Atlasi et al, 2019; Festuccia et al, 2018a; Festuccia et al, 2018b; Hackett and Surani, 2014; Qiu et al, 2015; Zhang et al, 2021). NR5A2 can form a regulatory module with ESRRB to assist in the binding of core pluripotency factors at most of their occupied regions, thereby regulating the naïve pluripotency network (Festuccia et al, 2021). However, the mechanism by which TEAD2 regulates ground-state pluripotency remains unknown. Despite displaying a flattened clone-like morphology, *Tead2*-knockdown cells retain their pluripotency, similar to cells with another factor knockdowns (Fig. 2E). We speculated that TEAD2 may not directly influence the core pluripotency but instead regulates the formation of the ground-state pluripotency during SL-to-2iL transition.

We measured TEAD2 expression levels in both mRNA and protein in both cell types and observed that TEAD2 expression in 2iL-ESCs was approximately 1.5-fold higher than that in SL-ESCs (Fig. 3A). To assess the importance of TEAD2 in 2iL-ESCs, we transfected *Tead2* siRNAs into both 2iL- and SL-ESCs. The results showed that knockdown of *Tead2* had little effect on the expression of core pluripotent factors *Oct4*, *Sox2*, and *Nanog* (the change was less than 1.2 times) in both ESCs (Fig. EV4A–D). However, *Tead2* knockdown induced heterogeneity in 2iL-ESCs, exhibiting a morphology similar to that of SL-ESCs (Fig. EV4B). During spontaneous differentiation after removing 2i/LIF or LIF, 2iL-

differentiated cells with *Tead2* knockdown resembled SL-differentiated cells in morphology (Fig. EV4B). *Tead2* knockdown had a minor effect on SL-ESCs (Fig. EV4B). Furthermore, *Tead2* knockdown upregulated serum-specific genes and downregulated 2i-specific genes in 2iL-ESCs (Fig. EV4E). In contrast, *Tead2* knockdown in SL-ESCs did not induce upregulation of serum-specific genes and had no or minor effect on downregulation of 2i-specific genes (Fig. EV4F). These results suggest that TEAD2 does not participate in regulating the core pluripotency of mESCs but instead stabilizes the ground-state regulatory network of 2iL-ESCs to prevent them from entering a metastable state.

## *Tead2* knockout fails to activate 2i-specific genes and to repress a few serum-specific genes during SL-to-2iL transition

To further investigate how TEAD2 regulates the establishment of ground-state pluripotency, we generated *Tead2* knockout ESC lines cultured in serum conditions by using CRISPR/Cas9 genome editing technology (Fig. 3B) and confirmed by PCR and Sanger sequencing (Fig. EV5A). We obtained both *Tead2*$^{-/-}$ and *Tead2*$^{+/-}$ lines (Fig. EV5B). Both homozygous (*Tead2*$^{-/-}$) and heterozygous (*Tead2*$^{+/-}$) knockouts had no effect on ESC morphology and the expression of core pluripotency marker genes in serum/LIF conditions (Appendix Fig. S1A–D). Consistent with the above results, knockout of *Tead2* disrupted colony formation during SL-to-2iL transition (Fig. 3C). The loss of *Tead2* resulted in cell deformation but did not lead the cells to exit pluripotency during SL-to-2iL transition; instead, the cells underwent spontaneous differentiation after removing 2i and LIF (Appendix Fig. S1E). The expression of core pluripotent factors *Oct4*, *Sox2*, and *Nanog* also showed no significant alteration (the change was less than 1.5-fold) in both wild-type and *Tead2*-knockout cells on day 0 and day 6 of the transition (Appendix Fig. S1F).

To investigate the molecular mechanism of TEAD2 in regulating SL-to-2iL transition, we performed RNA-seq experiments in both wild-type and *Tead2*-knockout SL-ESCs during SL-to-2iL transition (day 0, 3, and 6). PCA showed that *Tead2* loss altered the route during the transition from SL-to-2iL condition compared with wild-type cells (Fig. 3D). In order to analyze the abnormally expressed genes, we identified 2278 upregulated genes and 2128 downregulated genes in 2iL-ESCs compared with SL-ESCs (q-value < 0.05 and fold-change >2), defining them as 2i-specific genes and serum-specific genes, respectively (Appendix Fig. S1G; Dataset EV2). We then categorized the differentially expressed genes (DEGs) into five groups based on their expression patterns following *Tead2* loss (Fig. 3E). Gene expression levels of cluster 1 (C1, 961 genes) were unchanged; cluster 2 (C2, 663 genes) exhibited a slight downregulation in SL-ESCs but were not affected

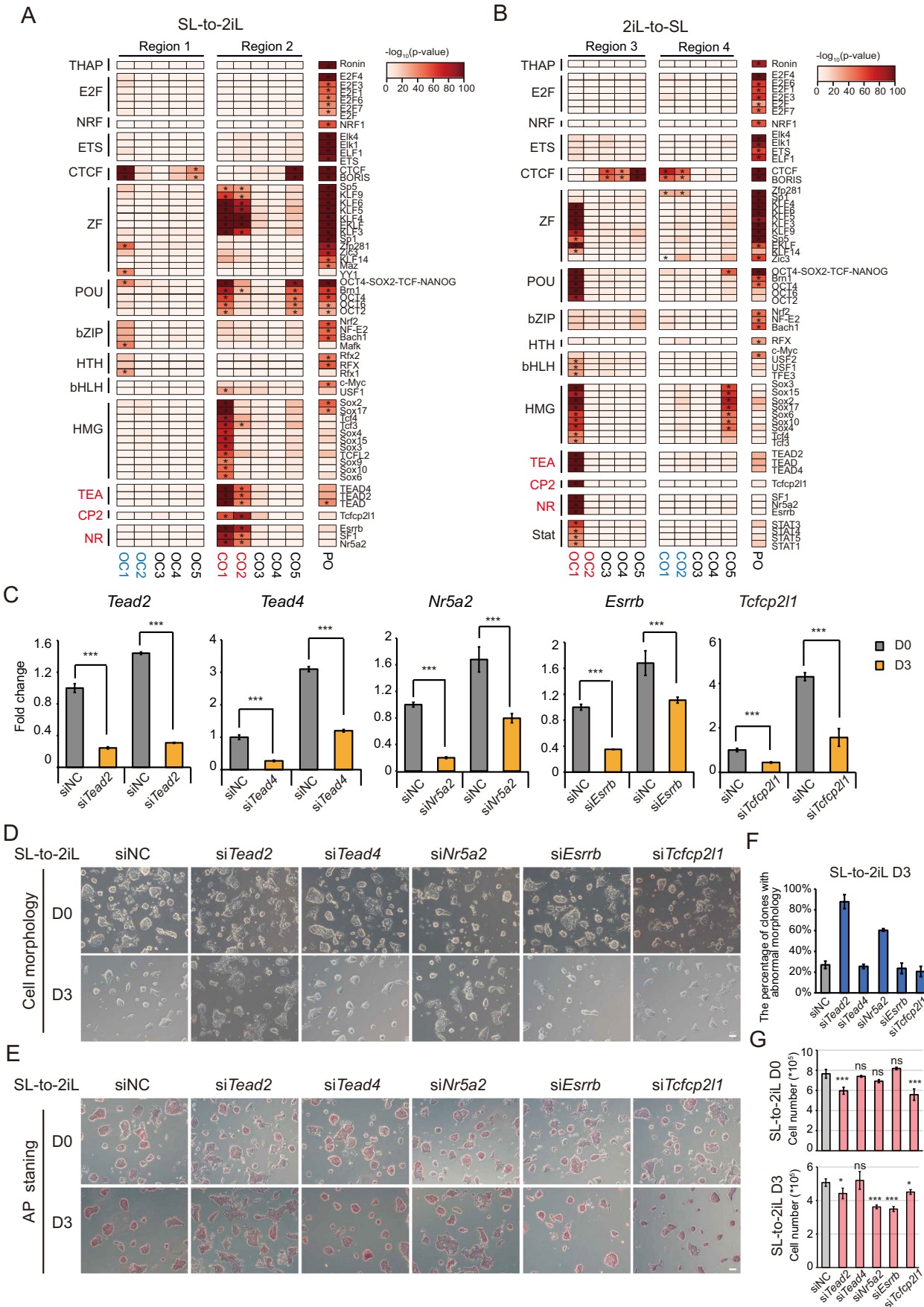

Figure 2. TEAD2 regulates the transition between 2iL-ESCs and SL-ESCs.

(A, B) TF motifs are significantly enriched in CO/OC/PO categories of ATAC-seq peaks during the transition from SL-to-2iL (A) and 2iL-to-SL (B). The motifs for TFs are indicated on the right of the heatmap. *$p < 1e{-}30$. $P$-value was calculated by hypergeometric enrichment calculations from Homer software. (C) RT-qPCR testing siRNA knockdown efficiencies for *Tead2*, *Tead4*, *Tcfcp2l1*, *Nr5a2*, and *Esrrb* during SL-to-2iL transition. Cells were treated with specific siRNAs for every 3 days along with control siRNA. Data are shown as the mean ± SD. Indicated significances are tested using Student's $t$-test analyses (***$p < 0.001$). $n = 3$ biological replicates. (D) Representative images of cells transfected with siNC (negative control) and siRNAs for *Tead2*, *Tead4*, *Tcfcp2l1*, *Nr5a2*, and *Esrrb* during the SL-to-2iL process. Scale bar, 100 μm. (E) Representative images of AP staining of cells transfected with siNC (negative control) and siRNAs for *Tead2*, *Tead4*, *Tcfcp2l1*, *Nr5a2*, and *Esrrb* during SL-to-2iL process. Scale bar, 100 μm. (F) The percentage of clones with abnormal morphology on day 3 of SL-to-2iL transition in (E). Data are shown as mean ± SD of three independent fields of view. (G) Number of cells transfected with siNC (negative control) and siRNAs of *Tead2*, *Tead4*, *Tcfcp2l1*, *Nr5a2*, and *Esrrb*. Cells were grown for 3 days during the SL-to-2iL process. Data are presented as the mean ± SD of three independent wells. Indicated significances are tested using Student's $t$-test analyses (*$p < 0.05$, ***$p < 0.001$). $n = 3$ biological replicates. Source data are available online for this figure.

during the transition; and cluster 4 (C4, 1101 genes) was normally upregulated during the transition. Cluster 3 genes (C3, 472 genes), involved in muscle structure and utero embryonic development (such as *Mmp2* and *Ank*), demonstrated high expression after *Tead2* loss during SL-to-2iL transition (Fig. 3E,F; Appendix Fig. S1H; Dataset EV3). Concurrently, cluster 5 genes (C5, 1210 genes), associated with carbohydrate and lactate metabolic processes (such as *B4galt6*, *Kit*, *Idh2*, and *Ldhb*), experienced downregulation after *Tead2* loss during SL-to-2iL transition (Fig. 3E,G; Appendix Fig. S1I; Dataset EV3). Furthermore, *Tead2*-knockout cells exhibited persistent cellular phenotype, abnormal gene expression pattens, and cluster 5 gene changes even after long-term culture in 2i/LIF (Appendix Fig. S2A–D). Consistent with the results at D6, *Tead2* knockout led to the downregulation of 2i-specific genes and the upregulation of serum-specific genes at D15 and D21, respectively, during the transition (Appendix Fig. S2E,F). These findings underscore the crucial role of TEAD2 in activating a sub set of 2i-specific genes during SL-to-2iL transition.

### *Tead2* overexpression did not enhance SL-to-2iL transition but conferred SL-ESCs with expression of partial 2i-specific genes

To investigate the impact of *Tead2* overexpression on SL-to-2iL conversion, we ectopically expressed *Tead2* in SL-ESCs (Appendix Fig. S3A,B) and conducted SL-to-2iL transition experiments in both control and *Tead2*-overexpressed mESCs. *Tead2*-overexpressed cells exhibited normal morphological changes compared to control cells (Appendix Fig. S3C). *Tead2* overexpression had no discernible effect on AP staining and the expression levels of core pluripotent genes on day 6 of the transition (Appendix Fig. S3D,E). Meanwhile, most 2i- and serum-specific genes showed minimal changes on day 6 of the transition but were significantly up- and down-regulated, respectively, on day 0 (Appendix Fig. S3F,G). These results suggest that *Tead2* overexpression does not impact SL-to-2iL transition but induces the expression of 2i-specific genes in SL-ESCs.

### TEAD2 occupies more binding sites in 2iL-ESCs and binds to active chromatin regions to regulate the expression of 2i-specific genes

We next explored the binding pattern of TEAD2 in 2iL-ESCs and compared it with SL-ESCs. Unfortunately, attempts to perform chromatin immunoprecipitation followed by high-throughput sequencing (ChIP-seq) for TEAD2 using commercial antibodies

were unsuccessful. In response, we generated stable 2iL- and SL-ESC lines with endogenous expression of biotin-tagged TEAD2 through CRISPR/Cas9 technique (Fig. 4A), confirmed by Western blot (Fig. 4B). Subsequent BIOTIN ChIP-seq experiments identified 24,994 and 5837 peaks in 2iL-ESCs and SL-ESCs, respectively. Motif enrichment analysis indicated significant enrichment of TEAD2 binding motifs in both 2iL- and SL-ESCs (Fig. 4C). Notably, 10,315 specific peaks were identified in 2iL-ESCs, while only 47 were specific to SL-ESCs, suggesting a potential regulatory role for TEAD2 in 2iL-ESCs (Fig. 4D,E). These 2i-specific peaks predominantly enriched in intergenic regions with more open chromatin regions (Fig. 4F,G). About 43.13% (4449/10,315) of TEAD2 peaks localize to either promoters or enhancers (Fig. 4H). This observation implies the potential involvement of TEAD2 in gene expression regulation.

To discern whether TEAD2 functions as an activator or repressor, we conducted the analysis of the binding relationships between TEAD2 and active/repressive histone marks in 2iL- and SL-ESCs. Utilizing published ChIP-seq data for histone marks H3K36me3, H3K4me1, H3K4me3, H3K27ac, H3K27me3, and H3K9me3 in 2iL- and SL-ESCs (Aljazi et al, 2020; Joshi et al, 2015; Marks et al, 2012), our investigation revealed that 2i-specific TEAD2 sites predominantly marked by active histone marks, notably H3K27ac, H3K4me1, and H3K4me3 (Fig. 4I). Furthermore, these active histone marks exhibited a more pronounced enrichment at these sites in 2iL-ESCs compared to SL-ESCs (Fig. 4I). We then identified the genes directly bound by TEAD2, with approximately 15.9% (192/1210) of cluster 5 genes being targeted by TEAD2 (Fig. 4J; Dataset EV4). Notably, these 192 genes exhibited reduced expression following *Tead2* loss (Fig. 4K). GO analysis indicated their involvement in glycolipid metabolic processes and lipid catabolic processes (Fig. 4L), exemplified by *B4galt6* (Fig. 4M). These findings suggest that TEAD2 binds to both promoters and enhancers of 2i-specific genes, directly influencing their expression.

### TEAD2 governs switching of A/B compartments during SL-to-2iL transition

Transcription factors can serve as anchor proteins orchestrating cell-type-specific 3D genome architecture (Kim and Shendure, 2019; Stadhouders et al, 2019). To investigate whether TEAD2 shapes the 3D genome organization of 2iL-ESC identity during the SL-to-2iL transition, we conducted Bridge Linker-Hi-C (BL-Hi-C) experiments (Liang et al, 2017) and examined the 3D characteristics on day 6 of SL-to-2iL transition with or without

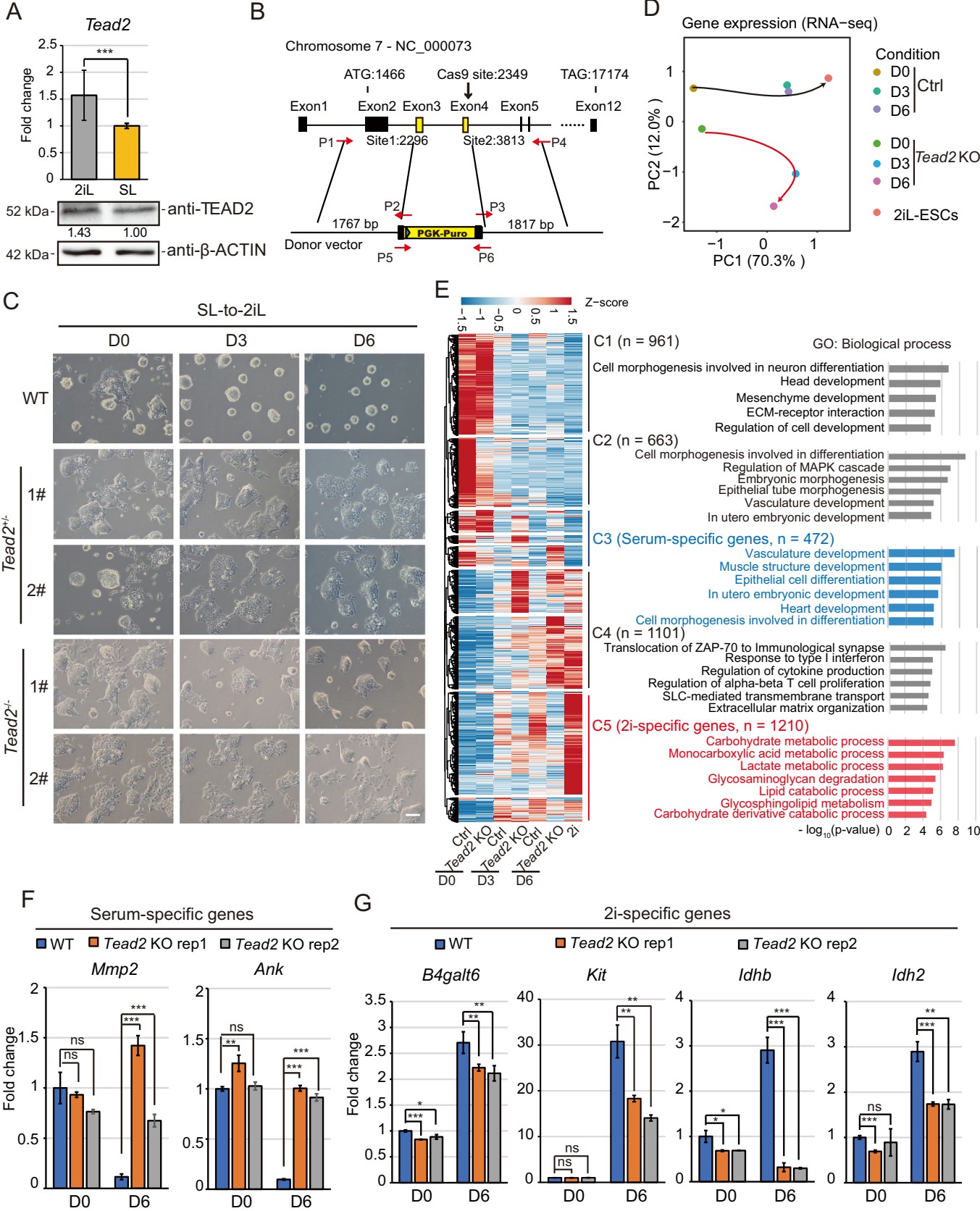

**Figure.** Panels A–G. (A) Tead2 fold change and anti-TEAD2 / anti-β-ACTIN western blots in 2iL and SL conditions. (B) Schematic of Chromosome 7 - NC_000073 targeting strategy. (C) SL-to-2iL morphology at D0, D3, D6 for WT, Tead2+/−, Tead2−/−. (D) Gene expression (RNA-seq) PCA. (E) Heatmap of clusters C1–C5 with GO: Biological process. (F) Serum-specific genes Mmp2, Ank. (G) 2i-specific genes B4galt6, Kit, Idhb, Idh2.

**Figure 3.   Effect of *Tead2*-knockout on colony formation and gene expression during SL-to-2iL transition.**

(A) RT-qPCR and Western blot examining TEAD2 expression in both 2iL- and SL-ESCs. mRNA expression was tested in triplicate in three independent experiments. Data are presented as the mean ± SD. *P*-values were determined using two-sided Student's *t*-test (***$p < 0.001$). Quantification of protein signal was performed using Fiji image analysis software. (B) Strategy of generating *Tead2*-knockout cell lines in SL-ESCs. (C) Representative cellular morphologies of wild-type ESCs and two clones of *Tead2*$^{-/-}$ and *Tead2*$^{+/-}$ ESCs during the SL-to-2iL transition at day 0, 3, and 6. Scale bar, 100 μm. (D) PCA of the RNA-seq data from wild-type ESCs and *Tead2*-knockout ESCs collected at different time points during the SL-to-2iL transition. (E) Heatmap of the expression of 2i- and serum-specific genes during the SL-to-2iL transition with or without *Tead2*-knockout. Clustering analysis and the enrichment of GO terms in each group of genes are also provided. The enrichment *p*-value was calculated by using Metascape software. (F, G) RT-qPCR analysis testing the expression of serum-specific genes (F) and 2i-specific genes (G) in wild-type and *Tead2*-knockout cells on day 0 and day 6 of the transition. Data are presented as the mean ± SD. Indicated significances are testing using Student's *t*-test analyses (*$p < 0.05$, **$p < 0.01$, ***$p < 0.001$). $n = 3$ biological replicates. Source data are available online for this figure.

*Tead2* knockout. Our BL-Hi-C data exhibited high reproducibility across different samples (Appendix Fig. S4A,B). Correlation analysis revealed minimal differences in TADs between wild-type and *Tead2* knockout samples (Appendix Fig. S4C). However, a distinct switch of A/B compartments was observed in *Tead2* knockout ESCs compared to wild-type ESCs on day 6 of SL-to-2iL transition (6.82% B to A, 4.48% A to B) (Appendix Fig. S4D,E). This compartment switch could account for the abnormal activation of certain serum-specific genes, such as *Mmp2* and *Arhgef26* (Appendix Fig. S4F,G), consistent with their abnormal upregulation after *Tead2* knockout during the transition (Appendix Fig. S4H,I). These results indicate that *Tead2* loss leads to a switching of A/B compartments, subsequently causing aberrant activation of some serum-specific genes.

## TEAD2 enhances the expression of partial 2i-specific genes by regulating EP interactions

As TEAD2 directly binds to enhancers and promoters in 2iL-ESCs, we investigated the impact of *Tead2* loss on enhancer marks H3K4me1 and H3K27ac at TEAD2 binding sites. CUT&Tag analysis for wild-type and *Tead2*-knockout cells revealed minimal alteration in the enrichment of both H3K4me1 and H3K27ac marks at these sites, suggesting that loss of TEAD2 does not diminish enhancer activity (Appendix Fig. S5A,B). Considering this, we explored whether TEAD2 influences the expression of 2i-specific genes by mediating EP interactions during the SL-to-2iL transition. Analyzing TEAD2 binding peaks alongside BL-Hi-C data, we observed a significant reduction (28.95%) in interactions between TEAD2-occupied enhancers and promoters following *Tead2* knockout (Fig. 5A). Categorizing the interactions post-*Tead2* loss, we noted a notably higher decrease in the degrees of TEAD2-mediated EP interactions compared to all EP interactions and all interactions (Fig. 5B). Furthermore, *Tead2* loss resulted in a substantial decrease in contact frequency for 3391 TEAD2-mediated EP interactions, surpassing the increased contact frequency of 1438 interactions (Fig. 5C). And the degree of decreased contact frequency was much stronger than the increased contact frequency (Fig. 5D). Aggregate peak analysis (APA) scores confirmed the normal increase in frequency for these EP interactions during the SL-to-2iL transition but a significant decrease after *Tead2* loss (Fig. 5E), indicating disruption caused by *Tead2* knockout in the frequency of EP interactions during this transition. In addition, 190 2i-specific genes with downregulated expression in cluster 5 exhibited a marked decrease in EP interactions following *Tead2* knockout (Fig. 5F). GO analysis revealed their involvement in negative regulation of stem cell

differentiation and the phospholipid biosynthetic process (exemplified by *B4galt6*) (Fig. 5G,H; Dataset EV5). These findings suggest that TEAD2 regulates the expression of 2i-specific genes through TEAD2-mediated EP interactions during the SL-to-2iL transition.

## Mediation of EP interactions by TEAD2 may involve Cohesin but not structural proteins such as YY1 and CTCF

EP interactions are known to be facilitated by architectural proteins, including Mediator, Cohesin complexes, and DNA-binding proteins such as CTCF and YY1 (Arzate-Mejia et al, 2018; Gómez-Díaz and Corces, 2014; Hu et al, 2020; Weintraub et al, 2017). To ascertain the dependence of TEAD2-mediated EP interactions on these architectural proteins, we examined the locations of CTCF, YY1, and SMC1 (the main subunit of Cohesin) in 3391 decreased TEAD2-mediated EP interactions using the published ChIP-seq database in 2iL-ESCs (Atlasi et al, 2019). Subdividing these interactions into 2614 promoter anchor regions and 2410 enhancer anchor regions, we observed significant co-localization of both YY1 and SMC1 with TEAD2 in the identified promoter and enhancer anchor regions (Fig. 6A,B). In contrast, CTCF exhibited relatively weak co-localization with TEAD2 (Fig. 6A,B). These indicate that TEAD2-mediated EP interactions may collaboratively function with YY1 and SMC1. YY1 and SMC1 ChIP-seq experiments were conducted in cells on day 6 during the SL-to-2iL transition upon *Tead2* knockout. The data reveal that *Tead2* loss has no effect on YY1 occupancy but leads to a slight decrease in SMC1 occupancy at these enhancer anchor regions (Fig. 6C,D). These findings suggest that the reduction of TEAD2-mediated EP interactions is not attributed to changes in YY1 binding at the anchor regions but may function in conjunction with SMC1.

## Mutation of TEAD2 binding motifs results in loss of EP interactions in 2i-specific *B4galt6* gene

To further explore whether EP interactions in 2i-specific genes are mediated by TEAD2 binding sites in 2iL-ESCs, we selected the 2i-specific gene, *B4galt6*, as a representative case. *Tead2* knockdown led to decreased expression of *B4galt6* in 2iL-ESCs (Appendix Fig. S6A), consistent with its reduced expression at day 6 of the SL-to-2iL transition (Fig. 3G). Based on the endogenous biotin-tagged TEAD2 ChIP-seq data in 2iL-ESCs, two putative TEAD2 binding motifs were identified in the promoter region of the *B4galt6* gene. Subsequently, base substitutions were introduced into these two TEAD2 binding motifs in 2iL-ESCs using CRISPR/Cas9 (Fig. 6E).

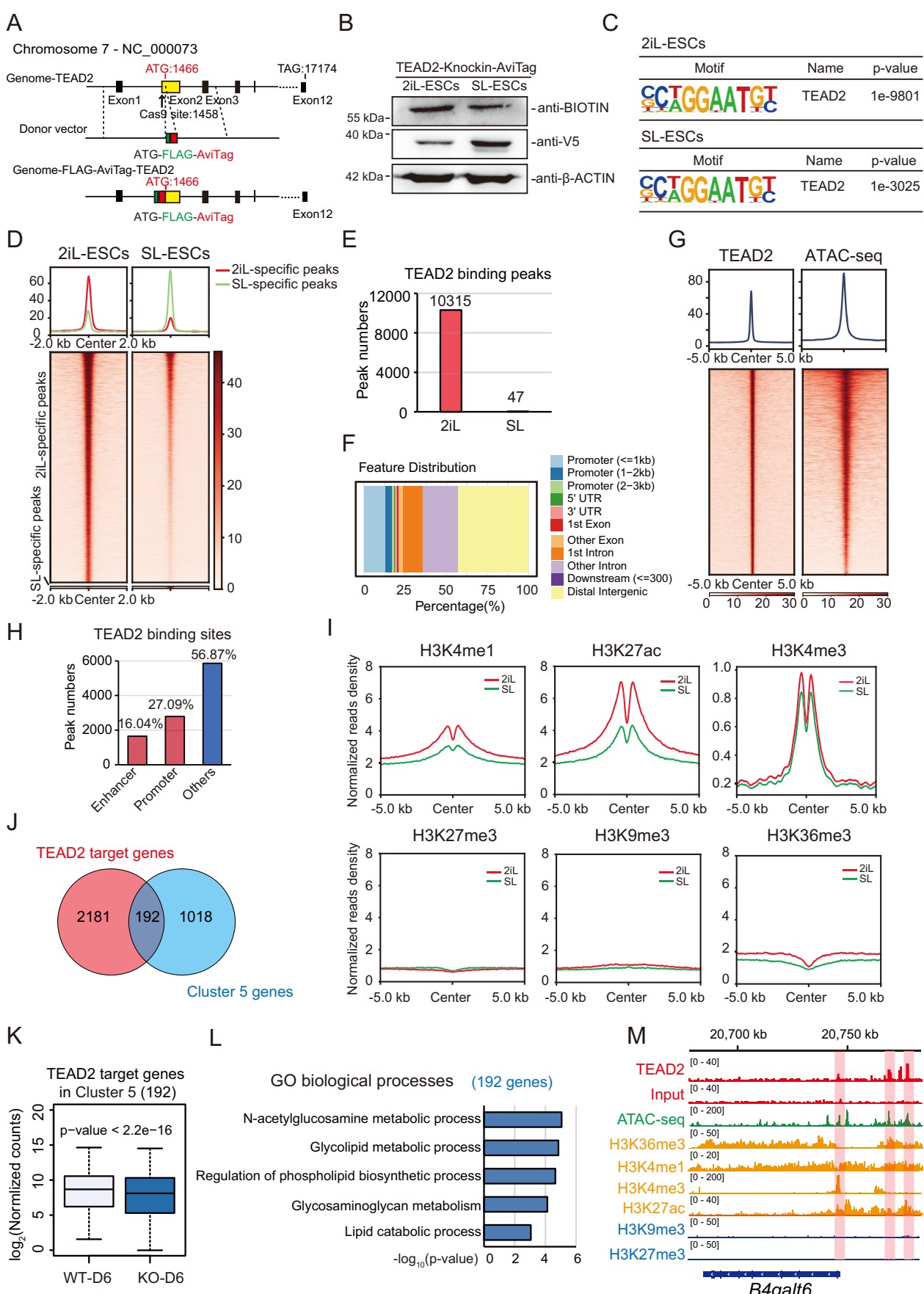

◀ **Figure 4.  TEAD2 binds to the active chromatin regions of 2i-specific genes.**

(A) Strategy for generating of *Tead2*-FLAG-AviTag knock-in cell lines in 2iL- and SL-ESCs. (B) Western blot analysis for BIOTIN and V5 with cell lysates from *Tead2*-FLAG-AviTag-knock-in 2iL- and SL-ESC lines. β-ACTIN was used as a loading control. (C) Motif-enrichment analysis of BIOTIN-binding sites in 2iL- and SL-ESCs. *P*-value was calculated by using hypergeometric enrichment calculations from Homer software. (D) Heatmap showing the comparison of TEAD2 binding sites between 2iL- and SL-ESCs. (E) Number of 2iL- and SL-ESCs-specific TEAD2 binding peaks. (F) Pie charts showing the genomic distribution of 2i-specific TEAD2 peaks. (G) Heatmaps of sequence read density for ATAC-seq in 2i-specific TEAD2 binding peaks. (H) Bar plot showing the number of 2i-specific TEAD2 peaks that overlap with both promoters and enhancers. (I) Tag-density pileup showing H3K4me1, H3K27ac, H3K4me3, H3K27me3, H3K9me3, and H3K36me3 ChIP-seq signals at the 2i-specific TEAD2 binding sites in both 2iL- and SL-ESCs. (J) Venn plots showing the overlap among 2i-specific TEAD2 target genes and cluster 5 genes. (K) Boxplots showing expression level of overlapping genes in (J) between wild-type and *Tead2*-knockout cells at day 6 of the transition. The centerline indicates the median value, while the box and whiskers represent the interquartile range (IQR) and 1.5 × IQR, respectively, $n = 192$. *P*-value was calculated by Mann–Whitney U test. (L) GO categories of the overlapping genes shown in (J). The enrichment *p*-value was calculated by using Metascape software. (M) Genomic views of enrichment for TEAD2 and histone modifications in the *B4galt6* gene. Source data are available online for this figure.

By digesting the genomic DNA with *XhoI* and *HindIII* (Appendix Fig. S6B) and performing Sanger sequencing, we finally yielded two homozygous clones with both TEAD2 binding motifs mutated (Appendix Fig. S6C,D). RT-qPCR results indicated that the loss of TEAD2 motifs at the gene promoter in these two mutant clones had no effect on the expression of *Tead2* (Fig. 6F) but resulted in lower expression of *B4galt6* (Fig. 6G). To further demonstrate that the downregulation of *B4galt6* gene expression in the mutant clones resulted from the attenuation of TEAD2-mediated EP interactions, quantitative high-resolution chromosome conformation capture copy (QHR-4C) experiments were performed in wild-type and two mutant 2iL-ESCs, wild-type and *Tead2*-knockout cells at day 6 of transition. The results showed that, similar to *Tead2* knockout, the frequency of EP interactions at the *B4galt6* gene locus with TEAD2 binding peaks was significantly reduced in the two mutant clones compared to wild-type 2iL-ESCs (Fig. 6H and Appendix Fig. S6E). *Tead2* knockout had no effect on the levels of H3K4me1 and H3K27ac on these EP interactions at day 6 of SL-to-2iL transition (Fig. 6H). In addition, there was no change in H3K27ac enrichment at the *B4galt6* locus in either of the mutant clones (Appendix Fig. S6F). These results collectively demonstrate that TEAD2 contributes to EP interactions for 2i-specific genes.

In summary, this study reveals that TEAD2 is crucial for maintaining ground-state pluripotency by regulating the expression of 2i-specific genes through TEAD2-mediated EP interactions. Furthermore, TEAD2 may function in collaboration with the cohesion subunit SMC1 to mediate EP interactions, rather than with structural proteins such as YY1 and CTCF.

## Discussion

Numerous endeavors have sought to unravel unidentified regulatory factors and the fundamental mechanisms at play. The pivotal occurrence determining the transformation of ESCs from a metastable state to the ground-state pluripotency is the establishment or exit of ground-state pluripotency. During this transition, the transcription factors governing the expression of 2i-specific genes assume crucial roles. In the present study, we systematically mapped chromatin accessibilities, revealing over 60 specific transcription factors potentially mediating the interconversion of 2iL-ESCs and SL-ESCs (Fig. 2A,B).

Among these, TEAD2 emerged as a prominent transcription factor important for transitioning from SL-ESCs to 2i-ESCs and maintaining the ground-state pluripotency. TEAD transcription factors possess N-terminal domains (TEA) binding to DNA and

C-terminal domains (YBD) interacting with YAP/TAZ (Anbanandam et al, 2006; Bürglin, 1991). Individually, TEA and YBD exhibit high homology within the TEAD family (Appendix Fig. S7A). Despite this, TEADs serve diverse functions during early embryonic development and various organogenesis processes (Currey et al, 2021). Our findings disclosed higher expression levels of *Tead1-4* in 2iL-ESCs compared to SL-ESCs, with *Tead1* being predominant, *Tead3* barely detectable, and *Tead2* and *Tead4* at comparable levels (Appendix Fig. S7B). TEAD3 may play a minor role in ESCs. Previous study showed that knocking TEAD1/3/4 down in ESCs results in downregulation of both *Oct4* and *Sox2* and loss of pluripotency (Lian et al, 2010). In contrast, knockdown of *Tead2* had little effect on the expression of core pluripotent factors in ESCs (Fig. EV4C,D). We postulate that TEAD1 and TEAD2 function differently by interacting with diverse co-activators to modulate the pluripotency of stem cells. For TEAD4, despite the absence of a discernible phenotype following *Tead4* depletion (Fig. 2C–G), RNA-seq experiments were conducted to explore potential redundancy between TEAD2 and TEAD4 and identify genes regulated by *Tead4* knockdown. Consistently, *Tead4* knockdown minimally affected gene expression during the transition (Appendix Fig. S7C–E), suggesting that TEAD2, but not TEAD4, modulates the SL-to-2iL transition. And TEAD4 is absent in nucleus of ESCs (Home et al, 2012), in which might compromise the function as TEAD4 in regulating SL-to-2iL transition.

It is established that gene activation in 2iL- and SL-ESCs primarily relies on TF binding and is modulated by hardwired EP interactions. Our data indicate that TEAD2 binds to active regions of 2i-specific genes, activating their expression by regulating EP interactions. This revelation of TEAD2's role in modulating EP interactions adds to the understanding of the 3D genome. Steroid receptor coactivator (SRC) and PARP protein, acting as transcriptional co-activators of TEAD2, maybe involved in regulating chromatin conformation (Belandia and Parker, 2000; Landin-Malt et al, 2016). However, the mechanism underlying how TEAD family proteins regulate chromatin interactions remain unclear.

YY1 and CTCF actively bind to both active enhancers and promoter-proximal elements, forming dimers to facilitate EP interactions (Arzate-Mejia et al, 2018; Gómez-Díaz and Corces, 2014; Hu et al, 2020; Weintraub et al, 2017). The DNA-binding activity of TEAD transcription factors is localized within their N-terminal domains (TEAD-DBD) (Anbanandam et al, 2006; Bürglin, 1991). The TEAD-DBD, with a truncated L1 loop, can form homodimers through domain swapping, thereby regulating the DNA selectivity of TEAD proteins (Lee et al, 2016). To ascertain whether TEAD2 can form dimers in vivo, we generated

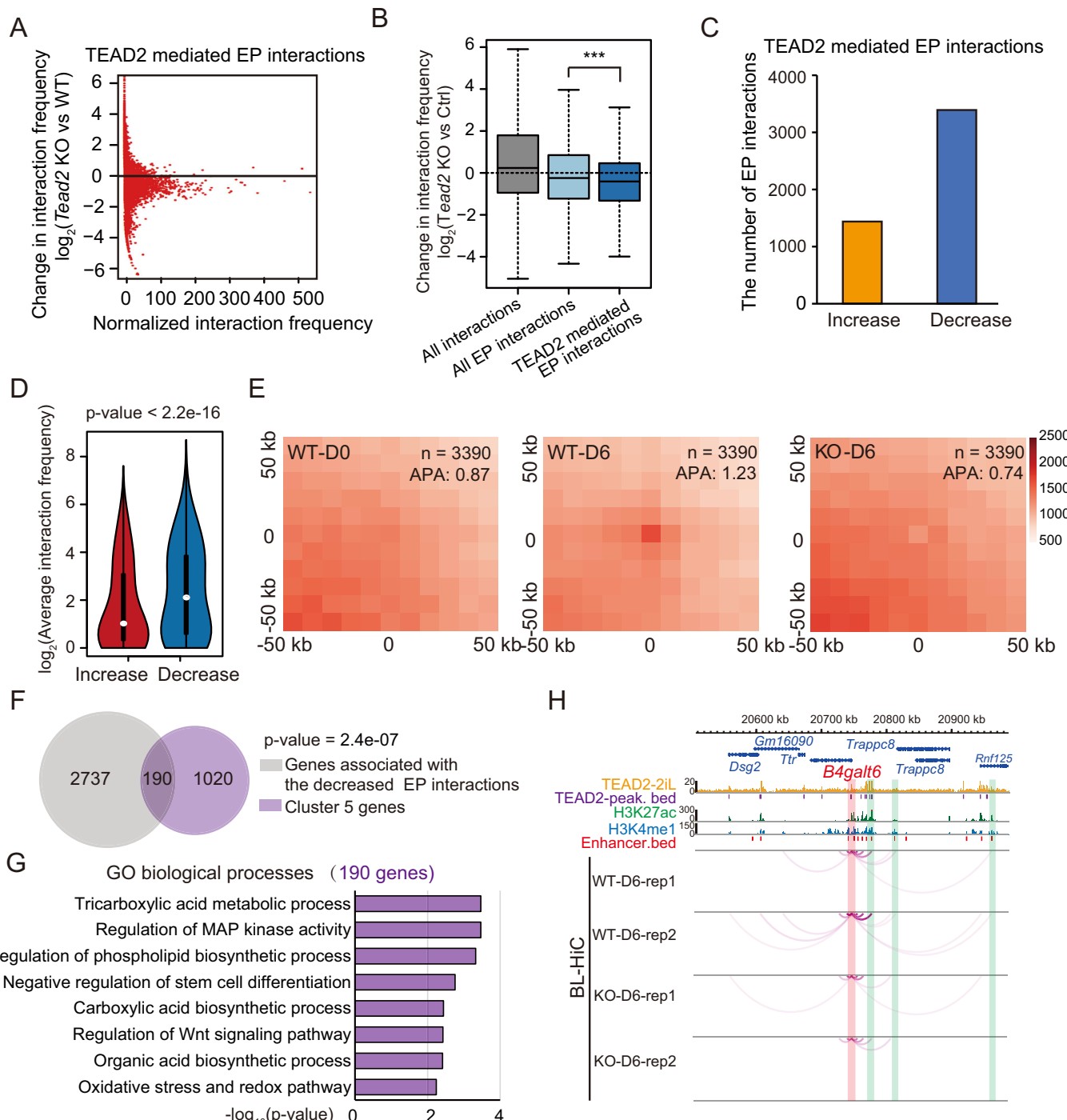

and expressed both FLAG-tagged and HA-tagged TEAD2 plasmids in cells. Subsequent FLAG co-IP experiments, after elution with high salt concentration, confirmed the interaction between HA-tagged TEAD2 and FLAG-tagged TEAD2 (Appendix Fig. S8A). Therefore, we hypothesized that TEAD2 mediates EP interactions through its dimerization, analogous to the mechanisms observed for CTCF and YY1. Cohesin, by folding chromosomes via DNA loop extrusion (Davidson and Peters, 2021), regulates transcription through the formation of long-range EP interactions (Cheng et al, 2022). Our results lead us to conclude that TEAD2 participates in

EP interactions of 2i-specific genes in a similar way as CTCF and YY1, and these interactions might be further stabilized by the SMC1-cohesin complex.

YAP1 and TAZ function as co-activators of TEAD (Pocaterra et al, 2020). Notably, we observed a correlation between the expression of *Taz* and *Tead2* during the interconversion of 2iL- and SL-ESCs (Appendix Fig. S8B). Knocking down *Taz*, but not *Yap1*, disrupted the cell morphology during SL-to-2iL transitions (Appendix Fig. S8C,D). These findings suggest the involvement of TAZ in the regulation of TEAD2 in ground-state pluripotency. The

**Figure 5. *Tead2*-knockout disrupts the EP interactions of 2i-specific genes during the SL-to-2iL transition.**

(**A**) Scatter plot displaying the changes of TEAD2-mediated EP interactions in the normalized interaction frequency ($\log_2$ fold change) between wild-type cells and *Tead2*-knockout cells collected at day 6 during the SL-to-2iL transition. (**B**) Changes in the normalized interaction frequency ($\log_2$ fold change) in cells at day 6 during the SL-to-2iL transition upon *Tead2*-knockout in the following three different categories: all interactions ($n = 701{,}601$), interactions unrelated to TEAD2 binding sites ($n = 84{,}504$), and TEAD2-mediated EP interactions ($n = 16{,}683$). The centerline indicates the median value, while the box and whiskers represent the interquartile range (IQR) and $1.5 \times$ IQR, respectively. \*\*\*$p < 0.001$. *P*-value was calculated by using Mann–Whitney U test. (**C**) Bar plot showing the numbers of the differential TEAD2-mediated EP interactions after *Tead2*-knockout. (**D**) Normalized values of the differential TEAD2-mediated EP interactions frequency (Increased interactions, $n = 1438$; Decreased interactions, $n = 3391$). The centerline indicates the median value, while the box and whiskers represent the interquartile range (IQR) and $1.5 \times$ IQR, respectively. *P*-value was calculated by using Mann–Whitney U test. (**E**) Heatmaps showing APA of differential TEAD2-mediated EP interactions in both wild-type and *Tead2*-knockout cells at day 0 and day 6 of the transition. (**F**) Venn diagrams showing overlap between the genes with decreased EP interactions and genes in cluster 5. *P*-values were calculated with the hypergeometric distribution test. (**G**) GO categories of overlapping genes shown in (**F**). (**H**) Representative genomic locus showing the binding of TEAD2 and H3K27ac, and decreased chromatin interactions after *Tead2*-knockout. *B4galt6* promoter is highlighted with red-shaded rectangles, and its associated enhancers are highlighted with green-shaded rectangles.

specific roles of YAP1 and TAZ in regulating stem cell pluripotency necessitate further investigation. The elucidation of Hippo signaling factors' roles in chromatin structure remains an open question. It is yet to be determined whether TAZ collaborates with TEAD2 to modulate chromatin structure, thereby regulating stem cell ground-state pluripotency.

In summary, we have identified a novel ancillary factor, TEAD2, that initiates ground-state pluripotency by mediating chromatin looping. This study contributes to our understanding of the molecular mechanisms underlying stem cell fate determination and unveils a previously unrecognized molecular function of TEAD2 in higher-order chromatin structure.

# Methods

## Cell culture

HEK293T cells were maintained in DMEM high-glucose media (HyClone) with 10% FBS (Excell) under 5% $CO_2$ at 37 °C. E14Tg2a (E14) ESCs were cultured in standard culture medium on gelatin-coated dishes without feeder cells. For serum/LIF culture, DMEM high-glucose media supplemented with 15% FBS, 1 mM sodium pyruvate (Gibco), 1% nonessential amino acids (Gibco), 1% GlutaMAX (Gibco), 0.1 mM β-mercaptoethanol (Life Technologies), 1% penicillin-streptomycin (Gibco), 1000 U/mL LIF (Millipore). For 2i/LIF culture, mouse ESCs were cultured in 2i medium as previously described (Ying et al, 2008). In total, 500 mL of 2i medium were prepared with 240 mL of DMEM/F12 (Thermo Fisher Scientific), 240 mL of neurobasal (Thermo Fisher Scientific), 2.5 mL of N2 supplement (Thermo Fisher Scientific), 5 mL of B27 supplement (Thermo Fisher Scientific), 1% GlutaMAX, 1% nonessential amino acids, 0.1 mM β-mercaptoethanol, and 1% penicillin-streptomycin, 1000 U/mL LIF, 3 μM CHIR99021 (Selleck), and 1 μM PD0325901 (Selleck). All cells were maintained at 37 °C in an incubator with 5% $CO_2$. The medium was changed every day, and the cells were passaged with 0.25% trypsin-EDTA (Gibco).

## Interconversion of 2iL-ESCs and SL-ESCs

2iL-ESCs and SL-ESCs were cultured on gelatinized plates for 3 days, and were digested with 0.25% trypsin. Then 2iL-ESCs or SL-ESCs were plated into gelatinized six-well plates with $6$–$7 \times 10^4$ cells per well in either serum/LIF or 2i medium. The medium was changed daily.

## Generation of *Tead2*-knockout SL-ESC lines

The CRISPR/Cas9 system was used to genetically engineer ESC lines. For generating of *Tead2*-knockout SL-ESCs, the exon 3 and exon 4 (consisting of TEA domain) of *Tead2* gene sequence containing 43 amino acids were replaced with PGK-Puro by homologous recombination. $5'$ and $3'$ homology arms were amplified from genomic DNA for donor DNA using primers P1/P2 and P3/P4. A loxP-flanked PGK-puromycin cassette was cloned between two homology arms in the pMD18-T vector using primers P5 and P6. The *Tead2*-sgRNA target sequence was inserted into the plasmid pX330. Then, pX330 along with donor vector were transfected into SL-ESCs for gene editing. Next, $1 \times 10^6$ SL-ESCs were transfected with 2 μg of donor DNA, 2 μg of pX330-sgRNA and 12 μL FuGENE® 6 transfection reagent (Promega) according to the manufacturer's instructions. Positive clones were selected by 2 μg/mL puromycin for 5 days. Individual clones were picked and re-plated on gelatin-coated 12-well plates for further screening. The selected colonies were verified by genomic PCR and DNA sequencing. All primers and sgRNA used are listed in Table EV1.

## PCR verification of homozygote or heterozygote clones of *Tead2*-knockout SL-ESCs

PCR was performed using LA Taq (Takara) according to the manufacturer's instructions. 50–100 ng of genomic DNA templates was used in all reactions. $5'$-arm F/R (P7/P8), $3'$-arm F/R (P9/P10) were used to identify whether the segment was inserted in the correct position. Primers F/R1/R2 were used to identify whether the clone was homozygote or heterozygote. Primers including P7 (upstream of $5'$ homology arm) and P8 (in the drug resistance cassette) were used to amplify a 2086 bp product of the $5'$ junction of a targeted integration. Primers including P9 (in the drug resistance cassette) and P10 (downstream of $3'$ homology arm) were used to amplify a 2227 bp product of the $3'$ junction of a targeted integration. All primers used are listed in Table EV2.

## Generation of *Tead2*-FLAG-AviTag knock-in 2iL- and SL-ESC lines

Enzymatic biotinylation with *E. coli* biotin ligase (BirA) is highly specific in covalently attaching biotin to the 15 amino acid AviTag peptide (Fairhead and Howarth, 2015). To generate in vivo biotinylated-TEAD2 in 2iL- and SL-ESC lines, we express BirA in ESCs. Lentivirus for lenti-birA-V5 assembled with psPAX2,

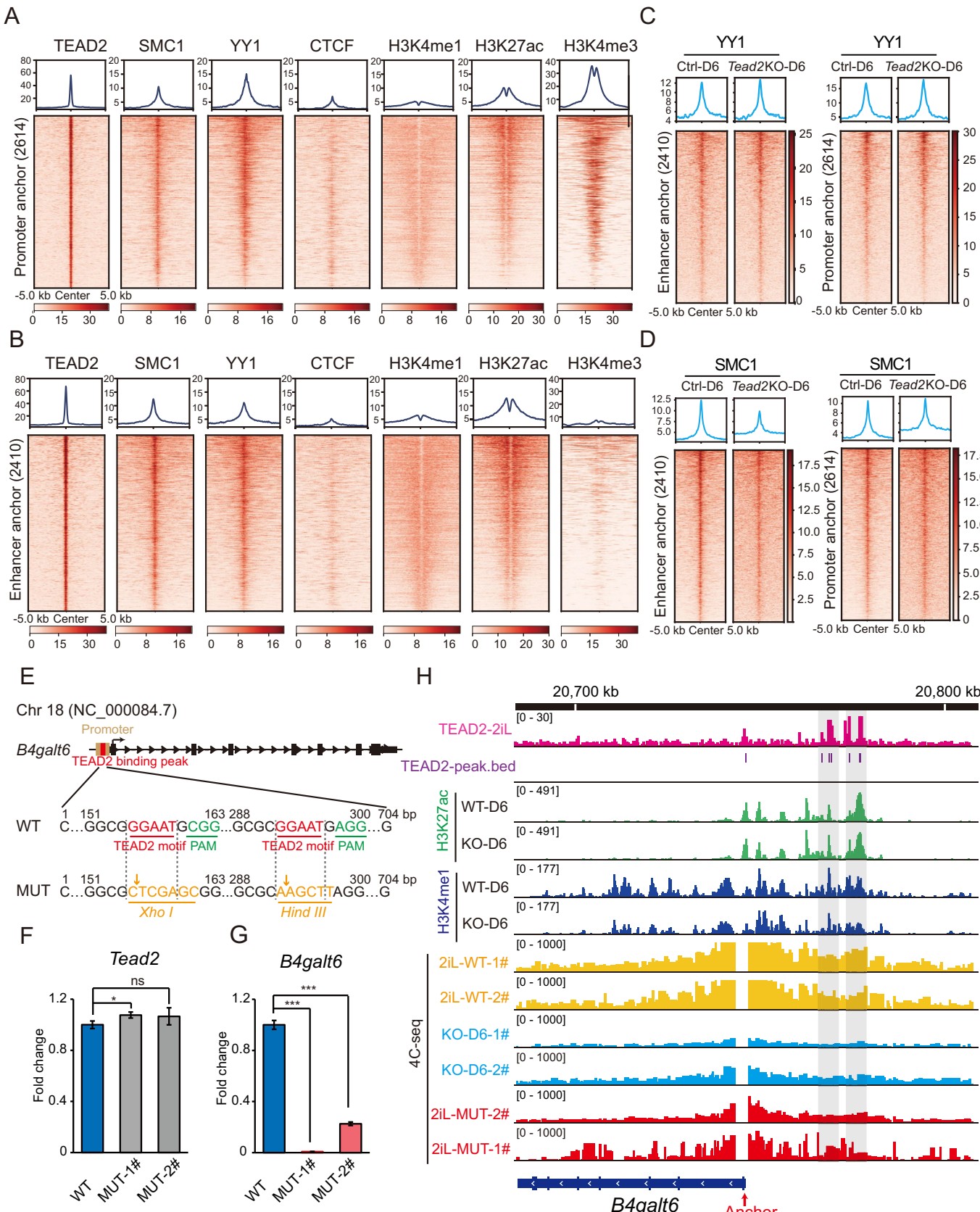

Figure 6. Mediation of EP interactions by TEAD2 may involve Cohesin but not structural proteins such as YY1 and CTCF.

(A, B) Heatmaps showing sequence read density for TEAD2, YY1, SMC1, CTCF, H3K4me1, H3K27ac, and H3K4me3 at promoter (A) and enhancer (B) anchor regions. (C, D) Heatmaps showing read density for YY1 (C) and SMC1 (D) in promoter and enhancer anchor regions at day 6 during the SL-to-2iL transition after *Tead2* knockout. (E) Strategy of generating 2iL-ESC lines with TEAD2 motif mutations at the *B4galt6* promoter region. (F, G) RT-qPCR analysis testing the expression of *Tead2* (F) and *B4galt6* (G) in wild-type 2iL-ESCs and two homozygous clones with base alterations in both TEAD2 binding motifs. Data are presented as the mean ± SD. *P*-values were determined using the two-sided Student's *t*-test (*$p < 0.05$, ***$p < 0.001$). (H) 4C tracks showing the interactions between the promoters and enhancers of *B4galt6* in wild-type and two mutant 2iL-ESCs, wild-type and *Tead2*-knockout cells at day 6 of the transition. The anchor region from QHR-4C is indicated. Source data are available online for this figure.

pMD2.G vectors in HEK293T cells. And lentiviral supernatants were collected and transfected using the modified polyethylenimine (PEI, Polysciences). 2iL-ESCs and SL-ESCs were infected with lenti-birA-V5 lentivirus and selected with 10 μg/mL of blasticidin for at least 5 days. BirA-V5 overexpression was analyzed by Western blot with Anti-V5 Tag monoclonal antibody.

The CRISPR/Cas9 system was used to genetically generate *Tead2*-FLAG-AviTag knock-in ESC lines. An ATG-FLAG-AviTag was inserted upstream of the start codon of exon 2 of *Tead2* gene. The ATG-FLAG-AviTag and the 5′ and 3′ homology arms amplified from the genome were cloned into the pMD18-T vector as a donor construct. The *Tead2*-sgRNA target sequence was inserted into the plasmid pX459. Then, 2 μg of pX459-sgRNA, 2 μg of donor vector were co-transfected with 12 μL FuGENE® 6 transfection reagent into 2iL- and SL-ESCs overexpressing BirA-V5 for gene editing following the manufacturer's protocol. Positive clones were selected by 2 μg/mL puromycin for 5 days. Individual clones were picked and re-plated on gelatin-coated 12-well plates for further screening. The selected colonies were verified by genomic PCR and DNA sequencing. Finally, the in vivo biotinylation of TEAD2 was detected with anti-BIOTIN, HRP-linked antibody (Cell Signaling Technology) with a dilution ratio of 1:1000. The sequences of ATG-FLAG-AviTag and sgRNA used are listed in Table EV1.

## Generation of *Tead2* stably overexpressed SL-ESC lines

*Tead2* cDNAs were cloned into the pSin-Flag vector. The plasmids used for the transfections were purified with a HiPure Plasmid EF Mini Kit (Magen, P1112-03). The sequences of primers used for the *Tead2* CDS amplification are listed in Table EV3. All constructs were confirmed by Sanger sequencing. Then, lentivirus for pSin-Flag, pSin-Flag-*Tead2*, were assembled with psPAX2, pMD2.G vectors in HEK293T cells. Then lentiviral supernatants were collected and transfected using the modified PEI. SL-ESCs were then infected with pSin-Flag and pSin-Flag-*Tead2* lentivirus, respectively. The positive cells were selected with 2 μg/mL puromycin for 5 days.

## Generation 2iL-ESC lines with mutation of TEAD2 motifs in the *B4galt6* promoter region

We generated 2iL-ESCs with mutated TEAD2 motifs by using CRISPR/Cas9. The *B4galt6* promoter region had a TEAD2 peak sequence (704 bp) with two TEAD2 binding motifs. We replaced the TEAD2 motifs with *XhoI* and *HindIII* restriction sites by PCR and *DpnI* digestion. We cloned this sequence with the restriction sites and the 5′ and 3′ homology arms from the genome into the pEASY-Blunt vector as a donor construct. We designed sgRNAs with an online website tool (http://benchling.com), then synthesized, annealed and cloned the sgRNA primers into the pX459 vector. We transfected these

vectors into 2iL-ESCs for genomic editing with FuGENE® 6 transfection reagent (Promega) as described above. Then we selected positive clones with 2 μg/mL puromycin for 5 days. To verify corrected clones, we used the cloned genome as a template, PCR with primers F and R were performed to obtain the 704 bp sequence containing *Xho I* and *Hind III* sites. The PCR products of the clones with homozygous mutations were digested with *Xho I* enzyme to yield bands of 155 bp and 549 bp, and with *Hind III* enzyme to yield bands of 292 bp and 412 bp. Then the clones were verified by Sanger sequencing. The primer and sequences of sgRNA used are listed in Table EV1.

## Genomic DNA isolation and DNA methylation analysis

Genomic DNA (gDNA) was extracted with TIANamp Genomic DNA Kit (TIANGEN). DNA methylation analysis was performed as previously described (Graf et al, 2017). Briefly, two micrograms of gDNA were digested overnight at 37 °C with 10 U *HpaII* (New England Biolabs) or *McrBC* (New England Biolabs) in 20 μL total reaction volume. Digested gDNA was loaded on a 0.8% agarose gel and gDNA fragments were separated by electrophoresis. Quantification of DNA signal was measured using ImageJ software.

## siRNAs transfection

siRNAs were designed and synthesized by Shanghai GenePharma (GenePharma, China). Three siRNAs targeting on *Tead2*, *Tead4*, *Esrrb*, *Nr5a2*, and *Tcfcp2l1* genes were designed and synthesized, the most effective siRNA identified by qPCR was used for further experiments. 24 h prior to transfection, cells were plated onto a 6-well plate at 40–60% confluence. Transfection was performed with Lipofectamine 2000 (Invitrogen, USA) according to the manufacturer's protocol. 7.5 μL Lipofectamine 2000 reagent and siRNAs were diluted in Opti-MEM (Gibco) and incubated at room temperature (RT) for 10 min. Then the mixtures were added to cells, and the final concentration of siRNAs was 50 nM. The medium was replaced 6–8 h after transfection with fresh culture medium. The sequences of siRNAs are listed in Table EV4.

## AP staining

AP staining was performed with BCIP/NBT Alkaline Phosphatase Color Development Kit (Beyotime, C3206) and Alkaline Phosphatase Staining Kit II (Stemgent, 00-0055) by following the manufacturer's instructions.

## Quantitative RT–PCR

Total RNAs were extracted with TRIzol® reagent. For quantitative PCR, cDNAs were synthesized with HiScript® III RT SuperMix for

qPCR (+gDNA wiper) (Vazyme Biotech). Real-time PCR was performed using SYBR Green mix (Genstar) in a CFX96 real-time PCR system (Bio-Rad) according to the manufacturer's instruction. mRNA levels were normalized to *Gapdh* level. The primers used in RT-qPCR assays are listed in Table EV5.

## Immunoprecipitation and Western blotting

Cells were resuspended in RIPA buffer (1% Triton X-100, 0.1% sodium dodecyl sulfate (SDS), 150 mM KCl, 50 mM Tris-HCl (pH 7.4),1 mM EDTA, 1% sodium deoxycholate), 1 mM PMSF, and 1 × protease inhibitor cocktails. Total soluble proteins were obtained by centrifugation at 15,294 × *g* for 10 min. Samples were separated on SDS-PAGE gel and transferred onto a polyvinyl difluoride (PVDF) membrane (Millipore). The PVDF membrane was blocked with 5% milk in TBS-T buffer. Immunoblot analysis was performed with the indicated antibodies. Then the membrane was washed with TBS-T buffer and immunoblotted.

For immunoprecipitation, cells were lysed in lysis buffer (20 mM Tris-HCl (pH 7.5), 0.1% Triton X-100, 300 mM KCl, and 5 mM EDTA (pH 8.0)) supplemented with 1 × protease inhibitor cocktails. 2 μg of indicated antibodies and the protein extract were incubated with M2 magnetic beads (Sigma-Aldrich) at 4 °C for 16 h, and then immunocomplexes were washed four times with wash buffer (20 mM Tris-HCl (pH 7.5), 0.1% Triton X-100, 500 mM KCl, and 5 mM EDTA (pH 8.0)), resolved on SDS-PAGE gel, and analyzed using immunoblotting.

The antibodies used in these studies included anti-Oct-3/4 antibody (C-10) (Santa Cruz Biotechnology, sc-5279), anti-β-Actin antibody (Sigma-Aldrich, A2228), anti-Sox2 antibody (Abcam, ab79351), anti-Biotin antibody (Cell Signaling Technology, 7075), anti-FLAG antibody (Sigma-Aldrich, F1804), anti-HA antibody (Abcam, ab9110), anti-V5 Tag antibody (Thermo Fisher Scientific, R960-25), anti-TEF-4 (Santa Cruz Biotechnology, sc-81397).

## RNA-seq and bioinformatics analysis

RNA-seq libraries were constructed using the VAHTS mRNA-seq V3 Library Prep Kit (Vazyme Biotech). The libraries were denatured and diluted at a proper concentration, then were sequenced on Illumina NovaSeq (Annoroad Gene Technology Co., Ltd).

Raw reads were qualified with FastQC tool and trimmed with trim_galore if reads contained adapters. Processed reads were mapped to Ensembl transcriptome version 95 (mm10) by using RSEM (Li and Dewey, 2011) with parameters "-star". Raw tag counts were extracted from sample.gene.results files and merged together, and then GC-normalized using EDASeq (Risso et al, 2011). The DESeq2 package was used to analyze DEGs (Love et al, 2014). A DEG was defined as a gene with *q*-value < 0.05 and fold change >1.5. The top Gene Ontology (GO) processes were enriched by Metascape web-based platform (Zhou et al, 2019). Other analysis was performed using glbase (Hutchins et al, 2014).

## ATAC-seq and bioinformatics analysis

ATAC-seq experiments were performed as previously described (Buenrostro et al, 2015b). Briefly, 50,000 cells were harvested and washed once with 50 μL cold PBS, and were resuspended in 50 μL

of lysis buffers (10 mM Tris-HCl (pH 7.4), 10 mM NaCl, 3 mM MgCl₂, 0.2% (v/v) IGEPAL CA-630). The nuclei suspension was centrifuged 500 × *g* at 4 °C for 10 min. The pellet was resuspended in 50 μL of transposition reaction mix (10 μL TD buffer, 5 μL Tn5 transposase, and 35 μL nuclease-free H₂O), and incubated at 37 °C for 30 min. DNA was isolated using a MinElute PCR Purification Kit (QIAGEN). ATAC-seq libraries were constructed and purified with AMPure XP beads (Beckman Coulter), and then denatured, diluted, and sequenced on the HiSeq X-Ten platform (Annoroad Gene Technology Co., Ltd). After trimming the adapter sequence with Cutadapt, ATAC-seq reads were aligned to mm10 genome using Bowtie2 with default parameters. Reads mapping to mitochondrial DNA or unassigned sequences were discarded. Finally, concordantly aligned pairs were retained. The BAM files of biological replicates were merged, and peaks were called by using dfilter (Kumar et al, 2013) with the settings of "-bs=100 -ks=50 -pe -lpval=2". Alignment BAM files were converted into read coverage files (bigWig format) by using deepTools (Ramírez et al, 2016). Motif analysis was performed using HOMER (v.4.10) (Heinz et al, 2010) with the settings of "--size given".

## ChIP-seq and bioinformatics analysis

ChIP experiments were performed as previously described (Li et al, 2017b). Briefly, 1 × 10⁷ cells were cross-linked with 1% formaldehyde at RT for 10 min. The reaction was stopped by adding glycine (final concentration of 0.125 M). Cross-linked cells were lysed in ChIP SDS lysis buffer (1% SDS, 10 mM EDTA, 50 mM Tris-HCl (pH 8.0)) containing 1 × protease inhibitor cocktail and PMSF, and then sonicated to achieve a DNA size of 200–400 bp. After sonication, the supernatant was diluted with IP buffer and then co-incubated with antibody–Dynabeads protein A/G (1:1 mixed) at 4 °C overnight with rotation. Antibodies used were anti-YY1 antibody (Abcam, ab109237) and anti-SMC1 antibody (Bethyl Laboratories, A300-055A).

ChIP for Biotin. Biotin ChIP was performed as previously described (Li et al, 2019). Cells stably expressed biotin-alone or biotin-TEAD2 were expanded and cross-linked with 1% formaldehyde. Cross-linked cells were sonicated and diluted tenfold with ChIP dilution buffer, and then incubated with Dynabeads M-280 streptavidin at 4 °C overnight. Streptavidin dynabeads-bound DNA was subsequently washed twice with wash buffer 1 (2% SDS), once with wash buffer 2 (50 mM HEPES (pH 7.5), 1 mM EDTA, 500 mM NaCl, 0.1% sodium deoxycholate, 1% Triton X-100), once with wash buffer 3 (10 mM Tris-HCl (pH 8.0), 1 mM EDTA, 250 mM LiCl, 0.5% NP-40, 0.5% sodium deoxycholate) and then twice with TE wash buffer (10 mM Tris-HCl (pH 8.0), 1 mM EDTA). ChIPed DNA was reverse-cross-linked and purified for DNA library construction followed by sequencing.

ChIP-seq libraries were constructed using the VAHTSTM Universal DNA Library Prep Kit for Illumina® V2 (Vazyme Biotech). After PCR library amplification, size selection of adapter-ligated DNA was performed using Agencourt AMPure XP Beads (Beckman Coulter). The libraries were sequenced on the Illumina NovaSeq or DNBSEQ-T7 platform (Geneplus-Beijing Institute (Beijing, China)). After trimming the adapter sequence with Cutadapt, ChIP-seq reads were aligned to mm10 genome using Bowtie2 with the default parameters. Reads mapping to mitochondrial DNA or unassigned sequences were discarded.

Finally, concordantly aligned pairs were retained. The peaks were called using MACS2 (Zhang et al, 2008) with the default parameters. Alignment BAM files were converted into read coverage files (bigWig format) using deepTools (Ramírez et al, 2016). Peak overlap was done by using bedtools (v2.25.0). Differential peaks were called by MACS2 bdgdiff function with the default parameters. Motif analysis of ChIP-seq peaks was performed by using Homer.

## BL-Hi-C

The BL-Hi-C experiments were performed as previous described (Dong et al, 2022; Liang et al, 2017). Briefly, cells were treated with 1% formaldehyde at RT for 10 min followed by quenching with 0.2 M glycine. Nuclei were isolated for subsequent experiments, including digestion with *HaeIII* (New England Biolabs), end-plus-A treatment, proximity ligation with biotin-labeled Bridge Linker, DNA purification and enrichment of biotin-labeled DNA with Dynabeads M-280 streptavidin. The enriched bead-bound DNA was subjected to end repair, adapter ligation, PCR amplification and DNA library construction, followed by sequencing on the Illumina NovaSeq platform (Annoroad Gene Technology Co., Ltd.).

## BL-Hi-C data analysis

The BL-Hi-C data were processed with ChIA-PET2 v0.9.2 software (Li et al, 2017a) with the parameter "-A ACGCGATATCTTATC -B AGTCAGATAAGATAT -s 1 -m 1 -t 4 -k 2 -e 1 -l 15 -S 500" to identify chromatin interactions that annotated with the genome mm10. The biological replicates in each group were merged to perform the A/B compartments and TAD analysis. The interaction matrix was generated by HiC-Pro (Servant et al, 2015). Normalization was performed by HiCExplorer (Wolff et al, 2018) hicCorrectMatrix (--correctionMethod KR). TADs and TAD boundaries were defined by HiCExplorer hicFindTADs (--correctForMultipleTesting fdr --thresholdComparisons 0.05 --minDepth 120000 --maxDepth 200000 --step 40000) at 40-kb resolution. Compartments were analyzed using HOMER (v.4.10) tools(Heinz et al, 2010) with the default parameters at 100 kb resolution. APA analysis was performed using juicer tools (Durand et al, 2016). The producibility of BL-Hi-C datasets was calculated on pairs of raw Hi-C contact matrices at 100 kb resolution by using HiCRep (Yang et al, 2017).

High confidence interactions were defined as those with an FDR < 0.05 for downstream analysis. Differential EP loops mediated by TEAD2 were identified using the diffloop (Lareau and Aryee, 2018) pipeline, with quickAssoc function, which was based on an overdispersed Poisson regression model. Differential loops with high confidence were chosen by applying the following criteria: $p$-value < 0.05 and $|\log_2 FC| > 1$. The data was further visualized in the WashU Epigenome Browser (Zhou et al, 2011).

## QHR-4C

QHR-4C experiments were performed as previously described (Jia et al, 2020). Briefly, digested suspensions of $1 \times 10^5$–$1 \times 10^6$ cells from tissues were cross-linked with 2% formaldehyde for 10 min and the reaction was terminated by 0.2 M glycine. The pellet was permeabilized and digested with *DpnII* overnight followed by proximity ligation. Then, DNA under 1000 bp was extracted and sonicated. To enrich ligation events associated with a specific viewpoint, an appropriate amount of sonicated DNA was taken as a template to linearly amplify for 100 cycles using a 5′ biotin-labeled probe of the viewpoint of interest. The amplification products were incubated at 95 °C for 5 min, immediately cooled on ice to obtain amplified ssDNA and then enriched with Dynabeads M-280 streptavidin. The bead-bound ssDNA was then ligated with adapters. Finally, QHR-4C libraries were constructed with specific primer pairs (forward primers containing Illumina P5 with sequences near a specific viewpoint and reverse primers containing Illumina P7 with an index and sequences matching the adapter) and then sequenced on the Illumina NovaSeq platform (Annoroad Gene Technology Co., Ltd.). The primers for QHR-4C used in this study are listed in Table EV6.

## QHR-4C data analysis

Adapter sequences in raw paired-end reads were removed with Trim Galore and then subjected to cutadapt (version 3.4) (Kechin et al, 2017) to trim the primer sequence at the 5′ end of read 1. Reads that did not contain primer sequences were discarded. Reads were mapped to the mm10 genome using bowtie2 with the following parameters: --very-sensitive --end-to-end --no-unal -X 2000. Bam files were imported into the r3Cseq package (version 1.38.0) (Thongjuea et al, 2013). Normalized bedgraph files were thus generated and then transformed into bigwig files using the bedGraphToBigWig tool.

## CUT&Tag and bioinformatics analysis

The CUT&Tag experiments were performed as previously described (Kaya-Okur et al, 2019). In brief, 100,000 cells were harvested and washed twice with 200 μL of wash buffer. 10 μL concanavalin A beads (Bangs Laboratories) were added per sample and incubated at room temperature for 15 min. Then, 100 μL of dig-wash buffer containing 2 mM EDTA and 1 μg of primary antibody were added. The primary antibody incubation was performed on a rotating platform at 4 °C overnight. Two hundred microliters of dig-wash buffer were added to remove the unbound antibodies. Then, the reaction was incubated with pAG-Tn5 (homemade) at 4 °C for 2 h. Two hundred microliters of dig-med buffer were added to remove unbound pAG-Tn5 protein. Next, the cells were resuspended in 100 μL of tagmentation buffer and incubated at 37 °C for 1 h. To stop tagmentation, 2.25 μL of 0.5 M EDTA, 2.75 μL of 10% SDS and 0.5 μL of 20 mg/mL proteinase K were added and incubated at 55 °C for 30 min and then at 70 °C for 30 min to inactivate proteinase K. Then, DNA was extracted. The antibodies included H3K27ac (Active Motif, 39133), H3K4me1 (Active Motif, 39297).

To generate the sequencing libraries, 21 μL DNA were mixed with 2 μL of a universal i5 and a uniquely barcoded i7 primer (Buenrostro et al, 2015a) using a different barcode per sample. A volume of 25 μL of NEBNext high-fidelity 2 × PCR master mix (New England Biolabs) was added and mixed. The sample was placed in a thermocycler with a heated lid using the following cycling conditions: 72 °C for 5 min (gap filling), 98 °C for 30 s, 14 cycles of 98 °C for 10 s and 63 °C for 30 s, final extension at 72 °C

for 1 min, and then hold at 8 °C. Post-PCR clean-up was performed by adding 1 × volume of Ampure XP beads (Beckman Counter), and the libraries were incubated with beads at room temperature for 15 min, washed twice gently with 80% ethanol, and eluted in 25 μL of 10 mM Tris (pH 8.0). The libraries were sequenced on Illumina NovaSeq (Annoroad Gene Technology Co., Ltd.). After trimming the adapter sequence with Cutadapt (v.0.6.1) (Martin, 2011), the CUT&Tag reads were aligned to the mm10 genome by using Bowtie2 (v.2.4.1) (Langdon, 2015) with the default parameters. Low-mapping-quality reads were filtered by SAMtools (v.1.9), and duplicate reads were removed by Picard tools (v1.90). MACS2 (Zhang et al, 2008) was used to call peaks with $q$-value < 0.01. BigWig files were generated by deepTools (Ramirez et al, 2014) by the RPKM normalization method and visualized in the WashU Epigenome Browser (Zhou et al, 2011).

## Quantification and statistical analysis

Data are presented as mean values ± SD unless otherwise indicated in the figure legend. Sample numbers and experimental repeats are indicated in figure legends. All statistical analyses were done in R. Statistical significance was determined by Student's *t*-test analysis (two-tailed) for two groups. Differences in means were considered statistically significant at $p < 0.05$.

## Data availability

The NCBI GEO number of the RNA-seq, ATAC-seq, ChIP-seq, CUT&Tag, BL-Hi-C, and QHR-4C data described in this paper is GSE226316 and the accession number of the Genome Sequence Archive of the Beijing Institute of Genomics (BIG) Data Center is CRA009963. Published ChIP-seq datasets GSE72164, GSE23943, GSE92407, GSE157748 were used in this study.

## Peer review information

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

## Acknowledgements

This work was supported by the National Key R&D Program of China [2021YFA1100300]; National Natural Science Foundation of China [31925009, U21A20195, 32300466, 32100463]; Major Project of Guangzhou National Laboratory [GZNL2023A02010, GZNL2023A02008]; Guangdong Basic and Applied Basic Research Foundation [2022A1515111085, 2023A1515010730]; China Postdoctoral Science Foundation [2022M723167]; Guangzhou Key R&D Program [2023B03J1230]; Science and Technology Projects in Guangzhou [202201010117, 202201010157, 202201010755].

## Author contributions

**Rong Guo**: Conceptualization; Validation; Investigation; Visualization; Writing—original draft; Writing—review and editing. **Xiaotao Dong**: Data curation; Formal analysis; Visualization; Writing—review and editing. **Feng Chen**: Validation. **Tianrong Ji**: Validation. **Qiannan He**: Validation. **Jie Zhang**: Validation. **Yingliang Sheng**: Data curation. **Yanjiang Liu**: Validation. **Shengxiong Yang**: Validation. **Weifang Liang**: Validation. **Yawei Song**: Supervision; Writing—review and editing. **Ke Fang**: Validation; Writing—review and editing. **Lingling Zhang**: Supervision; Funding acquisition; Project administration; Writing—review and editing. **Gongcheng Hu**: Supervision; Writing—review and editing. **Hongjie Yao**: Supervision; Funding acquisition; Validation; Project administration; Writing—review and editing.

## Disclosure and competing interests statement

The authors declare no competing interests.

# Expanded View Figures

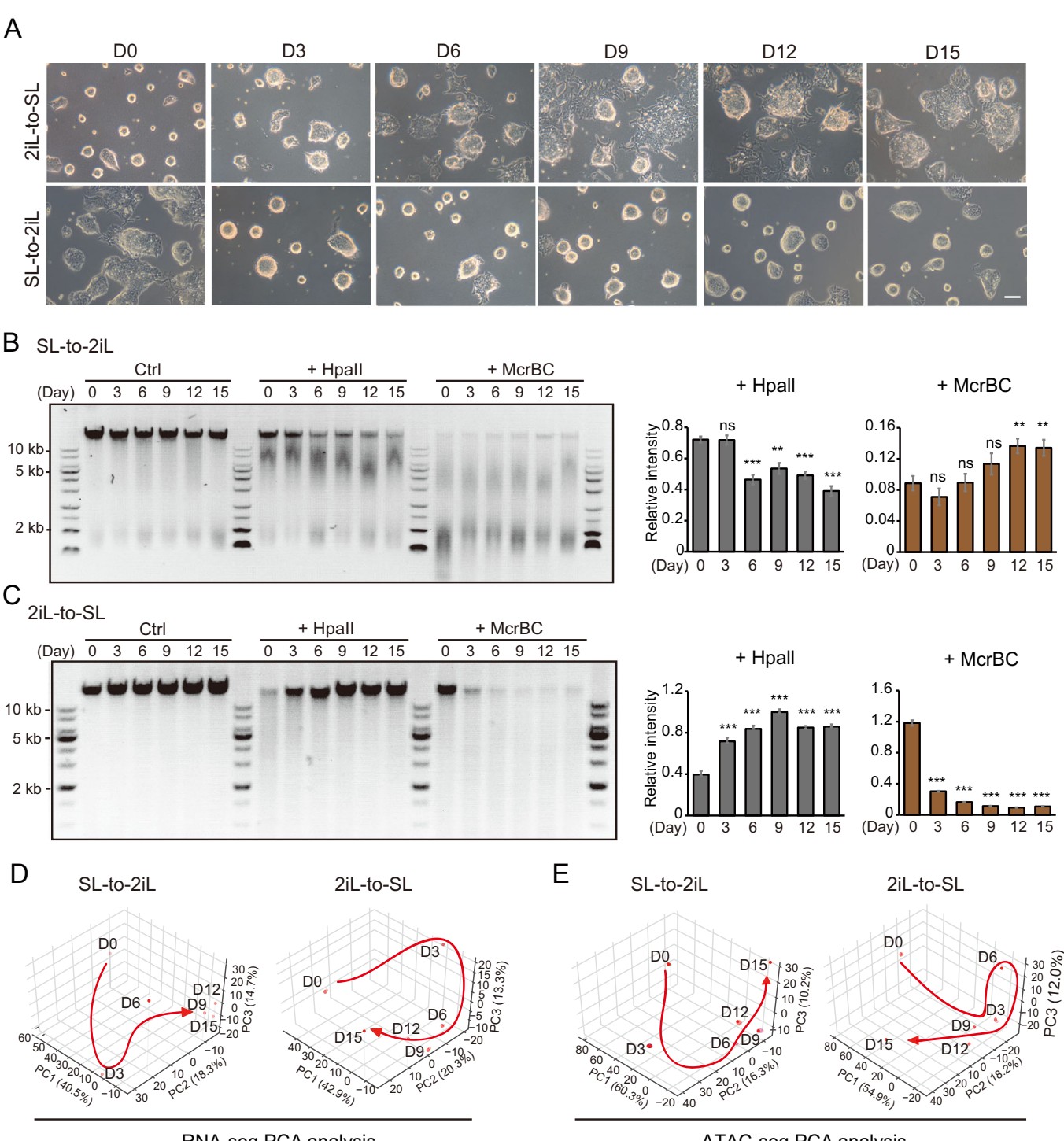

**Figure EV1. Dynamic conversion between 2iL-ESCs and SL-ESCs.**

(**A**) Dynamic changes of mESC morphology during the 2iL-to-SL transition from day 0 to day 15 or during the SL-to-2iL transition from day 0 to day 15. Scale bar, 100 μm.
(**B, C**) CpG methylation levels of genomic DNA measured by digestion with the methylation-sensitive restriction enzymes *HpaII* and *McrBC* at different time points during the SL-to-2iL (**B**) and 2iL-to-SL (**C**) transitions. Quantification of DNA signals of the upper band of the agarose gel using Fiji image analysis software. Data is presented as the mean ± SD. Indicated significances are tested using Student's *t*-test analyses (**$p < 0.01$, ***$p < 0.001$). $n = 3$ replications. M, DNA ladder. (**D, E**) PCA of RNA-seq data (**D**) and ATAC-seq data (**E**) during the transition between 2iL-ESCs and SL-ESCs at different time points.

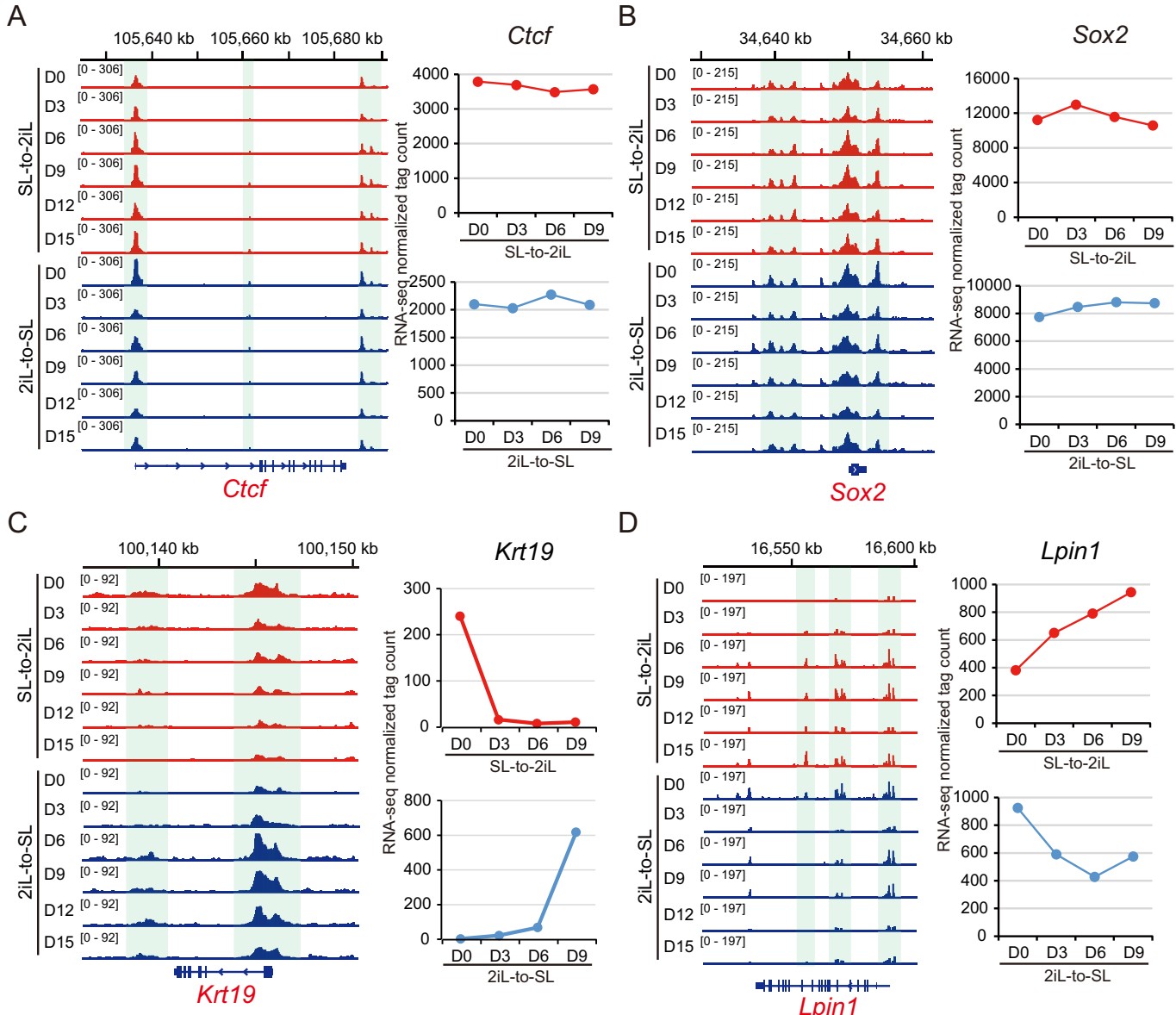

**Figure EV2. Chromatin accessibility for the loci with PO, OC, and CO peaks, and expression levels of the respective genes.**

(A, B) Representative loci of *Ctcf* and *Sox2* with PO peaks during the transition between SL-ESCs and 2iL-ESCs, respectively (left). Expression values of *Ctcf* and *Sox2* from RNA-seq data (right). (C, D) Representative locus of *Krt19* and *Lpin1* with Region 1/4 (C) and Region 2/3 (D) defined by ATAC-seq during the transition between SL-ESCs and 2iL-ESCs, respectively (left). Expression values of *Krt19* and *Lpin1* from RNA-seq data (right).

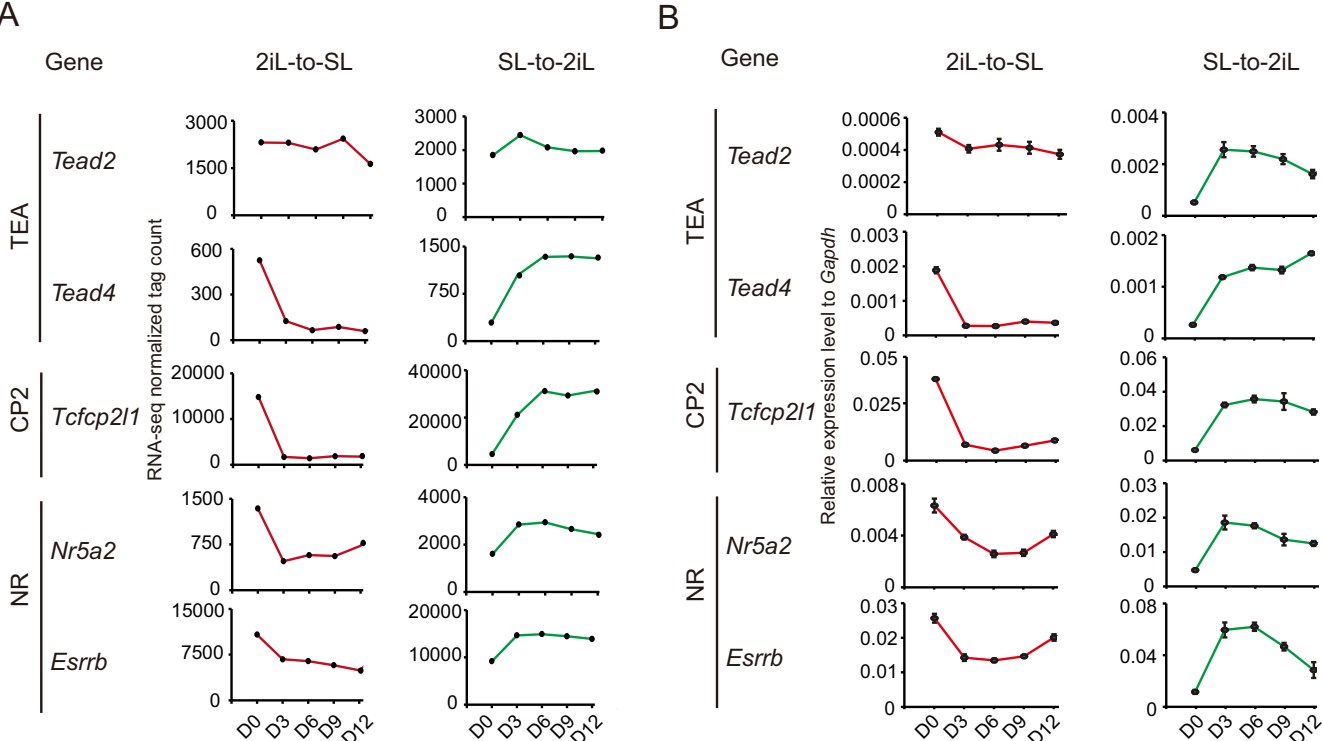

**Figure EV3. Expression levels of candidate factors during the conversion between 2iL-ESCs and SL-ESCs.**

(A) Expression values of the candidate genes from RNA-seq data. (B) RT-qPCR analyzing the expression levels of candidate genes from the TEA, CP2, and NR families. These are enriched in Region 2/3 during the transition between 2iL-ESCs and SL-ESCs. Data are presented as the mean ± SD. *n* = 3 biological replicates.

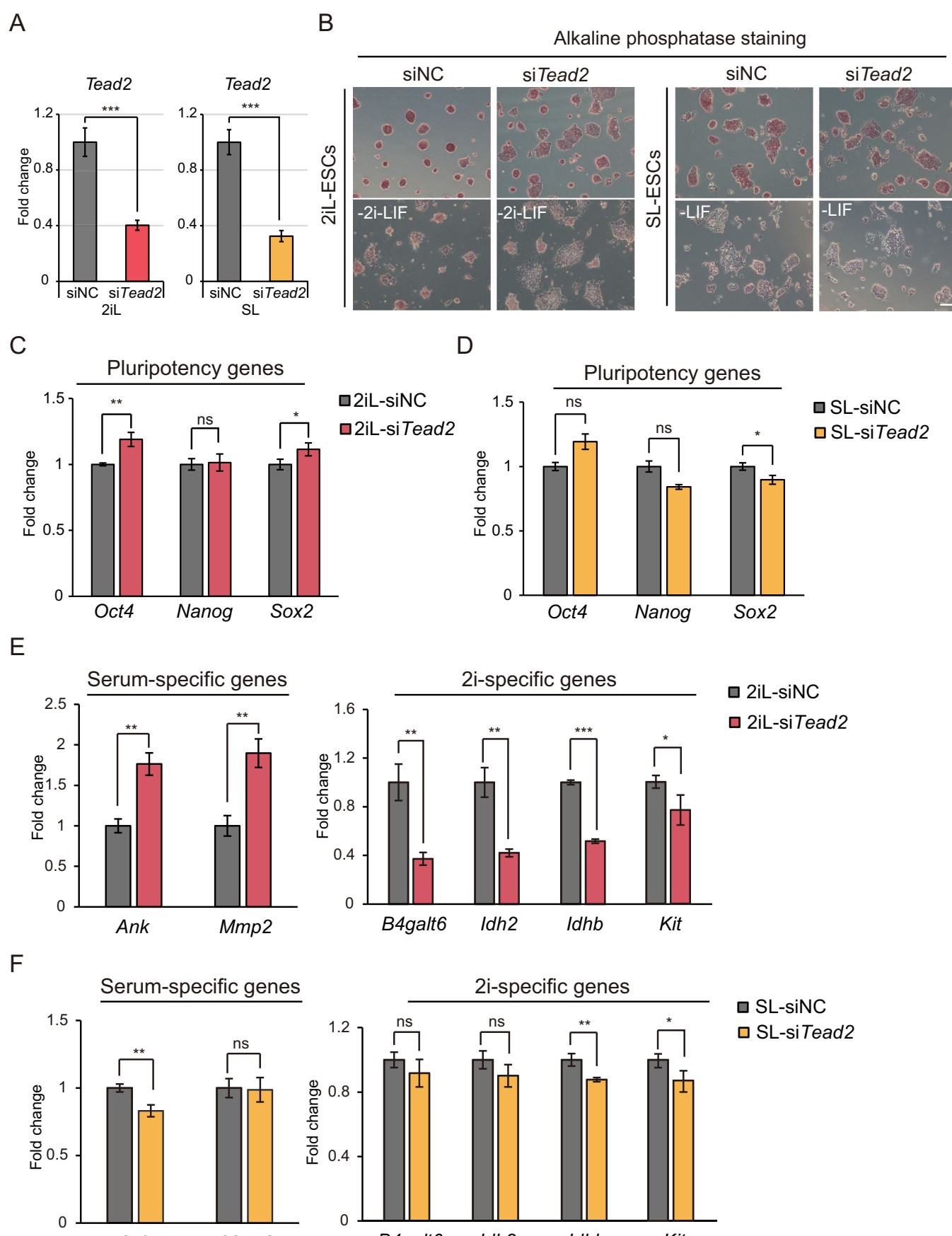

◄ **Figure EV4. Effect of TEAD2 on the ground-state pluripotency in 2iL-ESCs.**

(A) RT-qPCR determining the *Tead2* knockdown efficiency in 2iL-ESCs and SL-ESCs. Data are presented as the mean ± SD. Indicated significances are testing using Student's *t*-test analyses (***$p < 0.001$). $n = 3$ biological replicates. (B) Representative images of AP staining of 2iL- and SL-ESCs and cells after 3 days of differentiation in complete medium containing 10% serum or in the absence of LIF. Cells were treated with control siRNA or siRNA targeting *Tead2*. Scale bar, 100 µm. (C, D) RT-qPCR testing the expression of pluripotent genes in 2iL-ESCs (C) and SL-ESCs (D). Data are presented as the mean ± SD. Indicated significances are tested using Student's *t*-test analyses (*$p < 0.05$, **$p < 0.01$). $n = 3$ biological replicates. (E, F) RT-qPCR testing the expression of 2i- and serum-specific genes in 2iL-ESCs (E) and SL-ESCs (F). Data are presented as the mean ± SD. Indicated significances are tested using Student's *t*-test analyses (*$p < 0.05$, **$p < 0.01$, ***$p < 0.001$). $n = 3$ biological replicates.

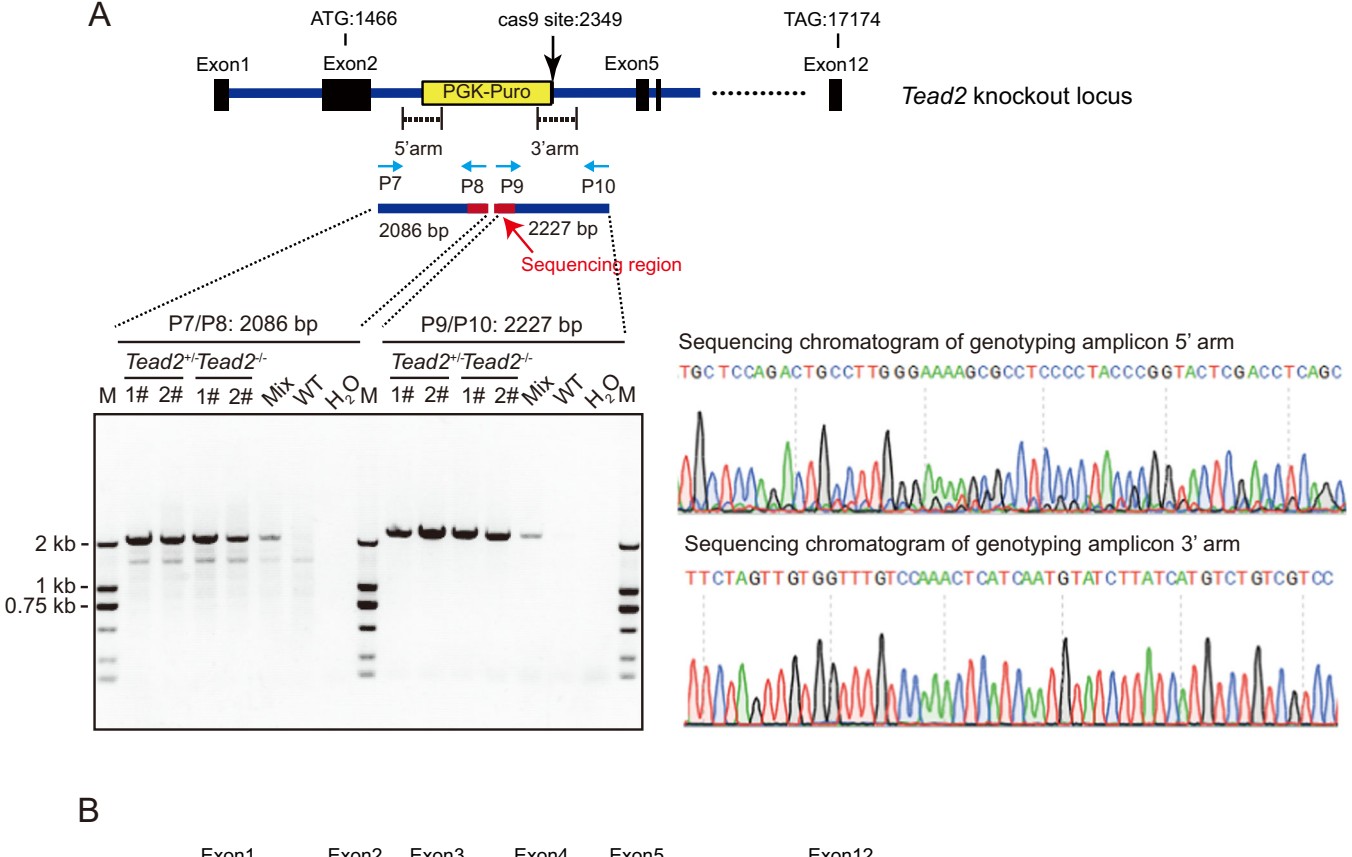

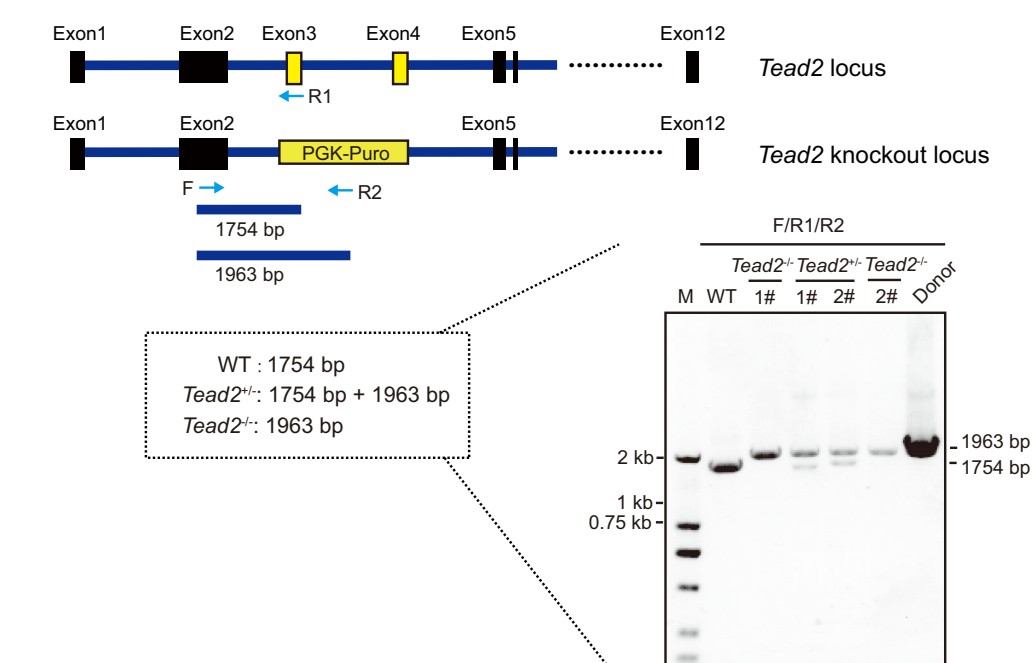

**Figure EV5.   PCR verification of *Tead2*-knockout in SL-ESCs.**

(A) Genomic PCR verification of corrected clones. Primers were designed on both sides of the homologous arm to ensure that the sequence was inserted in the right position. The size of the 5'-arm terminal PCR product is 2086 bp, and the size of the 3'-arm terminal PCR product is 2227 bp. (B) Genomic PCR analysis identifying homozygous or heterozygous clones of *Tead2*-knockout SL-ESCs.

