## [Peer Review File · The EMBO Journal]

TEAD2 initiates ground-state pluripotency by mediating chromatin looping

Rong Guo, Xiaotao Dong, Feng Chen, Tianrong Ji, Qiannan He, Jie Zhang, Yingliang Sheng, Yanjiang Liu, Shengxiong Yang, Weifang Liang, Yawei Song, Ke Fang, Lingling Zhang, Gongcheng Hu, and Hongjie Yao

Corresponding author(s): Hongjie Yao (yao_hongjie@gzlab.ac.cn)

Review Timeline:

Submission Date:	25th Jun 23
Editorial Decision:	28th Jul 23
Revision Received:	23rd Jan 24
Editorial Decision:	19th Feb 24
Revision Received:	26th Feb 24
Accepted:	3rd Mar 24

Editor: Daniel Klimmeck

Transaction Report:

Dear Dr Yao,

Thank you for submitting your manuscript for consideration by the EMBO Journal. It has now been seen by three referees whose comments are shown below.

Should you be able to address these criticisms - detailing gaps in the choice of appropriate cellular models, robustness of the biochemical data presented and completeness of the senescence characterisation - in full, we could consider a revised manuscript. I should remind you that it is EMBO Journal policy to allow a single round of revision only and that, therefore, acceptance or rejection of the manuscript will depend on the completeness of your responses in this revised version. I do realize that addressing all the referees' criticisms will require a lot of additional time and effort and be technically challenging.

If you decide to thoroughly revise the manuscript for the EMBO Journal, please include a detailed point-by-point response to the referees' comments. Please bear in mind that this will form part of the Review Process File, and will therefore be available online to the community. For more details on our Transparent Editorial Process, please visit our website: <https://www.embo.org/embo-press>

Thank you for the opportunity to consider your work for publication. I look forward to your revision.

Kind regards,

Daniel Klimmeck

Daniel Klimmeck, PhD
Senior Editor
The EMBO Journal

Instruction for the preparation of your revised manuscript:

- 1) a .docx formatted version of the manuscript text (including legends for main figures, EV figures and tables). Please make sure that the changes are highlighted to be clearly visible.
 - 2) individual production quality figure files as .eps, .tif, .jpg (one file per figure).
 - 3) a .docx formatted letter INCLUDING the reviewers' reports and your detailed point-by-point response to their comments. As part of the EMBO Press transparent editorial process, the point-by-point response is part of the Review Process File (RPF), which will be published alongside your paper.
 - 4) a complete author checklist, which you can download from our author guidelines ([https://wol-prod-cdn.literatumonline.com/pb-assets/embo-site/Author Checklist%20-%20EMBO%20J-1561436015657.xlsx](https://wol-prod-cdn.literatumonline.com/pb-assets/embo-site/Author%20Checklist%20-%20EMBO%20J-1561436015657.xlsx)). Please insert information in the checklist that is also reflected in the manuscript. The completed author checklist will also be part of the RPF.
 - 5) Please note that all corresponding authors are required to supply an ORCID ID for their name upon submission of a revised manuscript.
 - 6) It is mandatory to include a 'Data Availability' section after the Materials and Methods. Before submitting your revision, primary datasets produced in this study need to be deposited in an appropriate public database, and the accession numbers and database listed under 'Data Availability'. Please remember to provide a reviewer password if the datasets are not yet public (see <https://www.embopress.org/page/journal/14602075/authorguide#datadeposition>). In case you have no data that requires deposition in a public database, please state so in this section. Note that the Data Availability Section is restricted to new primary data that are part of this study.
- *** Note - All links should resolve to a page where the data can be accessed. ***

7) Our journal encourages inclusion of *data citations in the reference list* to directly cite datasets that were re-used and obtained from public databases. Data citations in the article text are distinct from normal bibliographical citations and should directly link to the database records from which the data can be accessed. In the main text, data citations are formatted as follows: "Data ref: Smith et al, 2001" or "Data ref: NCBI Sequence Read Archive PRJNA342805, 2017". In the Reference list, data citations must be labeled with "[DATASET]". A data reference must provide the database name, accession number/identifiers and a resolvable link to the landing page from which the data can be accessed at the end of the reference. Further instructions are available at .

8) At EMBO Press we ask authors to provide source data for the main and EV figures. Our source data coordinator will contact you to discuss which figure panels we would need source data for and will also provide you with helpful tips on how to upload and organize the files.

Numerical data can be provided as individual .xls or .csv files (including a tab describing the data). For 'blots' or microscopy, uncropped images should be submitted (using a zip archive or a single pdf per main figure if multiple images need to be supplied for one panel). Additional information on source data and instruction on how to label the files are available at .

9) We replaced Supplementary Information with Expanded View (EV) Figures and Tables that are collapsible/expandable online (see examples in <https://www.embopress.org/doi/10.15252/emboj.201695874>). A maximum of 5 EV Figures can be typeset. EV Figures should be cited as 'Figure EV1, Figure EV2' etc. in the text and their respective legends should be included in the main text after the legends of regular figures.

11) For data quantification: please specify the name of the statistical test used to generate error bars and P values, the number (n) of independent experiments (specify technical or biological replicates) underlying each data point and the test used to calculate p-values in each figure legend. The figure legends should contain a basic description of n, P and the test applied. Graphs must include a description of the bars and the error bars (s.d., s.e.m.).

We realize that it is difficult to revise to a specific deadline. In the interest of protecting the conceptual advance provided by the work, we recommend a revision within 3 months (26th Oct 2023). Please discuss the revision progress ahead of this time with the editor if you require more time to complete the revisions. Use the link below to submit your revision:

Referee #1:

In this paper, Guo et al introduce Tead2 as an auxiliary transcription factor (TF) that aids in the transition from serum/LIF (SL) to 2i/LIF (2iL) conditions by facilitating EP interaction at 2iL specific genes. The initial part of the paper is not novel and covers well-established information on the transcriptional and epigenetic profiling during the conversion from SL to 2iL and vice versa, which has been previously reported by Atlasi et al in NCB 2019, Atlasi et al in Nat Com 2020, Marks et al in Cell 2012, and Joshi et al Cell Stem Cell 2015. The authors observe a similar significant gradual change in the transcriptional and epigenetic landscape, consistent with the previous findings.

However, the more interesting aspect of the manuscript lies in the second part. Here, the authors delve into the role of Tead2 in the SL-to-2iL transition, demonstrating that its deletion can influence cell morphology, transcriptional profiles, and chromatin interactions during this transition. They identify Tead2 binding to enhancer and promoter regions that are co-occupied by YY1 and Cohesin (SMC1), and show that Tead2 mediates enhancer-promoter (E-P) interactions. Moreover, the deletion of Tead2 leads to a reduction in a specific set of E-P interactions without significantly affecting YY1 and SMC1 binding, indicating that Tead2 mediates chromatin interactions independently of YY1-Cohesin.

Although the authors have performed extensive series of experiments, the paper lacks a comprehensive exploration of these interesting phenotypes. It appears somewhat disjointed and would benefit from further refinement to create a cohesive and compelling narrative. Specifically, I suggest that the authors focus on investigating the Tead2 knockout ESC phenotype in greater detail and provide explanations for the mechanisms underlying the impact of Tead2 deletion on the 2iL transition. The paper also contains many mistakes in figure legends.

Major comments:

- 1) Figure-1 legend is missing.
- 2) Motif analysis method is not clear. I suspect that the authors have used the mm10 genome as a background for motif analysis of differential ATAC-seq peaks resulting in such a high enrichment of CTCF sites. The authors need to use total ATAC-seq peaks as background to avoid this issue.
- 3) The effects of siRNA on formation of 2iL is based on cell morphology and the experiment lacks quantitative data. The authors should add quantitative experiments such as AP staining to validate their findings. Similar for Fig 3B.
- 4) Fig 2C: Neither Tead2 expression is strongly different between SL and 2iL ESCs, nor the authors compared Tead2 ChIP-seq pattern in SL and 2iL ESCs. At this stage, it's not clear why and how Tead2 plays a 2iL-specific role.
- 5) The authors performed most of their experiments at day6 of SL to 2iL transition. The authors need to show the long term effects of Tead2 deletion in fully adapted 2iL ESCs (> d20). For example, what is the phenotype of Tead2 ESCs cultured for long term in 2iL in terms of AP-staining, transcriptional profiles and differentiation capacity (exit from pluripotency e.g in N2B27 medium)? The same is true for chromatin interactions: do Tead2 KO ESCs show similar or different chromatin interactions in long term 2iL or are these interactions are buffered out over time indicating that Tead2 is not sufficient to mediate these interactions.
- 6) It has been concluded that TEAD2 plays an important role in SL to 2iL transition based on siRNA knockdown/ knockout experiments. A gain of function expression (overexpression of TEAD2) is needed to further support whether TEAD2 enhance SL-to-2iL conversion.
- 7) The authors suggest that some E-P interactions are dependent Tead2. The authors need to show whether these interactions normally change during SL-to-2iL transition.
- 8) What are the 118 genes that are 2i-specific genes directly bound by Tead2 and are affected by Tead2 KO? How are they involved in 2i-ESCs biology?
- 9) Tead2 and Tead4 have similar motifs. Does Tead4 KO phenocopy Tead2 KO phenotype? Also since the TEA motif is shared with other TEADs, what about Tead1 and 3 expression?
- 10) Fig 4C: how did the authors specified 'enhancers' in this analysis? Are these ATAC-seq or H3K27ac, or simply intergenic regions?
- 11) Fig 4D: The authors should add the previously published ChIP-seq profiles of SL ESCs to establish a link between Tead2 binding and change of histone marks at Tead2 binding sites in 2iL vs SL ESCs.
- 12) Does Tead2 deletion affect epigenetic marking at Tead2 binding sites (proxy to enhancer activity)? The authors should perform H3K27ac/ H3K4me1 ChIP-seq in Tead2 KO vs WT ESCs to address this point.
- 13) What is the epigenetic marking at interactions significantly affected by Tead2 deletion? What makes them different from other E-P interactions?
- 14) Fig 6G: The authors claim that deletion of Tead2 motif at B4galt6 promoter reduces enhancer-promoter interactions. To support this the authors performed 4C-seq in WT and tead2-motif-Mut ESCs. However, this experiment is not informative and is missing many controls. First, the Tead2-KO samples should be included in this experiment as control. Secondly, it's not possible to conclude a reduction in the interaction frequency between WT and MUT cells based on this figure. The authors need to perform differential analysis of interaction between WT and Mut cells and show whether these changes are statistically significant (e.g as shown before). Thirdly, the induced deletion in MUT cells is 64 bp long. How can they be sure that the effect that they are seeing (although subtle) is due to loss of Tead2 binding and no other motifs in this fragment? The authors need to confirm their data by mutating the motif sequence using base substitution.

Minor points:

- 1) Fig EV3: Y-axis represents 'fold change'. What is compared here?

- 2) What is the average distance of E-P interactions mediated by Tead2. The authors should compare this with other E-P , P-P and E-E interactions.
- 3) Please include the % of E-P or % of all interactions that are significantly different between KO vs WT ESCs?
- 4) Authors show that loss of TEAD2 leads to lower SMC1 occupancy at enhancer anchors. It is helpful to show the expression level of SMC1 in the KO cells.
- 5) Fig 3B legend says YY1 mediated interactions whereas the figure shows Tead2 dependent interactions.
- 6) FigEV8-F legends does not correspond to the figure.
- 7) Figure6-G and H are swapped and do not correspond to the figure legends.

Referee #2:

In this manuscript, Guo et al. monitor transcription and changes of accessibility over time after switching culture conditions of mouse embryonic stem cells (ESCs) from serum/LIF, where the cells are more prone to differentiation, to 2i/LIF media, representing a stable naïve pluripotent state arguably closer to that of epiblast cells, and vice-versa. They identify genomic regions becoming accessible or losing accessibility during these transitions, and search for transcription factor motifs enriched in the underlying DNA sequences, in an attempt to pinpoint new regulators of the conversion. Among the candidates identified is Tead2. The authors show that knockdown of this factor deteriorates the morphology of ESCs upon transfer to 2i/LIF. Knockout lines are then derived, showing that the loss of Tead2 affects gene expression changes that normally accompany the transition to stable naïve pluripotency. These effects appear due to direct regulation, as determined analysing the genome-wide DNA binding profile of Tead2. The authors go further, determining that Tead2 inactivation compromises the reorganization of some aspects of the 3D genome organization of ESCs subject to media switch. Finally, setting up an elegant genetic experiment, the authors show that the ability of Tead2 to bind to specific genomic regions is required for rewiring spatial organization at these loci, and conclude that the effects of Tead2 in this context seem independent from other regulators of chromatin looping, such as CTCF and YY1. Overall, this paper identifies in Tead2 a regulator of naïve pluripotency.

Major concerns and criticisms:

While, globally, the analysis is technically sound - the experiments are performed and analysed appropriately - the work presented in the manuscript seems incomplete, preventing the reader from fully judging the importance of Tead2 in pluripotent cells. This is put in evidence by the concluding remarks of the authors that read: "In summary, we revealed that TEAD2 is essential for maintaining the ground state pluripotency by regulating the expression of 2i-specific genes ...". Unfortunately, no evidence is provided in the manuscript that allows determining whether Tead2 expression is indeed required to maintain ESCs in 2i/LIF, besides a transient knockdown experiment. This is particularly surprising, given that the authors have generated all the tools required to address this possibility. In short, are Tead2 KO ESCs still able to self-renew in 2i/LIF? The fact that TEAD2 is dispensable during pre-implantation development does not appear to suggest an essential role in pluripotent cells. The relevance of the study depends on this distinction: it is debatable that the serum to 2i transition represents a model of any direct developmental relevance. Moreover, irrespectively of the fact that the gene might be dispensable or essential, it is important that the authors include a rigorous characterization of the effect of the loss of Tead2 in cells stably maintained in 2i/LIF. Do the changes in gene expression and genomic organization reported at day 6 of the conversion eventually normalise? Linked to the previous remark, how do the authors exclude a possible functional complementation by Tead4? Functional redundancy between TEAD factors is well described during development. Also, in light of the ability of Tead2 to interact with coactivators that are crucial to the function of nuclear receptors, it would be important to explore the interplay between Tead2 and this class of transcription factors.

Minor remarks:

The authors refer to the pluripotency state captured by ESCs maintained in serum/LIF as "confused". We feel the use of this term is not appropriate: the term is colloquial and misleading in the context. Please find an alternative. Possibly the authors are referring to the fact that ESCs are "metastable" or "prone to differentiation" in serum/LIF.

In several plots presenting expression dynamics of genes the Y axis is labeled "RNA-seq counts". This unit is unclear. Are the counts in question normalized? (RPKM, RPM.. etc)

Figs 2C and EV3A show expression changes that seem minimal and strongly variable. The authors should adjust accordingly how these results are discussed in the manuscript.

Line 155 The phrasing "the expression of Tead2, Tead4, Tcfcp211, Esrrb and Nr5a2 in region 2/3 was gradually downregulated" is unclear. Are the authors referring to the fact that these genes are located near accessible peaks that are classified as Regions 2/3, and that they are downregulated?

Line 185: "homogeneous". Do the authors mean "homozygous"?

Line 207: It is not clear what is meant by exit of serum specific genes? Possibly downregulation?

Line 231: The phrasing "20.83% (225/1080) of the downregulated genes were 2i-specific genes after Tead2 knockout" is not understandable.

Referee #3:

In this work, Guo et al identified the TF TEAD2 to be important for ESCs culture in 2i. This is an interesting finding in the field of pluripotency. The data are in general technically well performed. However, before publication, there are some points to be better clarified and corrected. In particular, it should be better investigated how TEAD2 binding to chromatin differ between 2i and serum.

Major points

1. The images of Fig. EV1A showed cell morphology changes during 2i-serum transition. However, many, many cells had a differentiation phenotype. Are these cells still all pluripotent? Can the authors show AP staining and compared it to differentiated cells?
2. Fig. EV1B,C, right panels. It should be reported in the corresponding legend what has been quantified in the agarose gel (I guess the upper band).
3. Lane 142-143: "These results suggested that the overlapping accessible peaks between these two processes were essential for the transition between 2i-ESCs and serum-ESCs.". This conclusion is too strong. The authors have only described chromatin accessibility during 2i-serum conversion and vice versa but not a function for the transition.
4. Lanes 154-155. The authors "discovered that the expression of Tead2, Tead4, Tcfcp2l1, Esrrb and Nr5a2 in Region 2/3 was gradually downregulated during 2i-to-serum transition..." The authors should also refer to published data since this analysis has been extensively performed in the past. Are these data consistent with previous work? Moreover, although there is a gradual downregulation of mRNA, the authors should show TEAD2 protein levels by WB. Is still expressed? This would be important to better understand the role of TEAD2 in 2i-ESC regulation (see also my comment on the TEAD2-ChIPseq).
5. Fig. 2D. Lanes 158-169. The authors described that Esrrb KD has less effects on cell morphology. Moreover, they indicated that Nr5a2 has already been reported to be critical for 2i-specific enhancers (Atlasi et al. PMID:31036938). However, the data shown in Atlasi et al. indicated the opposite: a key role for Esrrb whereas Nr5a2 appears to be dispensable. The authors should clarify or correct this point.
6. Fig EV4A-C. Lanes 172-174. "... knockdown Tead2 had little effect on both the expression of core pluripotent factors Oct4, Sox2, Nanog... and the spontaneous differentiation of ESCs". Could the authors explain how did they measure "the spontaneous differentiation of ESCs"? Moreover, the AP staining of Fig. EV4C does not show any difference between ESCs and differentiated cells (-2i-LIF, -LIF in serum).
7. Fig. 3D, Lanes 205-207. "These results indicate that TEAD2 is required for the expression of 2i-specific genes and the exit of serum-specific genes during serum-to-2i transition.". Since many other 2i-genes are not affected, it is more appropriated to conclude that TEAD2 is required for the regulation of "a set" of 2i-specific genes and serum specific genes. Given the known role of ESRRB in serum-2i transition, can the author comment on differences in gene expression/chromatin binding between ESRRB and TEAD2? C2 group was not described.
8. The BIOTIN-TEAD2 ChIP-seq experiment is not clearly described. Was it performed in 2i-ESCs? What is biotin-tag? BirA, BioID? Which was the negative control? Was the biotin-tagged TEAD2 construct overexpressed? It is not clear whether TEAD2 is also expressed in serum-ESCs (see point 4). Is there a difference in TEAD2 binding to chromatin between 2i- and serum-ESCs? It might also help to compare the RNAseq in TEAD2-KD 2i- and serum-ESCs to determine which genes are specifically or commonly regulated by TEAD2 in 2i- and serum-ESCs.
Fig. EV7C (scatter plot of RNAseq between KO Tead2-D6 vs WT-D6) is not cited in the text. Does this experiment belong to this Figure supporting the biotin-ChIP?
9. Fig. EV8: Change of A/B compartments in Tead2-KD ESC+2i. Are domains switching from A to B bound by TEAD2?
10. Fig. 5H. The loss of interactions around B4galt6 gene upon TEAD2-KO is not very clear. It seems more that the interactions are changed instead to be lost. Can the authors show in the image the bed file for TEAD2 called peaks? Are the other anchor points from B4galt6 gene known enhancers?
11. Fig. 6E TEAD2 dimerization. While I consider this result interesting, there is nothing new relative to previous work cite by the authors (Lee et al, 2016). Moreover, there is no information for the silver staining of Fig. EV9 showing the (minimal) loss of TEAD2 dimer in the presence of DTT. The samples were run on an SDS-PAGE, which is not not a native gel. How is it possible to observe TEAD2 dimer in such denaturing conditions? Moreover, the conclusions should be toned down since no data are provided showing that TEAD2 dimerization might mediate EP interactions. Maybe it is better to delete this part from the manuscript since it is out of the scope of the paper and not conclusive.
12. Fig. 6A-C. The results indicated that upon TEAD2-KD in serum-to-2i transition SMC1 occupancy at enhancer is decreased whereas YY1 occupancy is not affected. The authors concluded that "TEAD2 might mediate EP interactions by functioning

together with SMC1 to regulate the expression of 2i-specific genes rather than depending of YY1 and CTCF." I do not see any data that exclude the role of YY1. The lack of decrease in YY1 binding to chromatin upon KD of TEAD2, which is DNA sequence dependent, only shows that YY1 does not depend on TEAD2 and not the contrary.

13. Figure 6. How big was the CRISPR deletion in the promoter region of B4galt6 gene? Does it include the whole promoter? Is this deletion affecting H3K27ac at the contact sites surrounding B4galt6 gene?

14. Discussion, lanes 342-343 "The transition between 2i-ESCs and serum-ESCs can simulate the changes of pluripotency in early embryonic development in vitro". I am not sure that this is the case. Do ESC+serum refer to a specific developmental stage relative to ESC+2i? I think that the comparison between 2i-ESC and serum-ESC only serves to identify factors implicated in ground-state pluripotency.

15. Lanes 377-380. Conclusion sentence. "Therefore, it is of great significance to use chemical small molecules with high selectivity targeting TEAD2 to modulate stem cell fate determination instead of using genes in the future". This sentence is very unclear. Maybe the authors should conclude their work with something more realistic and relevant to the results of their manuscript.

16. The authors referred very often to Atlasi et al. (PMID:31036938). However, other recent works have started to find factors required for ground-state or 2i-ESCs but none of them has been considered.

Minor points

1. Legend of Fig. 1 is missing. Instead, there is the legend of Fig. 6.

2. lane 41. "...ESC+serum....are postulated to represent the confused pluripotency". The term "confused" for pluripotency is very strange. Could the author find another term (advanced, primed, etc.)?

3. Lane 79. "ATAC-seq, BL-Hi-C, RNA-seq and ChIP-seq experiments followed by high throughput sequencing,..". "high throughput sequencing" is redundant since all the listed methods imply this.

4. Fig. EV3. In the images, there is no SD and P values described in the corresponding Figure legend.

5. BL-Hi-C should be introduced with the full name (Bridge Linker-Hi-C)

Point-by-point Response to Reviewers

Referee #1:

In this paper, Guo et al introduce Tead2 as an auxiliary transcription factor (TF) that aids in the transition from serum/LIF (SL) to 2i/LIF (2iL) conditions by facilitating EP interaction at 2iL specific genes. The initial part of the paper is not novel and covers well-established information on the transcriptional and epigenetic profiling during the conversion from SL to 2iL and vice versa, which has been previously reported by Atlasi et al in NCB 2019, Atlasi et al in Nat Com 2020, Marks et al in Cell 2012, and Joshi et al Cell Stem Cell 2015. The authors observe a similar significant gradual change in the transcriptional and epigenetic landscape, consistent with the previous findings.

However, the more interesting aspect of the manuscript lies in the second part. Here, the authors delve into the role of Tead2 in the SL-to-2iL transition, demonstrating that its deletion can influence cell morphology, transcriptional profiles, and chromatin interactions during this transition. They identify Tead2 binding to enhancer and promoter regions that are co-occupied by YY1 and Cohesin (SMC1), and show that Tead2 mediates enhancer-promoter (E-P) interactions. Moreover, the deletion of Tead2 leads to a reduction in a specific set of E-P interactions without significantly affecting YY1 and SMC1 binding, indicating that Tead2 mediates chromatin interactions independently of YY1-Cohesin.

Although the authors have performed extensive series of experiments, the paper lacks a comprehensive exploration of these interesting phenotypes. It appears somewhat disjointed and would benefit from further refinement to create a cohesive and compelling narrative. Specifically, I suggest that the authors focus on investigating the Tead2 knockout ESC phenotype in greater detail and provide explanations for the mechanisms underlying the impact of Tead2 deletion on the 2iL transition. The paper also contains many mistakes in figure legends.

Answer: We have followed the Reviewer's comments, performed more experiments and bioinformatic analysis, and revised this paper in detail.

Major comments:

1) Figure-1 legend is missing.

Answer: We apologize for this mistake. We have added Figure-1 Legend in this new manuscript as below.

Figure 1. Dynamics of chromatin accessibility during the interconversion between SL-ESCs and 2iL-ESCs.

A and B. Chromatin loci arranged into groups according to time and status of being closed or opened, closed to open (CO) or open-to-closed (OC), or permanently open (PO) during the transition from SL-to-2iL (A) and 2iL-to-SL (B). Representative genes are noted for each subgroup on the right.

C. Number of peaks defined in CO/OC and PO for (A and B).

D. Venn diagrams of CO/OC and PO peaks during interconversion between SL-ESCs and 2iL-ESCs.

E. Statistics of the number of genes that were switched at different time points of interconversion between SL-ESCs and 2iL-ESCs on the loci of Region 1/4 and Region 2/3.

F. GO analysis of 481 genes in Region 1/4-CO1/OC1 and CO2/OC2, and 766 genes in Region 2/3-OC1/CO1 and OC2/CO2 in (E).

G and H. Representative loci of *Mmp2* and *B4galt6* within Region 1/4 (G) and Region 2/3 (H) defined by ATAC-seq during the transition between SL-ESCs and 2iL-ESCs, respectively (left). Expression values of *Mmp2* and *B4galt6* from RNA-seq data (right).

2) Motif analysis method is not clear. I suspect that the authors have used the mm10 genome as a background for motif analysis of differential ATAC-seq peaks resulting in such a high enrichment of CTCF sites. The authors need to use total ATAC-seq peaks as background to avoid this issue.

Answer: We have followed the reviewer's suggestion and used total ATAC-seq peaks as the background. However, the results still contain a high enrichment of CTCF sites.

By combining the CTCF ChIP-seq data in both 2iL- and SL-ESCs (GSE92407) (Atlasi *et al*, 2019), we demonstrated that CTCF binding was markedly enriched at the regions harboring CTCF motif, such as PO, Region 1-OC1, Region 2-CO5, Region 4-CO1, Region 3-OC5 (see in Fig. 2A, B), compared to regions lacking CTCF motif (as Region 2-CO1, Region 3-OC1). This finding was consistent with both 2iL- and SL-ESCs (Fig. R1). These results validated the reliability of our motif-analyzing results of differential ATAC-seq peaks.

Figure R1

Figure R1. Tag density pileup for the CTCF ChIP-seq signals at PO, CO1, OC5 and OC1 sites during interconversion between SL-ESCs and 2iL-ESCs.

3) The effects of siRNA on formation of 2iL is based on cell morphology and the experiment lacks quantitative data. The authors should add quantitative experiments such as AP staining to validate their findings. Similar for Fig 3B.

Answer: By following the Reviewer's suggestions, for the experiments of using siRNA to screen key regulatory factors, we have further performed AP staining and cell proliferation assays, and demonstrated that "*knocking down either Tead2 (siTead2) or Nr5a2 (siNr5a2) impeded domed colony formation during SL-to-2iL transition, with Tead2-knockdown exhibiting a more pronounced effect than Nr5a2 knock down (Fig. 2D). Conversely, knockdown of the other three factors (siTead4, siEsrrb, and siTcfp2l1) had minimal effects on cell morphology (Fig. 2D). With the exception of*

siTead4, knockdown of any of the other four factors diminished self-renewal and proliferation capacity to varying degrees, with *siTead2* and *siTcfcp211* also impacting day 0 during the transition (**Fig. 2E**). *ESRRB* and *TCFCP2L1* play crucial roles in stabilizing the regulatory network of naïve pluripotency and preventing the loss of self-renewal (Atlasi et al., 2019; Festuccia et al, 2018a; Festuccia et al, 2018b; Hackett & Surani, 2014; Qiu et al, 2015; Zhang et al, 2021). *NR5A2* can form a regulatory module with *ESRRB* to assist in the binding of core pluripotency factors at most of their occupied regions, thereby regulating the naïve pluripotency network (Festuccia et al, 2021). However, the mechanism by which *TEAD2* regulates ground-state pluripotency remains unknown. Despite displaying a flattened clone-like morphology, *Tead2*-knockdown cells retain their pluripotency, similar to cells with another factor knockdowns (**Fig. 2F**). We speculated that *TEAD2* may not directly influence the core pluripotency but instead regulates the formation of the ground-state pluripotency during *SL-to-2iL* transition". And we have added these new data into our new manuscript at page 7, line 173.

Figure 2.

D. Representative images of cells transfected with siNC (negative control) and siRNAs for *Tead2*, *Tead4*, *Tcfcp211*, *Nr5a2*, and *Esrrb* during the SL-to-2iL process. Scale bar, 100 μ m.

E. Number of cells transfected with siNC (negative control) and siRNAs of *Tead2*, *Tead4*, *Tcfcp211*,

Nr5a2, and *Esrrb*. Cells were grown for 3 days during the SL-to-2iL process. Data are presented as the mean \pm SD of three independent wells.

F. Representative images of AP staining of cells transfected with siNC (negative control) and siRNAs for *Tead2*, *Tead4*, *Tcfcp2l1*, *Nr5a2* and *Esrrb* during SL-to-2iL process. Scale bar, 100 μ m.

4) Fig 2C: Neither *Tead2* expression is strongly different between SL and 2iL ESCs, nor the authors compared *Tead2* ChIP-seq pattern in SL and 2iL ESCs. At this stage, it's not clear why and how *Tead2* plays a 2iL-specific role.

Answer: We have examined the mRNA and protein levels of TEAD2 in both SL and 2iL ESCs, and added these data into the manuscript as “*We measured TEAD2 expression levels in both mRNA and protein in both cell types and observed that TEAD2 expression in 2iL-ESCs was approximately 1.5-fold higher than that in SL-ESCs (Fig. 3A)*” into our new manuscript at page 8, line 193.

Figure 3A. RT-qPCR and Western blot examining TEAD2 expression in both 2iL- and SL-ESCs. mRNA expression was tested in triplicate in three independent experiments. Data are presented as the mean \pm SD. p-values were determined using two-sided Student's t-test. Quantification of protein signal was performed using Fiji image analysis software.

To obtain and compare TEAD2 binding patterns in both 2iL- and SL-ESCs, we have tried several commercial and our self-made anti-TEAD2 antibodies for endogenous TEAD2 ChIP-seq or Cut&Run experiments. However, none of these antibodies worked well due to the high background or low affinity. Therefore, we have generated ESC lines with endogenous biotin-tagged TEAD2, and performed BIOTIN ChIP-seq experiments to identify TEAD2 binding sites in 2iL- and SL-ESCs, respectively. And our data indicated that TEAD2 occupies more binding sites in

2iL-ESCs and binds to active chromatin regions to regulate the expression of 2i-specific genes.

We have added the detailed method and results into the new manuscript as follows:

Generation of Tead2-FLAG-AviTag knock-in 2iL- and SL-ESC lines

“Enzymatic biotinylation with E. coli biotin ligase (BirA) is highly specific in covalently attaching biotin to the 15 amino acid AviTag peptide (Fairhead & Howarth, 2015). To generate in vivo biotinylated-TEAD2 in 2iL- and SL-ESC lines, we express BirA in ESCs. Lentivirus for lenti-birAV5 assembled with psPAX2, pMD2.G vectors in HEK293T cells. And lentiviral supernatants were collected and transfected using the modified polyethylenimine (PEI, Polysciences). 2iL-ESCs and SL-ESCs were infected with lenti-birA-V5 lentivirus and selected with 10 µg/mL of blasticidin for at least 5 days. BirA-V5 overexpression was analyzed by Western blot with Anti-V5 Tag Monoclonal Antibody.

The CRISPR/Cas9 system was used to genetically generate Tead2-FLAG-AviTag knock-in ESC lines. An ATG-FLAG-AviTag was inserted upstream of the start codon of exon 2 of Tead2 gene. The ATG-FLAG-AviTag and the 5' and 3' homology arms amplified from the genome were cloned into the pMD18-T vector as a donor construct. The Tead2-sgRNA target sequence was inserted into the plasmid pX459. Then, 2 µg of pX459-sgRNA, 2 µg of donor vector were co-transfected with 12 µL FuGENE® 6 transfection reagent into 2iL- and SL-ESCs overexpressing BirA-V5 for gene editing following the manufacturer's protocol. Positive clones were selected by 2 µg/mL puromycin for 5 days. Individual clones were picked and re-plated on gelatin-coated 12-well plates for further screening. The selected colonies were verified by genomic PCR and DNA sequencing. Finally, the in vivo biotinylation of TEAD2 was detected with anti-BIOTIN, HRP-linked antibody (Cell Signaling Technology) with a dilution ratio of 1:1000. The sequences of ATG-FLAG-AviTag and sgRNA used are listed in Table EV6.” at page 21, line 539.

TEAD2 occupies more binding sites in 2iL-ESCs and binds to active chromatin

regions to regulate the expression of 2i-specific genes

“we generated stable 2iL- and SL-ESC lines with endogenous expression of biotin-tagged TEAD2 through CRISPR/Cas9 technique (Fig. 4A), confirmed by Western blot (Fig. 4B). Subsequent BIOTIN ChIP-seq experiments identified 24,994 and 5,837 peaks in 2iL-ESCs and SL-ESCs, respectively. Motif enrichment analysis indicated significant enrichment of TEAD2 binding motifs in both 2iL- and SL-ESCs (Fig. 4C). Notably, 10,315 specific peaks were identified in 2iL-ESCs, while only 47 were specific to SL-ESCs, suggesting a potential regulatory role for TEAD2 in 2iL-ESCs (Fig. 4D, E). These 2i-specific peaks predominantly enriched in intergenic regions with more open chromatin regions (Fig. 4F, G). About 43.13% (4,489/10,315) of TEAD2 peaks localize to either promoters or enhancers (Fig. 4H). This observation implies the potential involvement of TEAD2 in gene expression regulation.” at page 11, line 269.

Figure 4

Figure 4. TEAD2 binds to the active chromatin regions of 2i-specific genes.

- A.** Strategy for generating of *Tead2*-FLAG-AviTag knock-in cell lines in 2iL- and SL-ESCs.
- B.** Western blot analysis for BIOTIN and V5 with cell lysates from *Tead2*-FLAG-AviTag-knock-in 2iL- and SL-ESC lines. β -ACTIN was used as a loading control.
- C.** Motif-enrichment analysis of BIOTIN-binding sites in 2iL- and SL-ESCs.
- D.** Heatmap showing the comparison of TEAD2 binding sites between 2iL- and SL-ESCs.
- E.** Number of 2iL- and SL-ESCs-specific TEAD2 binding peaks.
- F.** Pie charts showing the genomic distribution of 2i-specific TEAD2 peaks.
- G.** Heatmaps of sequence read density for ATAC-seq in 2i-specific TEAD2 binding peaks.
- H.** Bar plot showing the number of 2i-specific TEAD2 peaks that overlap with both promoters and enhancers.

5) The authors performed most of their experiments at day6 of SL to 2iL transition. The authors need to show the long-term effects of *Tead2* deletion in fully adapted 2iL ESCs (> d20). For example, what is the phenotype of *Tead2* ESCs cultured for long term in 2iL in terms of AP-staining, transcriptional profiles and differentiation capacity (exit from pluripotency e.g in N2B27 medium)? The same is true for chromatin interactions: do *Tead2* KO ESCs show similar or different chromatin interactions in long term 2iL or are these interactions are buffered out over time indicating that *Tead2* is not sufficient to mediate these interactions.

Answer: We appreciate these questions. As key factors regulating the accessibility of 2iL- and SL-chromatin are enriched in the initial stages (day 0-6) of Region 2/3 (Fig. 2A, B), and their expression levels showed significant fluctuations on day three of the transition (Fig. EV3A, B). Therefore, we performed the functional analysis for pluripotency, differentiation potential at the initial stages and added these data as “*The loss of *Tead2* resulted in cell deformation but did not lead the cells to exit pluripotency during SL-to-2iL transition; instead, the cells underwent spontaneous differentiation after removing 2i and LIF (Fig. EV6E). The expression of core pluripotent factors *Oct4*, *Sox2*, and *Nanog* also showed no significant alteration (the change was less than 1.5-fold) in both wild-type and *Tead2*-knockout cells on day 0*

and day 6 of the transition (Fig. EV6F).” into our new manuscript at page 9, line 222.

Figure EV6. TEAD2 has no effect on the core pluripotency establishment of ESCs.

E. Representative images of AP staining of wild-type and *Tead2*-knockout cells that were adapted to 2i conditions for days 0, 3, and 6. Cells were then induced for 5 days of differentiation in medium in the absence of LIF or 2i with LIF.

F. RT-qPCR testing the expression of pluripotent genes in wild-type and *Tead2*-knockout cells on day 0 and day 6 of the transition. Data are presented as the mean \pm SD. Indicated significances are testing using Student’s *t*-test analyses ($*p < 0.05$, $**p < 0.01$, $***p < 0.001$). $n = 3$ biological replicates.

Then we have measured 2i/serum-specific gene expression changes of the cells at day 0 and 6 of the transition and added these data into our new manuscript as “Cluster 3 genes (C3, 472 genes), involved in muscle structure and utero embryonic development (such as *Mmp2* and *Ank*), demonstrated high expression after *Tead2* loss during SL-to-2iL transition (Fig. 3E, F; Table EV3). Concurrently, cluster 5 genes (C5, 1210 genes), associated with carbohydrate and lactate metabolic processes (such as *B4galt6*, *Kit*, *Idh2*, and *Ldhb*), experienced downregulation after *Tead2* loss during SL-to-2iL transition (Fig. 3E, G; Table EV3).” at page 10, line 241.

Figure 3

Figure 3. Effect of *Tead2*-knockout on colony formation and gene expression during SL-to-2iL transition.

F and G. RT-qPCR analysis testing the expression of serum-specific genes (F) and 2i-specific genes (G) in wild-type and *Tead2*-knockout cells on day 0 and day 6 of the transition. Data are presented as the mean \pm SD. Indicated significances are testing using Student's *t*-test analyses (* $p < 0.05$, ** $p < 0.01$, *** $p < 0.001$). $n = 3$ biological replicates.

We have also performed long-term culture for *Tead2* knockout cells in 2i/LIF medium for a prolonged period (21 days) by following the Reviewer#1' suggestion, and analyzed cell morphology, pluripotency, differentiation potential, transcriptomic profiles as well as chromatin structure changes. Our new data indicated that “*Tead2*-knockout cells exhibited persistent cellular phenotype, abnormal gene expression patterns, and cluster 5 gene changes even after long-term culture in 2i/LIF (Fig. EV7A-D). Consistent with the results of D6, *Tead2* knockout led to the downregulation of 2i-specific genes and the upregulation of serum-specific genes at D15 and D21, respectively, during the transition (Fig. EV7E, F).” We have added these new data into our new manuscript at page 10, line 247.

Figure EV7

Figure EV7. Knockout of *Tead2* leads to an abnormal phenotype that can be maintained for an extended period during the SL-to-2iL transition.

A. Representative cellular morphology of wild-type ESCs and *Tead2*-knockout ESCs during the SL-to-2iL transition at days 0, 6, 15, and 21. Scale bar, 100 μ m.

B. Representative images of AP staining of wild-type and *Tead2*-knockout cells that were adapted to 2i conditions for 15 and 21 days. Cells were then induced for differentiation for 5 days in medium in the absence of LIF or 2i with LIF.

C. PCA of the RNA-seq data from wild-type ESCs and *Tead2*-knockout ESCs collected at different time points during the SL-to-2iL transition.

D. Heatmap showing the expression of cluster 5 genes in wild-type and *Tead2*-knockout cells on day 6 and 21 during the transition.

E and F. RT-qPCR testing the expression of 2i-specific genes (E) and serum-specific genes (F) in wild-type and *Tead2*-knockout cells on days 0, 6, 15, and 21 of the transition. Data are presented as the mean \pm SD. Indicated significances are tested using Student's *t*-test analyses ($*p < 0.05$, $**p < 0.01$, $***p < 0.001$). $n = 3$ biological replicates. The fold changes of these specific gene expressions after knocking *Tead2* out were calculated by using the wild-type of D6, D15, and D21 as controls, respectively.

The aggregate peak analysis (APA) scores showed that frequency for the reduced EP interactions at day 6 were also decreased at day 21 after *Tead2* loss (Fig. R2).

Figure R2. Heatmaps showing APA of differential TEAD2-mediated EP interactions in both wild-type and *Tead2*-knockout cells at day 21 of the transition.

6) It has been concluded that TEAD2 plays an important role in SL to 2iL transition based on siRNA knockdown/knockout experiments. A gain of function expression (overexpression of TEAD2) is needed to further support whether TEAD2 enhance SL-to-2iL conversion.

Answer: We have followed the Reviewer's suggestion and generated cell lines with *Tead2* overexpression, and then performed SL-to-2iL transition. Our data indicated that *Tead2* overexpression did not enhance SL-to-2iL transition but conferred SL-ESCs with expression of partial 2i-specific genes. We have added the detailed method and results into the new manuscript as follows:

“Generation of Tead2 stably overexpressed SL-ESC lines. Tead2 cDNAs were cloned into the pSin-Flag vector. The plasmids used for the transfections were purified with a HiPure Plasmid EF Mini Kit (Magen, P1112-03). The sequences of primers used for the Tead2 CDS amplification are listed in Table EV8. All constructs were confirmed

by Sanger sequencing. Then, lentivirus for pSin-Flag, pSin-Flag-Tead2, were assembled with psPAX2, pMD2.G vectors in HEK293T cells. Then lentiviral supernatants were collected and transfected using the modified PEI. SL-ESCs were then infected with pSin-Flag and pSin-Flag-Tead2 lentivirus, respectively. The positive cells were selected with 2 µg/mL puromycin for 5 days.” at page 22, line 564.

Tead2 overexpression did not enhance SL-to-2iL transition but conferred SL-ESCs with expression of partial 2i-specific genes

“To investigate the impact of Tead2 overexpression on SL-to-2iL conversion, we ectopically expressed Tead2 in SL-ESCs (**Fig. EV8A, B**) and conducted SL-to-2iL transition experiments in both control and Tead2-overexpressed mESCs. Tead2-overexpressed cells exhibited normal morphological changes compared to control cells (**Fig. EV8C**). Tead2 overexpression had no discernible effect on AP staining and the expression levels of core pluripotent genes on day 6 of the transition (**Fig. EV8D, E**). Meanwhile, most 2i- and serum-specific genes showed minimal changes on day 6 of the transition but were significantly up- and down-regulated, respectively, on day 0 (**Fig. EV8F, G**). These results suggest that Tead2 overexpression does not impact SL-to-2iL transition but induces the expression of 2i-specific genes in SL-ESCs.” at page 10, line 255.

Figure EV8

Figure EV8. *Tead2* overexpression endows SL-ESCs with the expression of partial 2i-specific genes.

A and B. RT-qPCR (A) and Western blot (B) analysis examining *Tead2* overexpression in SL-ESCs. Data are presented as the mean \pm SD. Indicated significances are tested using Student's *t*-test analyses (* $p < 0.05$, ** $p < 0.01$, *** $p < 0.001$). $n = 3$ biological replicates.

C. Representative cellular morphology of ESC colonies with the Flag control and Flag-*Tead2* overexpression during the SL-to-2iL transition at days 0, 3, and 6, respectively. Scale bar, 100 μ m.

D. Representative images of AP staining of the cells with the Flag control and Flag-*Tead2* overexpression that were adapted to 2i conditions for 0, 3, and 6 days.

E-G. RT-qPCR analysis testing the expression of pluripotency genes (E), 2i-specific genes (F), and serum-specific genes (G) in Flag control and Flag-*Tead2* overexpression cells on day 0 and day 6 of the transition. Data are presented as the mean \pm SD. Indicated significances are tested using Student's *t*-test analysis (* $p < 0.05$, ** $p < 0.01$, *** $p < 0.001$). $n = 3$ biological replicates.

7) The authors suggest that some E-P interactions are dependent Tead2. The authors need to show whether these interactions normally change during SL-to-2iL transition.

Answer: To answer this question, we have added the APA for the decreased EP interactions at D0 of the transition. Furthermore, we added the following sentence into the manuscript: *“Aggregate peak analysis (APA) scores confirmed the normal increase in frequency for these EP interactions during the SL-to-2iL transition but a significant decrease after Tead2 loss (Fig. 5E), indicating disruption caused by Tead2 knockout in the frequency of EP interactions during this transition.”* in the new manuscript at page 14, line 339.

Figure 5

Figure 5E. Heatmaps showing APA of differential TEAD2-mediated EP interactions in both wild-type and *Tead2*-knockout cells at day 0 and day 6 of the transition.

8) What are the 118 genes that are 2i-specific genes directly bound by Tead2 and are affected by Tead2 KO? How are they involved in 2i-ESCs biology?

Answer: 118 genes bound by TEAD2 were identified from previous BIOTIN-tagged TEAD2-ChIP-seq experiments. Since we have also generated endogenous knock-in TEAD2-BIOTIN cell lines and then perform ChIP-seq experiments, we identified 192 genes that were directly targeted by TEAD2 and downregulated after *Tead2* loss. We have added this gene list as **Table EV4**. Furthermore, GO analysis showed that these TEAD2 bound genes were involved in the glycolipid metabolic process and so on, in which were typical gene terms of 2iL-ESCs as described previously (Marks *et al*, 2012; Marks & Stunnenberg, 2014).

We have incorporated these new findings into the revised manuscript: *“We then identified the genes directly bound by TEAD2, with approximately 15.9% (192/1,210)*

of cluster 5 genes being targeted by TEAD2 (**Fig. 4J; Table EV4**). Notably, these 192 genes exhibited reduced expression following *Tead2* loss (**Fig. 4K**). GO analysis indicated their involvement in glycolipid metabolic processes and lipid catabolic processes (**Fig. 4L**)” at page 12, line 294.

Figure 4. TEAD2 binds to the active chromatin regions of 2i-specific genes.

J. Venn plots showing the overlap among 2i-specific TEAD2 target genes and cluster 5 genes.

K. Boxplots showing expression level of overlapping genes in (J) between wild-type and *Tead2*-knockout cells at day 6 of the transition.

L. GO categories of the overlapping genes shown in (J).

9) *Tead2* and *Tead4* have similar motifs. Does *Tead4* KO phenocopy *Tead2* KO phenotype? Also since the TEA motif is shared with other TEADs, what about *Tead1* and 3 expression?

Answer: This is an interesting question. Based on the previous publications and our RNA-seq data in *Tead4* knockdown experiments during SL-to-2iL transition, we demonstrated the unique regulatory role of TEAD2 but no other TEADs in this process. We have added this part of the content to the **Discussion** section of the new manuscript. The details are as follows:

“TEAD transcription factors possess N-terminal domains (TEA) binding to DNA and C-terminal domains (YBD) interacting with YAP/TAZ (Anbanandam et al, 2006; Bürklin, 1991). Individually, TEA and YBD exhibit high homology within the TEAD family (Fig. EV12A). Despite this, TEADs serve diverse functions during early embryonic development and various organogenesis processes (Currey et al, 2021). In mice, TEAD1 and TEAD3 share similar DNA binding motifs, while TEAD2 and

TEAD4 feature more analogous motifs (Fig. EV12B). Our findings disclosed higher expression levels of *Tead1-4* in 2iL-ESCs compared to SL-ESCs, with *Tead1* being predominant, *Tead3* barely detectable, and *Tead2* and *Tead4* at comparable levels (Fig. EV12C). Previous research has established that knocking down *TEAD1/3/4* in ESCs results in *Oct4* and *Sox2* downregulation and loss of pluripotency (Lian et al, 2010). Intriguingly, the 2i-specific chromatin accessibility sites (Region 2/3) did not exhibit enrichment for *TEAD1/3* motifs (Fig. 2A, B). We postulate that *TEAD1/3* expression in ESCs might be linked to the core pluripotency network, whereas *TEAD2/4* are implicated in the ground-state pluripotency. Despite the absence of a discernible phenotype following *Tead4* depletion (Fig. 2C-F), RNA-seq experiments were conducted to explore potential redundancy between *TEAD2* and *TEAD4* and identify genes regulated by *Tead4* knockdown. Consistently, *Tead4* knockdown minimally affected gene expression during the transition (Fig. EV12D-F), suggesting that *TEAD2*, but not *TEAD4*, modulates the SL-to-2iL transition. And *TEAD4* is absent in nucleus of ESCs (Home et al, 2012), in which might compromise the function as *TEAD4* in regulating SL-to-2iL transition.” at page 17, line 418.

Figure EV12

Figure EV12. TEAD4 and TEAD2 are not redundant during SL-to-2iL transition.

A. Schematic of the overall structure of the mammalian TEAD factors. The four TEADs present an overall homology and are divided into a TEA domain at the N-terminus (67 aa) and a C-terminal YAP/TAZ binding domain (YBD) (about 215 aa). Both domains are linked by a sequence of about 117–143 amino acids which has a low homology across the four TEADs. TEA: TEA DNA binding domain. YBD: YAP binding domain.

B. RT-qPCR analysis testing the expression of *Tead1-4* in 2iL- and SL-ESCs. Data are presented as the mean \pm SD. Indicated significances are tested using Student's *t*-test analyses ($*p < 0.05$, $**p < 0.01$, $***p < 0.001$). $n = 3$ biological replicates.

C. Sequence LOGOs of TEAD1-4 motifs enriched in ESCs.

D. Hierarchical cluster analysis (HCA) and heatmaps of control and *Tead4*-depleted cells at day 0 and day 6 of the transition. The heatmaps were based on rlog-transformed and DESeq2-normalized expression data. The color key shows the Euclidean distances between samples.

E and F. Volcano plots showing differential gene expression (fold change > 2 ; q -value < 0.05) between control and *Tead4*-depleted cells at day 0 (E) and day 6 (F) of the SL-to-2iL transition.

10) Fig 4C: how did the authors specified 'enhancers' in this analysis? Are these ATAC-seq or H3K27ac, or simply intergenic regions?

Answer: We defined the regions marked by both H3K27ac and H3K4me1 as the enhancers.

11) Fig 4D: The authors should add the previously published ChIP-seq profiles of SL ESCs to establish a link between Tead2 binding and change of histone marks at Tead2 binding sites in 2iL vs SL ESCs.

Answer: We have followed the Reviewer's suggestion and collected the published ChIP-seq data of H3K27me3, H3K36me3 and H3K9me3 from GSE23943, H3K4me3 from GSE157748, H3K4me1 and H3K27ac from GSE72164 in 2iL- and SL-ESCs to further demonstrate that “*2i-specific TEAD2 sites predominantly marked by active histone marks, notably H3K27ac, H3K4me1, and H3K4me3 (Fig. 4I). Furthermore, these active histone marks exhibited a more pronounced enrichment at these sites in 2iL-ESCs compared to SL-ESCs (Fig. 4I)*”. And we have added these new data into

Figure 4 in our new manuscript at page 12, line 291.

Figure 4

Figure 4I. Tag-density pileup showing H3K4me1, H3K27ac, H3K4me3, H3K27me3, H3K9me3, and H3K36me3 ChIP-seq signals at the 2i-specific TEAD2 binding sites in both 2iL- and SL-ESCs.

12) Does Tead2 deletion affect epigenetic marking at Tead2 binding sites (proxy to enhancer activity)? The authors should perform H3K27ac/H3K4me1 ChIP-seq in Tead2 KO vs WT ESCs to address this point.

Answer: By following the Reviewer’s suggestion, we have performed CUT&Tag experiments for both H3K27ac and H3K4me1 in both wild-type and *Tea2* knockout cells at day 6 of transition and found that “*CUT&Tag analysis for wild-type and TEAD2-knockout cells revealed minimal alteration in the enrichment of both H3K4me1 and H3K27ac marks at these sites, suggesting that loss of TEAD2 does not diminish enhancer activity (Fig. EV10A, B)*” We have added these results into **Figure EV10** of our new manuscript at page 13, line 325.

Figure EV10

Figure EV10. Supplementary data of ChIP-seq experiments.

A and B. Boxplots showing the CUT&Tag signal of both H3K27ac (A) and H3K4me1 (B) at TEAD2 binding sites in wild-type and *Tead2*-knockout cells, respectively, at day 6 of the SL-to-2i transition

13) What is the epigenetic marking at interactions significantly affected by Tead2 deletion? What makes them different from other E-P interactions?

Answer: Following the Reviewer's suggestion, we have performed more bioinformatic analysis and found that TEAD2 could also downregulate the E-E and P-P interactions at day 6 during SL-to-2iL transition (Fig. R3A). Furthermore, based on the ATAC-seq and ChIP-seq data from different epigenetic histone marks, we found that the anchor regions of E-P, E-E, P-P were enriched with the active epigenetic marks (Fig. R3B). In addition, the anchors for E-E interaction were enriched with more H3K27ac and H3K4me1 signal than those for E-P interactions. The anchors for P-P interactions were enriched with more H3K4me3 signal than those for E-P interactions (Fig. R3B).

Figure R3.

A. Bar plots showing the number of enhancer-promoter (E-P), enhancer-enhancer (E-E), and promoter-promoter (P-P) interactions downregulated after *Tead2* deletion.

B. Tag-density pileup showing ATAC-seq and H3K4me1, H3K27ac, H3K4me3, H3K27me3, H3K9me3, H3K36me3 ChIP-seq signals at the anchor regions of E-P, E-E, P-P interactions.

14) Fig 6G: The authors claim that deletion of Tead2 motif at B4galt6 promoter reduces enhancer-promoter interactions. To support this the authors performed 4C-seq in WT and *tead2*-motif-Mut ESCs. However, this experiment is not informative and is

missing many controls. First, the Tead2-KO samples should be included in this experiment as control. Secondly, it's not possible to conclude a reduction in the interaction frequency between WT and MUT cells based on this figure. The authors need to perform differential analysis of interaction between WT and Mut cells and show whether these changes are statistically significant (e.g as shown before. Thirdly, the induced deletion in MUT cells is 64 bp long. How can they be sure that the effect that they are seeing (although subtle) is due to loss of Tead2 binding and no other motifs in this fragment? The authors need to confirm their data by mutating the motif sequence using base substitution.

Answer: We are really appreciated for these very valuable questions and suggestions. Based on the Reviewer's comments, we further constructed a cell line with base substitution in the TEAD2 motifs of the *B4galt6* promoter region, and performed quantitative high-resolution chromosome conformation capture copy (QHR-4C) experiments in wild-type 2iL-ESCs, two homozygous mutant 2iL-ES cell lines (four to five base substitution), wild-type and *Tead2*-KO D6 cells. We also added statistical analysis to further demonstrate that mutation of TEAD2 binding motifs leads to loss of EP interactions in 2i-specific *B4galt6* gene. In our revised manuscript, we have removed the previous data with a cell line that had 64 bp (including two TEAD2 motifs) deleted from the *B4galt6* promoter region, instead using the new data from the cell lines with base substitution. The details are as follows:

Generation 2iL-ESC lines with mutation of TEAD2 motifs in the B4galt6 promoter region

“We generated 2iL-ESCs with mutated TEAD2 motifs by using CRISPR/Cas9. The B4galt6 promoter region had a TEAD2 peak sequence (704 bp) with two TEAD2 binding motifs. We replaced the TEAD2 motifs with XhoI and HindIII restriction sites by PCR and DpnI digestion. We cloned this sequence with the restriction sites and the 5' and 3' homology arms from the genome into the pEASY-Blunt vector as a donor construct. We designed sgRNAs with an online website tool (<http://benchling.com>), then synthesized, annealed and cloned the sgRNA primers into the pX459 vector. We transfected these vectors into 2iL-ESCs for genomic editing with FuGENE® 6

transfection reagent (Promega) as described above. Then we selected positive clones with 2 µg/mL puromycin for 5 days. To verify corrected clones, we used the cloned genome as a template, PCR with primers F and R were performed to obtain the 704 bp sequence containing Xho I and Hind III sites. The PCR products of the clones with homozygous mutations were digested with Xho I enzyme to yield bands of 155 bp and 549 bp, and with Hind III enzyme to yield bands of 292 bp and 412 bp. Then the clones were verified by Sanger sequencing. The primer and sequences of sgRNA used are listed in Table EV6.” at page 23.

Mutation of TEAD2 binding motifs cause loss of EP interactions in 2i-specific B4galt6 gene.

“Based on the endogenous biotin-tagged TEAD2 ChIP-seq data in 2iL-ESCs, two putative TEAD2 binding motifs were identified in the promoter region of the B4galt6 gene. Subsequently, base substitutions were introduced into these two TEAD2 binding motifs in 2iL-ESCs using CRISPR/Cas9 (Fig. 6E). By digesting the genomic DNA with XhoI and HindIII (Fig. EVIIB) and performing Sanger sequencing, we finally yielded two homozygous clones with both TEAD2 binding motifs mutated (Fig. EVIIC, D). RT-qPCR results indicated that the loss of TEAD2 motifs at the gene promoter in these two mutant clones had no effect on the expression of Tead2 (Fig. 6F) but resulted in lower expression of B4galt6 (Fig. 6G). To further demonstrate that the downregulation of B4galt6 gene expression in the mutant clones resulted from the attenuation of TEAD2-mediated EP interactions, quantitative high-resolution chromosome conformation capture copy (QHR-4C) experiments were performed in wild-type and two mutant 2iL-ESCs, wild-type and Tead2-knockout cells at day 6 of transition. The results showed that, similar to Tead2 knockout, the frequency of EP interactions at the B4galt6 gene locus with TEAD2 binding peaks was significantly reduced in the two mutant clones compared to wild-type 2iL-ESCs (Figs. 6H and EVIIE). Tead2 knockout had no effect on the levels of H3K4me1 and H3K27ac on these EP interactions at day 6 of SL-to-2iL transition (Fig. 6H). Additionally, there was no change in H3K27ac enrichment at the B4galt6 locus in either of the mutant clones (Fig. EVIIF). These results collectively demonstrate that TEAD2 contributes

to EP interactions for 2i-specific genes.” at page 15.

Figure 6. Mediation of EP interactions by TEAD2 may involve Cohesin but not structural proteins such as YY1 and CTCF.

E. Strategy of generating 2iL-ESC lines with TEAD2 motif mutations at the *B4galt6* promoter region.

F and G. RT-qPCR analysis testing the expression of *Tead2* (F) and *B4galt6* (G) in wild-type 2iL-ESCs and two homozygous clones with base alterations in both TEAD2 binding motifs. Data are presented as the mean \pm SD. *p*-values were determined using the two-sided Student’s *t*-test.

H. 4C tracks showing the interactions between the promoters and enhancers of *B4galt6* in wild-type and two mutant 2iL-ESCs, wild-type and *Tead2*-knockout cells at day 6 of the transition. The anchor region from QHR-4C is indicated.

Figure EV11

Figure EV11. Supplementary data of the QHC-4C experiments.

B. Restriction enzyme digestion strategy for identifying mutant clones.

C. Genomic PCR and enzyme digestion to verify corrected clones.

D. Sanger sequencing testing the region containing two TEAD2 motifs in wild-type and two homozygous mutant clones of 2iL-ESCs.

E. Barplot showing the normalized interaction frequency between the promoters and enhancers of *B4galt6* in wild-type and two mutant 2iL-ESCs, wild-type and *Tead2*-knockout cells at day 6 of the transition.

F. Genomic views of enrichment for H3K27ac at the *B4galt6* gene in wild-type and two homozygous mutant clones of 2iL-ESCs.

Minor points:

1) Fig EV3: Y-axis represents 'fold change'. What is compared here?

Answer: We used siNC samples collected at day 0 of SL-to-2iL transition as the references for all genes. To clarify the results, we have generated new graphs, and adjusted Fig EV3A to **Fig 2C** to demonstrate the knockdown efficiency for these

genes.

Figure 2

Figure 2C. RT-qPCR testing siRNA knockdown efficiencies for *Tead2*, *Tead4*, *Tcfcp2l1*, *Nr5a2*, and *Esrrb* during SL-to-2iL transition. Cells were treated with specific siRNAs for every 3 days along with control siRNA. Data are shown as the mean \pm SD. Indicated significances are tested using Student's *t*-test analyses ($*p < 0.05$, $**p < 0.01$, $***p < 0.001$). $n = 3$ biological replicates.

2) What is the average distance of E-P interactions mediated by Tead2. The authors should compare this with other E-P, P-P and E-E interactions.

Answer: The average distance of E-P interactions mediated by TEAD2 is 227,134 bp. We found that the distance of E-P, E-E, P-P interactions mediated by TEAD2 were more than others which were not mediated by TEAD2.

Figure R4

Figure R4. Barplot showing the average loop width of enhancer-promoter (EP), enhancer-enhancer (EE), and promoter-promoter (PP) interactions with or without TEAD2.

3) Please include the % of E-P or % of all interactions that are significantly different between KO vs WT ESCs?

Answer: Following the reviewer's suggestion, we have added that 28.95% of E-P that

are significantly different between KO vs WT ESCs. We have added these data into the new manuscript as “Analyzing TEAD2 binding peaks alongside BL-Hi-C data, we observed a significant reduction (28.95%) in interactions between TEAD2-occupied enhancers and promoters following *Tead2* knockout (Fig. 5A)” at page 13, line 330.

4) Authors show that loss of TEAD2 leads to lower SMC1 occupancy at enhancer anchors. It is helpful to show the expression level of SMC1 in the KO cells.

Answer: Following the reviewer’s suggestion, we have measured the expression level of *Smc1* after *Tead2* knockout and found that *Tead2* knockout slightly increased *Smc1* expression in wild-type and *Tead2*-knockout cells on day 0 and day 6 of the transition (Fig. R5), suggesting that, although *Tead2* loss leads to lower SMC1 occupancy at enhancer anchors, *Tead2* loss has little effect on the expression of *Smc1*.

Figure R5

Figure R5. RT-qPCR analysis testing the expression of *Smc1* in both wild-type and *Tead2*-knockout cells on day 0 and 6 of the transition. The data are reported as mean values \pm SD with the indicated significance by using Student’s *t*-test analysis (* $p < 0.05$, ** $p < 0.01$, *** $p < 0.001$). $n = 3$ biological replicates.

5) Fig 3B legend says YY1 mediated interactions whereas the figure shows Tead2 dependent interactions.

Answer: We sincerely apologize for this error and have corrected it in the legend of **Figure 5B** as “Changes in the normalized interaction frequency (\log_2 fold change) in cells at day 6 during the SL-to-2iL transition upon *Tead2*-knockout in the following

three different categories: all interactions, interactions unrelated to **TEAD2** binding sites, and **TEAD2**-mediated EP interactions.” in the new manuscript.

6) FigEV8-F legends does not correspond to the figure.

Answer: We really apologize for these mistakes. In the updated manuscript, Figure EV8 was adjusted as **Figure EV9**, we have corrected the captions of Figure EV9 F-I to match each figure. The details are as follows:

Figure EV9

- **F and G.** A/B compartment analysis at *Mmp2* (F) and *Arhgef26* (G) loci in both wild-type and *Tead2*-knockout cells at day 6 of the SL-to-2iL transition from BL-Hi-C experiments.
- **H and I.** RT-qPCR detecting the expression levels of *Mmp2* (H) and *Arhgef26* (I) genes. Data are presented as the mean \pm SD. *p*-values were determined by two-sided Student's *t*-test.

7) Figure6-G and H are swapped and do not correspond to the figure legends

Answer: We are really appreciated for the Reviewer's careful assessment. We have removed these data and figures from current version of manuscript.

Referee #2:

In this manuscript, Guo et al. monitor transcription and changes of accessibility over time after switching culture conditions of mouse embryonic stem cells (ESCs) from serum/LIF, where the cells are more prone to differentiation, to 2i/LIF media, representing a stable naïve pluripotent state arguably closer to that of epiblast cells, and vice-versa. They identify genomic regions becoming accessible or losing accessibility during these transitions, and search for transcription factor motifs enriched in the underlying DNA sequences, in an attempt to pinpoint new regulators of the conversion. Among the candidates identified is Tead2. The authors show that knockdown of this factor deteriorates the morphology of ESCs upon transfer to 2i/LIF. Knockout lines are then derived, showing that the loss of Tead2 affects gene expression changes that normally accompany the transition to stable naïve pluripotency. These effects appear due to direct regulation, as determined analyzing the genome-wide DNA binding profile of Tead2. The authors go further, determining that Tead2 inactivation compromises the reorganization of some aspects of the 3D genome organization of ESCs subject to media switch. Finally, setting up an elegant genetic experiment, the authors show that the ability of Tead2 to bind to specific genomic regions is required for rewiring spatial organization at these loci, and conclude that the effects of Tead2 in this context seem independent from other regulators of chromatin looping, such as CTCF and YY1. Overall, this paper identifies in Tead2 a regulator of naïve pluripotency.

Major concerns and criticisms:

While, globally, the analysis is technically sound - the experiments are performed and analysed appropriately - the work presented in the manuscript seems incomplete, preventing the reader from fully judging the importance of Tead2 in pluripotent cells. This is put in evidence by the concluding remarks of the authors that read: "In summary, we revealed that TEAD2 is essential for maintaining the ground state pluripotency by regulating the expression of 2i-specific genes ...". Unfortunately, no

evidence is provided in the manuscript that allows determining whether Tead2 expression is indeed required to maintain ESCs in 2i/LIF, besides a transient knockdown experiment. This is particularly surprising, given that the authors have generated all the tools required to address this possibility. In short, are Tead2 KO ESCs still able to self-renew in 2i/LIF? The fact that TEAD2 is dispensable during pre-implantation development does not appear to suggest an essential role in pluripotent cells. The relevance of the study depends on this distinction: it is debatable that the serum to 2i transition represents a model of any direct developmental relevance. Moreover, irrespectively of the fact that the gene might be dispensable or essential, it is important that the authors include a rigorous characterization of the effect of the loss of Tead2 in cells stably maintained in 2i/LIF. Do the changes in gene expression and genomic organization reported at day 6 of the conversion eventually normalise?

Answer: We are appreciated for the Reviewer's comments and constructive suggestions. By following the Reviewer's suggestion, we further conducted four additional functional experiments to elucidate the critical role of TEAD2 as an ancillary factor in maintaining the ground-state pluripotency of 2iL-ESCs and facilitating the transition from SL-ESCs to 2iL-ESCs.

1.To screen for key regulatory factors during SL-to-2iL transition, we further depleted our selected TF candidates and assessed the effects of knockdown on cell pluripotency and self-renewal during SL-to-2iL transition, revealing that *Tead2* knockout disrupts cell morphology and self-renewal without affecting pluripotency (Figure 2C-F).

The details are as follows:

*“Our findings revealed that knocking down either *Tead2* (*siTead2*) or *Nr5a2* (*siNr5a2*) impeded domed colony formation during SL-to-2iL transition, with *Tead2*-knockdown exhibiting a more pronounced effect than *Nr5a2* knock down (Fig. 2D). Conversely, knockdown of the other three factors (*siTead4*, *siEsrrb*, and *siTcfcp211*) had minimal effects on cell morphology (Fig. 2D). With the exception of *siTead4*, knockdown of any of the other four factors diminished self-renewal and proliferation capacity to*

varying degrees, with *siTead2* and *siTcfcp211* also impacting day 0 during the transition (**Fig. 2E**). *ESRRB* and *TCFCP2L1* play crucial roles in stabilizing the regulatory network of naïve pluripotency and preventing the loss of self-renewal (Atiasi et al., 2019; Festuccia et al., 2018a; Festuccia et al., 2018b; Hackett & Surani, 2014; Qiu et al., 2015; Zhang et al., 2021). *NR5A2* can form a regulatory module with *ESRRB* to assist in the binding of core pluripotency factors at most of their occupied regions, thereby regulating the naïve pluripotency network (Festuccia et al., 2021). However, the mechanism by which *TEAD2* regulates ground-state pluripotency remains unknown. Despite displaying a flattened clone-like morphology, *Tead2*-knockdown cells retain their pluripotency, similar to cells with another factor knockdowns (**Fig. 2F**). We speculated that *TEAD2* may not directly influence the core pluripotency but instead regulates the formation of the ground-state pluripotency during *SL-to-2iL* transition". And we have added these new data into our new manuscript at page 7, line 173.

Figure 2.

D. Representative images of cells transfected with siNC (negative control) and siRNAs for *Tead2*, *Tead4*, *Tcfcp211*, *Nr5a2*, and *Esrrb* during the SL-to-2iL process. Scale bar, 100 μ m.

E. Number of cells transfected with siNC (negative control) and siRNAs of *Tead2*, *Tead4*, *Tcfcp211*, *Nr5a2*, and *Esrrb*. Cells were grown for 3 days during the SL-to-2iL process. Data are presented as the

mean \pm SD of three independent wells.

F. Representative images of AP staining of cells transfected with siNC (negative control) and siRNAs for *Tead2*, *Tead4*, *Tcfcp2l1*, *Nr5a2* and *Esrrb* during SL-to-2iL process. Scale bar, 100 μ m.

2. We have conducted *Tead2* knockdown experiments in both 2iL-ESCs and SL-ESCs, re-evaluating their impact on cell morphology, pluripotency, differentiation capacity, and the expression of 2i- and serum-specific genes. Our findings indicate that *Tead2* knockdown did not affect pluripotency or the expression of core pluripotency factors *Oct4* and *Sox2* in either 2iL-ESCs or SL-ESCs. However, it led to a morphological and gene expression shift in 2iL-ESCs, making them more similar to SL-ESCs.

The details are as follows:

*“To assess the importance of TEAD2 in 2iL-ESCs, we transfected *Tead2* siRNAs into both 2iL- and SL-ESCs. The results showed that knockdown of *Tead2* had little effect on the expression of core pluripotent factors *Oct4*, *Sox2*, and *Nanog* (the change was less than 1.2 times) in both ESCs (Fig. EV4A-D). However, *Tead2* knockdown induced heterogeneity in 2iL-ESCs, exhibiting a morphology similar to that of SL-ESCs (Fig. EV4B). During spontaneous differentiation after removing 2i/LIF or LIF, 2iL-differentiated cells with *Tead2* knockdown resembled SL-differentiated cells in morphology (Fig. EV4B). *Tead2* knockdown had a minor effect on SL-ESCs (Fig. EV4B). Furthermore, *Tead2* knockdown upregulated serum-specific genes and downregulated 2i-specific genes in 2iL-ESCs (Fig. EV4E). In contrast, *Tead2* knockdown in SL-ESCs did not induce upregulation of serum-specific genes and had no or minor effect on downregulation of 2i-specific genes (Fig. EV4F). These results suggest that TEAD2 does not participate in regulating the core pluripotency of mESCs but instead stabilizes the ground-state regulatory network of 2iL-ESCs to prevent them from entering a metastable state.”* at page 8, line 195.

Figure EV4

Figure EV4. Effect of TEAD2 on the ground-state pluripotency in 2iL-ESCs.

A. RT-qPCR determining the *Tead2* knockdown efficiency in 2iL-ESCs and SL-ESCs. Data are presented as the mean \pm SD. Indicated significances are testing using Student's *t*-test analyses (* $p < 0.05$, ** $p < 0.01$, *** $p < 0.001$). $n = 3$ biological replicates.

B. Representative images of AP staining of 2iL- and SL-ESCs and cells after 3 days of differentiation in complete medium containing 10% serum or in the absence of LIF. Cells were treated with control siRNA or siRNA targeting *Tead2*.

C and D. RT-qPCR testing the expression of pluripotent genes in 2iL-ESCs (C) and SL-ESCs (D). Data are presented as the mean \pm SD. Indicated significances are tested using Student's *t*-test analyses (* $p < 0.05$, ** $p < 0.01$, *** $p < 0.001$). $n = 3$ biological replicates.

E and F. RT-qPCR testing the expression of 2i- and serum-specific genes in 2iL-ESCs (E) and SL-ESCs (F). Data are presented as the mean \pm SD. Indicated significances are tested using Student's *t*-test analyses (**p* < 0.05, ***p* < 0.01, ****p* < 0.001). n = 3 biological replicates.

3. We have maintained *Tead2* knockout cells in 2i/LIF medium for an extended period (21 days), and we conducted analyses on cell morphology, pluripotency, differentiation potential, transcriptomic profiles, and chromatin structure changes. Our findings indicate that “*Tead2*-knockout cells exhibited persistent cellular phenotype, abnormal gene expression patterns, and cluster 5 gene changes even after long-term culture in 2i/LIF (Fig. EV7A-D). Consistent with the results of D6, *Tead2* knockout led to the downregulation of 2i-specific genes and the upregulation of serum-specific genes at D15 and D21, respectively, during the transition (Fig. EV7E, F)”. We have added these new data into our manuscript at page 10, line 247.

Figure EV7

Figure EV7. Knockout of *Tead2* leads to an abnormal phenotype that can be maintained for an extended period during the SL-to-2iL transition.

A. Representative cellular morphology of wild-type ESCs and *Tead2*-knockout ESCs during the SL-to-2iL transition at days 0, 6, 15, and 21. Scale bar, 100 μ m.

B. Representative images of AP staining of wild-type and *Tead2*-knockout cells that were adapted to 2i conditions for 15 and 21 days. Cells were then induced for differentiation for 5 days in medium in the absence of LIF or 2i with LIF.

C. PCA of the RNA-seq data from wild-type ESCs and *Tead2*-knockout ESCs collected at different time points during the SL-to-2iL transition.

D. Heatmap showing the expression of cluster 5 genes in wild-type and *Tead2*-knockout cells on day 6 and 21 during the transition.

E and F. RT-qPCR testing the expression of 2i-specific genes (E) and serum-specific genes (F) in wild-type and *Tead2*-knockout cells on days 0, 6, 15, and 21 of the transition. Data are presented as the mean \pm SD. Indicated significances are tested using Student's *t*-test analyses ($*p < 0.05$, $**p < 0.01$, $***p < 0.001$). $n = 3$ biological replicates. The fold changes of these specific gene expressions after knocking *Tead2* out were calculated by using the wild-type of D6, D15, and D21 as controls, respectively.

We also performed BL-Hi-C with wild-type and *Tead2*-knockout cells in day 21 of SL-to-2iL transition. The aggregate peak analysis (APA) scores showed that frequency for the reduced EP interactions at day 6 were also decreased at day 21 after *Tead2* deletion (Fig. R2).

Figure R2. Heatmaps showing APA of differential TEAD2-mediated EP interactions of wild-type cells and *Tead2*-knockout cells at day 21 of the transition.

4. We have generated cell lines with *Tead2* overexpression and performed

SL-to-2iL transition assays. The results revealed that *Tead2* overexpression does not enhance SL-to-2iL transition. However, it imparts SL-ESCs with the expression of partial 2i-specific genes. We have added the following statements into the manuscript as: “To investigate the impact of *Tead2* overexpression on SL-to-2iL conversion, we ectopically expressed *Tead2* in SL-ESCs (Fig. EV8A, B) and conducted SL-to-2iL transition experiments in both control and *Tead2*-overexpressed mESCs. *Tead2*-overexpressed cells exhibited normal morphological changes compared to control cells (Fig. EV8C). *Tead2* overexpression had no discernible effect on AP staining and the expression levels of core pluripotent genes on day 6 of the transition (Fig. EV8D, E). Meanwhile, most 2i- and serum-specific genes showed minimal changes on day 6 of the transition but were significantly up- and down-regulated, respectively, on day 0 (Fig. EV8F, G). These results suggest that *Tead2* overexpression does not impact SL-to-2iL transition but induces the expression of 2i-specific genes in SL-ESCs.” at page 10, line 257.

Figure EV8

Figure EV8. *Tead2* overexpression endows SL-ESCs with the expression of partial 2i-specific genes.

A and B. RT-qPCR (A) and Western blot (B) analysis examining *Tead2* overexpression in SL-ESCs. Data are presented as the mean \pm SD. Indicated significances are tested using Student's *t*-test analyses (* $p < 0.05$, ** $p < 0.01$, *** $p < 0.001$). $n = 3$ biological replicates.

C. Representative cellular morphology of ESC colonies with the Flag control and Flag-*Tead2* overexpression during the SL-to-2iL transition at days 0, 3, and 6, respectively. Scale bar, 100 μm .

D. Representative images of AP staining of the cells with the Flag control and Flag-*Tead2* overexpression that were adapted to 2i conditions for 0, 3, and 6 days.

E-G. RT-qPCR analysis testing the expression of pluripotency genes (E), 2i-specific genes (F), and serum-specific genes (G) in Flag control and Flag-*Tead2* overexpression cells on day 0 and day 6 of the transition. Data are presented as the mean \pm SD. Indicated significances are tested using Student's *t*-test analysis (* $p < 0.05$, ** $p < 0.01$, *** $p < 0.001$). $n = 3$ biological replicates.

Linked to the previous remark, how do the authors exclude a possible functional complementation by Tead4? Functional redundancy between TEAD factors is well described during development. Also, in light of the ability of Tead2 to interact with coactivators that are crucial to the function of nuclear receptors, it would be important to explore the interplay between Tead2 and this class of transcription factors.

Answer: This question was also raised by the Reviewer#1. To address this question, we further performed RNA-seq experiments of *siTead4* during SL-to-2iL transition, and elucidated the unique regulatory role of TEAD2 in this process. We have added this part of the content to the **Discussion** section of the new manuscript. The details are as follows:

“TEAD transcription factors possess N-terminal domains (TEA) binding to DNA and C-terminal domains (YBD) interacting with YAP/TAZ (Anbanandam et al., 2006; Bürklin, 1991). Individually, TEA and YBD exhibit high homology within the TEAD family (Fig. EV12A). Despite this, TEADs serve diverse functions during early embryonic development and various organogenesis processes (Currey et al., 2021). In mice, TEAD1 and TEAD3 share similar DNA binding motifs, while TEAD2 and

TEAD4 feature more analogous motifs (Fig. EV12B). Our findings disclosed higher expression levels of *Tead1-4* in 2iL-ESCs compared to SL-ESCs, with *Tead1* being predominant, *Tead3* barely detectable, and *Tead2* and *Tead4* at comparable levels (Fig. EV12C). Previous research has established that knocking down *TEAD1/3/4* in ESCs results in *Oct4* and *Sox2* downregulation and loss of pluripotency (Lian et al., 2010). Intriguingly, the 2i-specific chromatin accessibility sites (Region 2/3) did not exhibit enrichment for *TEAD1/3* motifs (Fig. 2A, B). We postulate that *TEAD1/3* expression in ESCs might be linked to the core pluripotency network, whereas *TEAD2/4* are implicated in the ground-state pluripotency. Despite the absence of a discernible phenotype following *Tead4* depletion (Fig. 2C-F), RNA-seq experiments were conducted to explore potential redundancy between *TEAD2* and *TEAD4* and identify genes regulated by *Tead4* knockdown. Consistently, *Tead4* knockdown minimally affected gene expression during the transition (Fig. EV12D-F), suggesting that *TEAD2*, but not *TEAD4*, modulates the SL-to-2iL transition. And *TEAD4* is absent in nucleus of ESCs (Home et al., 2012), in which might compromise the function as *TEAD4* in regulating SL-to-2iL transition.” at page 17, line 418.

Figure EV12

Figure EV12. TEAD4 and TEAD2 are not redundant during SL-to-2iL transition.

A. Schematic of the overall structure of the mammalian TEAD factors. The four TEADs present an overall homology and are divided into a TEA domain at the N-terminus (67 aa) and a C-terminal YAP/TAZ binding domain (YBD) (about 215 aa). Both domains are linked by a sequence of about 117–143 amino acids which has a low homology across the four TEADs. TEA: TEA DNA binding domain. YBD: YAP binding domain.

B. RT-qPCR analysis testing the expression of *Tead1-4* in 2iL- and SL-ESCs. Data are presented as the mean \pm SD. Indicated significances are tested using Student's *t*-test analyses (* $p < 0.05$, ** $p < 0.01$, *** $p < 0.001$). $n = 3$ biological replicates.

C. Sequence LOGOs of TEAD1-4 motifs enriched in ESCs.

D. Hierarchical cluster analysis (HCA) and heatmaps of control and *Tead4*-depleted cells at day 0 and day 6 of the transition. The heatmaps were based on rlog-transformed and DESeq2-normalized expression data. The color key shows the Euclidean distances between samples.

E and F. Volcano plots showing differential gene expression (fold change > 2 ; q -value < 0.05) between control and *Tead4*-depleted cells at day 0 (E) and day 6 (F) of the SL-to-2iL transition.

YAP1 and TAZ act as co-activators of TEAD (Pocaterra et al, 2020). To explore whether *Yap1* and *Taz* are involved in the regulation of TEAD2 in ground-state pluripotency, we have also detected the expression levels of *Taz* and *Yap1* during the interconversion between 2iL- and SL-ESCs, and performed SL-to-2iL transition after knocking *Taz* or *Yap1* down. We found that “*Notably, we observed a correlation between the expression of Taz and Tead2 during the interconversion of 2iL- and SL-ESCs (Fig. EVI3B). Knocking down Taz, but not Yap1, disrupted the cell morphology during SL-to-2iL transitions (Fig. EVI3C, D). These findings suggest the involvement of TAZ in the regulation of TEAD2 in ground-state pluripotency. The specific roles of YAP1 and TAZ in regulating stem cell pluripotency necessitate further investigation. The elucidation of Hippo signaling factors' roles in chromatin structure remains an open question. It is yet to be determined whether TAZ collaborates with TEAD2 to modulate chromatin structure, thereby regulating stem cell ground-state pluripotency.*” We have incorporated this part of data into the **Discussion** section at page 19, line 469 to suggest the potential coordination of TAZ and TEAD2.

Figure EV13

Figure EV13. Supplementary data for the DISCUSSION section.

B. Expression patterns for *Yap1* and *Taz* during the conversion between 2iL-ESCs and SL-ESCs.

C. RT-qPCR testing knockdown efficiencies for *Yap1* and *Taz* during the SL-to-2iL transition. Cells were treated with specific siRNAs every 3 days along with control siRNA. Data are presented as the mean \pm SD. Indicated significances are tested using Student's *t*-test analyses ($*p < 0.05$, $**p < 0.01$, $***p < 0.001$). $n = 3$ biological replicates.

D. Representative images of cells transfected with siNC (negative control) and siRNAs targeting *Yap1* and *Taz*, respectively, during the SL-to-2iL process. Scale bar, 100 μ m.

Minor remarks:

The authors refer to the pluripotency state captured by ESCs maintained in serum/LIF as "confused". We feel the use of this term is not appropriate: the term is colloquial and misleading in the context. Please find an alternative. Possibly the authors are referring to the fact that ESCs are "metastable" or "prone to differentiation" in serum/LIF.

Answer: We apologize for this confusion. We have followed the Reviewer's suggestion and described these states as the Reviewer#2 suggested. The details are as

follows: “Mouse ESCs cultured in serum and leukemia inhibitory factor (serum/LIF; SL) display a metastable state, expressing various lineage-specific genes and being prone to differentiation (Chambers et al, 2007; Evans & Kaufman, 1981; Hayashi et al, 2008). In contrast, ESCs cultured in serum-free medium with LIF and two inhibitors (PD0325901 and CHIR99021) (2i/LIF; 2iL) exhibit a more homogeneous phenotype, resembling the inner cell mass of the preimplantation epiblast and reflecting a “ground-state” pluripotency (Boroviak et al, 2014; Marks et al., 2012; Marks & Stunnenberg, 2014; Ying et al, 2008).” in the first paragraph of the **Introduction** in our new manuscript at page 2, line 37.

In several plots presenting expression dynamics of genes the Y axis is labeled "RNA-seq counts". This unit is unclear. Are the counts in question normalized? (RPKM, RPM.. etc).

Answer: We apologize for this confusion. The Y axis should be labeled as “RNA-seq normalized tag count”. We have also added Fig. 2C into **Fig. EV3A, B** and revised the Y axis in our new manuscript.

Figure EV3. Expression levels of candidate factors during the conversion between 2iL-ESCs and SL-ESCs.

A. Expression values of the candidate genes from RNA-seq data.

B. RT-qPCR analyzing the expression levels of candidate genes from the TEA, CP2, and NR families. These are enriched in Region 2/3 during the transition between 2iL-ESCs and SL-ESCs. Data are presented as the mean \pm SD.

Figs 2C and EV3A show expression changes that seem minimal and strongly variable. The authors should adjust accordingly how these results are discussed in the manuscript. Line 155 The phrasing "the expression of Tead2, Tead4, Tcfcp211, Esrrb and Nr5a2 in region 2/3 was gradually downregulated" is unclear. Are the authors referring to the fact that these genes are located near accessible peaks that are classified as Regions 2/3, and that they are downregulated?

Answer: We apologize for unclear description in our previous version. We have re-organized this part of the results as follows: *“Subsequently, we explored the dynamic expression of TEA, CP2, and NR family transcription factors (Tead2, Tead4, Tcfcp211, Esrrb, and Nr5a2) during the transition between 2iL- and SL-ESCs. Integrating RNA-seq data and RT-qPCR results revealed a general upregulation of these genes during the SL-to-2iL transition and a downregulation during the 2iL-to-SL transition (Fig. EV3A, B)”* in our new manuscript at page 7, line 160.

Line 185: "homogeneous". Do the authors mean "homozygous"?

Answer: Thank the Reviewer#2's for pointing out this error. We have changed the word from *homogeneous* to *homozygous*.

Line 207: It is not clear what is meant by exit of serum specific genes? Possibly downregulation?

Answer: We have changed this sentence from *“These results indicate that TEAD2 is required for the expression of 2i-specific genes and the exit of serum-specific genes during SL-to-2iL transition.”* to *“These findings underscore the crucial role of TEAD2 in activating a sub set of 2i-specific genes during SL-to-2iL transition.”* in our new manuscript at page 10, line 252.

Line 231: The phrasing "20.83% (225/1080) of the downregulated genes were 2i-specific genes after Tead2 knockout" is not understandable.

Answer: We are sorry for this confusion. This sentence has been removed in the revised manuscript, and instead "*We then identified the genes directly bound by TEAD2, with approximately 15.9% (192/1,210) of cluster 5 genes being targeted by TEAD2 (Fig. 4J; Table EV4).*" into the new manuscript at page 12, line 294.

Referee #3:

In this work, Guo et al identified the TF TEAD2 to be important for ESCs culture in 2i. This is an interesting finding in the field of pluripotency. The data are in general technically well performed. However, before publication, there are some points to be better clarified and corrected. In particular, it should be better investigated how TEAD2 binding to chromatin differ between 2i and serum.

Major points

1. The images of Fig. EV1A showed cell morphology changes during 2i-serum transition. However, many, many cells had a differentiation phenotype. Are these cells still all pluripotent? Can the authors show AP staining and compared it to differentiated cells?

Answer: Thank the Reviewer#3 for this question. Mouse ESCs cultured in serum and leukemia inhibitory factor (serum/LIF; SL) display a metastable state, expressing various lineage-specific genes and being prone to differentiation (Chambers et al., 2007; Evans & Kaufman, 1981; Hayashi et al., 2008). Although SL-ESCs exhibit high heterogeneity, the cells still maintain pluripotency (Fig. EV6 B-D).

Figure EV6

Figure EV6. TEAD2 has no effect on the core pluripotency establishment of ESCs.

B. Cellular morphology analysis of 1×10^5 *Tead2*^{-/-} and *Tead2*^{+/-} SL-ESCs grown on a 6-well plate coated with gelatin for three days. Scale bar, 100 μ m.

C. AP-stained wells of *Tead2*^{-/-} and *Tead2*^{+/-} SL-ESCs after 5 days of culture.

D. Western blot analysis of the OCT4 and SOX2 protein levels in wild-type, *Tead2*^{-/-}, and *Tead2*^{+/-} SL-ESCs.

To distinguish from the differentiated cells, we have performed additional morphological comparisons and AP staining during SL-to-2iL and differentiation (cells cultured in medium in the absence of LIF or 2i with LIF). We observed that the clonal morphology during the transition differed markedly from that of differentiation, and the pluripotency of cells completely exits after differentiation (Figs. R6 and **Fig. EV6F**). Meanwhile, in conjunction with **Question 3 from Reviewer 1#**, we further explored whether the cell deformation caused by knockdown/knockout of *Tead2* would induce the cells to exit pluripotency, and found that “*The loss of Tead2 resulted in cell deformation but did not lead the cells to exit pluripotency during SL-to-2iL transition; instead, the cells underwent spontaneous differentiation after removing 2i and LIF (Fig. EV6E). The expression of core pluripotent factors Oct4, Sox2, and Nanog also showed no significant alteration (the change was less than 1.5-fold) in both wild-type and Tead2-knockout cells on day 0 and day 6 of the transition (Fig. EV6F).*” And we have added these new data into our new manuscript at page 9, line 222.

Figure R6

Figure R6. Representative cellular morphology in both wild-type and *Tead2*-knockout cells that were adapted to 2i conditions for 0 and 3 days and the cells that underwent 3 days of differentiation in the absence of LIF or 2i with LIF. Scale bar, 100 μ m.

Figure EV6. TEAD2 has no effect on the core pluripotency establishment of ESCs.

E. Representative images of AP staining of wild-type and *Tead2*-knockout cells that were adapted to 2i conditions for days 0, 3, and 6. Cells were then induced for 5 days of differentiation in medium in the absence of LIF or 2i with LIF.

F. RT-qPCR testing the expression of pluripotent genes in wild-type and *Tead2*-knockout cells on day 0 and day 6 of the transition. Data are presented as the mean \pm SD. Indicated significances are testing using Student's *t*-test analyses ($*p < 0.05$, $**p < 0.01$, $***p < 0.001$). $n = 3$ biological replicates.

2. Fig. EV1B, C, right panels. It should be reported in the corresponding legend what has been quantified in the agarose gel (I guess the upper band).

Answer: We have added the information of the quantified regions in the legends of Fig. EV1B, C: “Quantification of DNA signals of the upper band of the agarose gel using Fiji image analysis software” in our new manuscript.

3. Lane 142-143: “These results suggested that the overlapping accessible peaks between these two processes were essential for the transition between 2i-ESCs and

serum-ESCs.". This conclusion is too strong. The authors have only described chromatin accessibility during 2i-serum conversion and vice versa but not a function for the transition.

Answer: We have followed the Reviewer#3's suggestion and toned down our conclusion. We have changed the sentence from "*These results suggested that the overlapping accessible peaks between these two processes were essential for the transition between 2i-ESCs and serum-ESCs*" to "***These results suggest that the overlapping accessible peaks between these two processes are functionally related to 2iL- and SL-ESCs and begin to change at the initiation stage of the transition, representing a key region for the transition.***" at page 6, line 146 in our new manuscript.

4. Lanes 154-155. The authors "discovered that the expression of Tead2, Tead4, Tcfcp2l1, Esrrb and Nr5a2 in Region 2/3 was gradually downregulated during 2i-to-serum transition..." The authors should also refer to published data since this analysis has been extensively performed in the past. Are these data consistent with previous work? Moreover, although there is a gradual downregulation of mRNA, the authors should show TEAD2 protein levels by WB. Is still expressed? This would be important to better understand the role of TEAD2 in 2i-ESC regulation (see also my comment on the TEAD2-ChIPseq).

Answer: Following the Reviewer#3's suggestion, we have further analyzed the expression of *Tead2*, *Tead4*, *Tcfcp2l1*, *Esrrb* and *Nr5a2* in the published data of 2iL-and SL-ESCs. RNA-seq datasets revealed that all of these factors were highly expressed in 2iL-ESCs relative to SL-ESCs (Fig. R7A-C). Then, "*We measured TEAD2 expression levels in both mRNA and protein in both cell types and observed that TEAD2 expression in 2iL-ESCs was approximately 1.5-fold higher than that in SL-ESCs (Fig. 3A).*" And we have added these new data into our new manuscript at page 8, line 193.

Figure R7

Figure R7. The expression levels of five candidate genes from the published data. Expression values of candidate genes (*Tead2*, *Tead4*, *Nr5a2*, *Esrrb*, *Tcfcp2l1*) in (A) GSE23943, (B) GSE133794, (C) GSE123691.

Figure 3

Figure 3A. RT-qPCR and Western blot examining TEAD2 expression in both 2iL- and SL-ESCs. mRNA expression was tested in triplicate in three independent experiments. Data are presented as the mean \pm SD. p-values were determined using two-sided Student's t-test. Quantification of protein signal was performed using Fiji image analysis software.

5. Fig. 2D. Lanes 158-169. The authors described that *Esrrb* KD has less effects on cell morphology. Moreover, they indicated that *Nr5a2* has already been reported to be critical for 2i-specific enhancers (Atlasi et al. PMID:31036938). However, the data

shown in Atlasi et al. indicated the opposite: a key role for Esrrb whereas Nr5a2 appears to be dispensable. The authors should clarify or correct this point.

Answer: We sincerely apologize for this mistake. Our results revealed that with the exception of *Tead4* knockdown, knockdown of any of the other four factors (*Tead2*, *Esrrb*, *Tcfcp2l1* and *Nr5a2*) diminished self-renewal and proliferation capacity to varying degrees during the SL-to-2iL transition (**Figure 2C-F in our new manuscript**).

We have revised the description of the regulatory role of ESRRB: “*ESRRB and TCFCP2L1 play crucial roles in stabilizing the regulatory network of naïve pluripotency and preventing the loss of self-renewal (Atlasi et al., 2019; Festuccia et al., 2018a; Festuccia et al., 2018b; Hackett & Surani, 2014; Qiu et al., 2015; Zhang et al., 2021)*” in the revised manuscript at page 7, line 180.

Despite NR5A2 seems to be dispensable for SL-to-2iL transition (Atlasi et al., 2019), recent study has discovered that “*NR5A2 can form a regulatory module with ESRRB to assist in the binding of core pluripotency factors at most of their occupied regions, thereby regulating the naïve pluripotency network (Festuccia et al., 2021)*”. We also cited this reference in the revised manuscript at page 8, line 184.

6. Fig EV4A-C. Lanes 172-174. "... knockdown *Tead2* had little effect on both the expression of core pluripotent factors Oct4, Sox2, Nanog... and the spontaneous differentiation of ESCs". Could the authors explain how did they measure "the spontaneous differentiation of ESCs"? Moreover, the AP staining of Fig. EV4C does not show any difference between ESCs and differentiated cells (-2i-LIF, -LIF in serum).

Answer: In the previous manuscript, we treated cells with *Tead2* siRNAs for 48 hours. Then we performed spontaneous differentiation for 3 days in siRNA-treated cells cultured in either 2i medium (-2i-LIF) or serum/ILF medium (-LIF) and then followed AP staining. Since this short duration of siRNA treatment might not fully reveal the phenotype caused by loss of *Tead2*, we treated cells with *Tead2* siRNAs for 72 hours, and then proceeded with the differentiation and AP staining in the new manuscript.

Moreover, we used a more sensitive AP staining kit (Alkaline Phosphatase Staining Kit II, Stemgent, 00-0055) to enhance the detection of differences. Indeed, we found that: *“Tead2 knockdown induced heterogeneity in 2iL-ESCs, exhibiting a morphology similar to that of SL-ESCs (Fig. EV4B). During spontaneous differentiation after removing 2i/LIF or LIF, 2iL-differentiated cells with Tead2 knockdown resembled SL-differentiated cells in morphology (Fig. EV4B). Tead2 knockdown had a minor effect on SL-ESCs (Fig. EV4B). Furthermore, Tead2 knockdown upregulated serum-specific genes and downregulated 2i-specific genes in 2iL-ESCs (Fig. EV4E). In contrast, Tead2 knockdown in SL-ESCs did not induce upregulation of serum-specific genes and had no or minor effect on downregulation of 2i-specific genes (Fig. EV4F). These results suggest that TEAD2 does not participate in regulating the core pluripotency of mESCs but instead stabilizes the ground-state regulatory network of 2iL-ESCs to prevent them from entering a metastable state”*. And we have added these new data into our new manuscript at page 8, line 199.

Figure EV4

Figure EV4. Effect of TEAD2 on the ground-state pluripotency in 2iL-ESCs.

A. RT-qPCR determining the *Tead2* knockdown efficiency in 2iL-ESCs and SL-ESCs. Data are presented as the mean \pm SD. Indicated significances are testing using Student's *t*-test analyses (* $p < 0.05$, ** $p < 0.01$, *** $p < 0.001$). $n = 3$ biological replicates.

B. Representative images of AP staining of 2iL- and SL-ESCs and cells after 3 days of differentiation in complete medium containing 10% serum or in the absence of LIF. Cells were treated with control siRNA or siRNA targeting *Tead2*.

C and D. RT-qPCR testing the expression of pluripotent genes in 2iL-ESCs (C) and SL-ESCs (D). Data are presented as the mean \pm SD. Indicated significances are tested using Student's *t*-test analyses (* $p < 0.05$, ** $p < 0.01$, *** $p < 0.001$). $n = 3$ biological replicates.

E and F. RT-qPCR testing the expression of 2i- and serum-specific genes in 2iL-ESCs (E) and SL-ESCs (F). Data are presented as the mean \pm SD. Indicated significances are tested using Student's *t*-test analyses ($*p < 0.05$, $**p < 0.01$, $***p < 0.001$). n = 3 biological replicates.

7. Fig. 3D, Lanes 205-207. "These results indicate that TEAD2 is required for the expression of 2i-specific genes and the exit of serum-specific genes during serum-to-2i transition." Since many other 2i-genes are not affected, it is more appropriated to conclude that TEAD2 is required for the regulation of "a set" of 2i-specific genes and serum specific genes.

Given the known role of ESRRB in serum-2i transition, can the author comment on differences in gene expression/chromatin binding between ESRRB and TEAD2?

C2 group was not described.

Answer: We have followed the reviewer's suggestion and changed the sentence from "*These results indicate that TEAD2 is required for the expression of 2i-specific genes and the exit of serum-specific genes during SL-to-2iL transition.*" to "*These findings underscore the crucial role of TEAD2 in activating a sub set of 2i-specific genes during SL-to-2iL transition.*" in the new manuscript.

We have also added the description of C2 group: "*Gene expression levels of cluster 1 (C1, 961 genes) and cluster 4 (C4, 1101 genes) remained unchanged, whereas cluster 2 (C2, 662 genes) exhibited a slight downregulation in SL-ESCs but remained unaffected during the transition.*" in our new manuscript.

To investigate the differential roles of ESRRB and TEAD2 in regulating 2iL-ESCs, we analyzed the binding sites between ESRRB and TEAD2, and identified 5,030 common peaks, 20,939 ESRRB-specific peaks and 20,057 TEAD2-specific peaks, respectively (Fig. R8A). But both specific peaks showed similar genomic distribution, predominantly located in the distal intergenic regions (Fig. R8B). Moreover, ESRRB and TEAD2 target a comparable number of 2i-specific genes, 630 and 577, respectively (Fig. R8C). Among these genes, we identified that 235 genes were co-bound by both factors, while 395 target genes were ESRRB-specific and 342

genes were TEAD2-specific (Fig. R8C). These findings indicate that TEAD2 has an important role as ERSSB during SL-to-2iL transition.

Figure R8

A. Venn plots showing the overlap among ESRRB binding peaks on day 3 of the SL-to-2iL transition and TEAD2 binding peaks in 2iL-ESCs.

B. Pie charts showing the genomic distribution of the ESRRB-specific binding peaks and TEAD2-specific binding peaks.

C. Venn plots showing the overlap among ESRRB-specific binding peaks target genes and TEAD2-specific binding peaks target genes.

8. The BIOTIN-TEAD2 ChIP-seq experiment is not clearly described. Was it performed in 2i-ESCs? What is biotin-tag? BirA, BioID? Which was the negative control? Was the biotin-tagged TEAD2 construct overexpressed? It is not clear whether TEAD2 is also expressed in serum-ESCs (see point 4). Is there a difference in TEAD2 binding to chromatin between 2i- and serum-ESCs? It might also help to compare the RNAseq in TEAD2-KD 2i- and serum-ESCs to determine which genes are specifically or commonly regulated by TEAD2 in 2i- and serum-ESCs.

Fig. EV7C (scatter plot of RNAseq between KO Tead2-D6 vs WT-D6) is not cited in the text. Does this experiment belong to this Figure supporting the biotin-ChIP?

Answer: In the previous manuscript, we overexpressed biotin-tagged TEAD2 in 2iL-ESCs and performed BIOTIN-tagged TEAD2-ChIP-seq experiments. This could not reveal the difference in TEAD2 chromatin binding between 2iL- and SL-ESCs. In response, we generated stable 2iL- and SL-ESC lines with endogenous expression of

biotin-tagged TEAD2 through CRISPR/Cas9. We have added the detailed method for *“Generation of Tead2-FLAG-AviTag knock-in 2iL- and SL-ESC lines”* in the **Materials and Methods** section in the revised manuscript (please see in page 6 in this response letter).

Then we performed BIOTIN ChIP-seq experiments in both 2iL- and SL-ESCs and replaced the entire **Figure 4** in this revised manuscript. We have added the following sentences into the new manuscript *“24,994 and 5,837 peaks in 2iL-ESCs and SL-ESCs, respectively. Motif enrichment analysis indicated significant enrichment of TEAD2 binding motifs in both 2iL- and SL-ESCs (Fig. 4C). Notably, 10,315 specific peaks were identified in 2iL-ESCs, while only 47 were specific to SL-ESCs, suggesting a potential regulatory role for TEAD2 in 2iL-ESCs (Fig. 4D, E). These 2i-specific peaks predominantly enriched in intergenic regions with more open chromatin regions (Fig. 4F, G). About 43.13% (4,489/10,315) of TEAD2 peaks localize to either promoters or enhancers (Fig. 4H). This observation implies the potential involvement of TEAD2 in gene expression regulation.*

To discern whether TEAD2 functions as an activator or repressor, we conducted the analysis of the binding relationships between TEAD2 and active/repressive histone marks in 2iL- and SL-ESCs. Utilizing published ChIP-seq data for histone marks H3K36me3, H3K4me1, H3K4me3, H3K27ac, H3K27me3, and H3K9me3 in 2iL- and SL-ESCs (Aljazi et al, 2020; Joshi et al, 2015; Marks et al., 2012), our investigation revealed that 2i-specific TEAD2 sites predominantly marked by active histone marks, notably H3K27ac, H3K4me1, and H3K4me3 (Fig. 4I). Furthermore, these active histone marks exhibited a more pronounced enrichment at these sites in 2iL-ESCs compared to SL-ESCs (Fig. 4I). We then identified the genes directly bound by TEAD2, with approximately 15.9% (192/1,210) of cluster 5 genes being targeted by TEAD2 (Fig. 4J; Table EV4). Notably, these 192 genes exhibited reduced expression following Tead2 loss (Fig. 4K). GO analysis indicated their involvement in glycolipid metabolic processes and lipid catabolic processes (Fig. 4L), exemplified by B4galt6 (Fig. 4M). These findings suggest that TEAD2 binds to both promoters and enhancers of 2i-specific genes, directly influencing their expression.”

Figure 4

Figure 4. TEAD2 binds to the active chromatin regions of 2i-specific genes.

- A.** Strategy for generating of *Tead2*-FLAG-AviTag knock-in cell lines in 2iL- and SL-ESCs.
- B.** Western blot analysis for BIOTIN and V5 with cell lysates from *Tead2*-FLAG-AviTag-knock-in 2iL- and SL-ESC lines. β-ACTIN was used as a loading control.
- C.** Motif-enrichment analysis of BIOTIN-binding sites in 2iL- and SL-ESCs.
- D.** Heatmap showing the comparison of TEAD2 binding sites between 2iL- and SL-ESCs.
- E.** Number of 2iL- and SL-ESCs-specific TEAD2 binding peaks.
- F.** Pie charts showing the genomic distribution of 2i-specific TEAD2 peaks.

- G.** Heatmaps of sequence read density for ATAC-seq in 2i-specific TEAD2 binding peaks.
- H.** Bar plot showing the number of 2i-specific TEAD2 peaks that overlap with both promoters and enhancers.
- I.** Tag-density pileup showing H3K4me1, H3K27ac, H3K4me3, H3K27me3, H3K9me3, and H3K36me3 ChIP-seq signals at the 2i-specific TEAD2 binding sites in both 2iL- and SL-ESCs.
- J.** Venn plots showing the overlap among 2i-specific TEAD2 target genes and cluster 5 genes.
- K.** Boxplots showing expression level of overlapping genes in (J) between wild-type and *Tea2*-knockout cells at day 6 of the transition.
- L.** GO categories of the overlapping genes shown in (J).

Fig. EV7C does not belong to the results supporting the biotin-ChIP, and has been deleted in the new manuscript.

9. Fig. EV8: Change of A/B compartments in *Tea2*-KD ESC+2i. Are domains switching from A to B bound by TEAD2?

Answer: We found that 33.9% (392/1155) of TEAD2 bound domains that switch from A compartment to B compartment and 45% (791/1759) of TEAD2 bound domains that switch from B compartment to A compartment. We demonstrated the association between the B-to-A compartment switching and the abnormal activation of serum-specific genes after *Tea2* knockout.

10. Fig. 5H. The loss of interactions around *B4galt6* gene upon TEAD2-KO is not very clear. It seems more that the interactions are changed instead to be lost. Can the authors show in the image the bed file for TEAD2 called peaks? Are the other anchor points from *B4galt6* gene known enhancers?

Answer: We have reanalyzed the changes of the interaction on *B4galt6* gene after *Tea2* knockout by using the new TEAD2 binding sites as the anchor in 2iL-ESCs. And we added TEAD2-peak.bed and enhancer bed files into the dataset. The new result showed that *Tea2* knockout led to the loss of TEAD2-mediated EP interactions on 2i-specific gene-*B4galt6* (Fig. 5H).

Figure 5

Figure 5H. Representative genomic locus showing the binding of TEAD2 and H3K27ac, and decreased chromatin interactions after *Tead2*-knockout. *B4galt6* promoter is highlighted with red-shaded rectangles, and its associated enhancers are highlighted with green-shaded rectangles.

11. Fig. 6E TEAD2 dimerization. While I consider this result interesting, there is nothing new relative to previous work cite by the authors (Lee et al, 2016). Moreover, there is no information for the silver staining of Fig. EV9 showing the (minimal) loss of TEAD2 dimer in the presence of DTT. The samples were run on an SDS-PAGE, which is not a native gel. How is it possible to observe TEAD2 dimer in such denaturing conditions? Moreover, the conclusions should be toned down since no data are provided showing that TEAD2 dimerization might mediate EP interactions. Maybe it is better to delete this part from the manuscript since it is out of the scope of the paper and not conclusive.

Answer: We are appreciated for the reviewer’s comments. We performed native gel electrophoresis on the samples using the Native PAGE Gel Protein Electrophoresis Kit (RealTimes, RTD6135). Although we noticed a decrease in band intensity after DTT treatment, we lacked definitive proof for this band corresponding to TEAD2-dimer. To ensure the validity of the results, we removed this part of the data from the revised manuscript by following the Reviewer’s suggestion. We have also moved the co-IP results confirming the self-interaction of TEAD2 into the **Discussion**

section. The details are as follows:

“The DNA binding activity of TEAD transcription factors is localized within their N-terminal domains (TEAD-DBD) (Anbanandam et al., 2006; Bürglin, 1991). The TEAD DBD, with a truncated L1 loop, can form homodimers through domain swapping, thereby regulating the DNA selectivity of TEAD proteins (Lee et al, 2016). To ascertain whether TEAD2 can form dimers in vivo, we generated and expressed both FLAG-tagged and HA-tagged TEAD2 plasmids in cells. Subsequent FLAG co-IP experiments, after elution with high salt concentration, confirmed the interaction between HA-tagged TEAD2 and FLAG-tagged TEAD2 (Fig. EV13A). Therefore, we hypothesized that TEAD2 mediates EP interactions through its dimerization, analogous to the mechanisms observed for CTCF and YY1.” in the revised manuscript at page 18, line 452.

12. Fig. 6A-C. The results indicated that upon TEAD2-KD in serum-to-2i transition SMC1 occupancy at enhancer is decreased whereas YY1 occupancy is not affected. The authors concluded that "TEAD2 might mediate EP interactions by functioning together with SMC1 to regulate the expression of 2i-specific genes rather than depending of YY1 and CTCF." I do not see any data that exclude the role of YY1. The lack of decrease in YY1 binding to chromatin upon KD of TEAD2, which is DNA sequence dependent, only shows that YY1 does not depend on TEAD2 and not the contrary.

Answer: We thank the reviewer for pointing out this error. We have changed the sentence from *“These data suggested that TEAD2 might mediate EP interactions by functioning together with SMC1 to regulate the expression of 2i-specific genes rather than depending of YY1 and CTCF.”* to ***“These findings suggest that the reduction of TEAD2-mediated EP interactions is not attributed to changes in YY1 binding at the anchor regions but may function in conjunction with SMC1”*** at page 15, line 369 in our new manuscript.

13. Figure 6. How big was the CRISPR deletion in the promoter region of B4galt6

gene? Does it include the whole promoter? Is this deletion affecting H3K27ac at the contact sites surrounding B4galt6 gene?

Answer: In the previous manuscript, we deleted a 64 bp sequence with the TEAD2 motifs from the *B4galt6* promoter region. To prevent the downregulation of *B4galt6* expression by excessive promoter deletion, and to address question 14 from the Reviewer#1, we constructed two mutant 2iL-ESC line with base substitution within TEAD2 motifs in the *B4galt6* promoter region (**Fig. 6E**). We have added the detailed method of “*Generation 2iL-ESC lines with mutation of TEAD2 motifs in the B4galt6 promoter region*” in the **Materials and Methods** section of this revised manuscript (please see in page 21 in this response letter).

RT-qPCR and QHR-4C experiments were then performed. We have added the following sentences “*the loss of TEAD2 motifs at the gene promoter in these two mutant clones had no effect on the expression of Tead2 (Fig. 6F) but resulted in lower expression of B4galt6 (Fig. 6G). To further demonstrate that the downregulation of B4galt6 gene expression in the mutant clones resulted from the attenuation of TEAD2-mediated EP interactions, quantitative high-resolution chromosome conformation capture copy (QHR-4C) experiments were performed in wild-type and two mutant 2iL-ESCs, wild-type and Tead2-knockout cells at day 6 of transition. The results showed that, similar to Tead2 knockout, the frequency of EP interactions at the B4galt6 gene locus with TEAD2 binding peaks was significantly reduced in the two mutant clones compared to wild-type 2iL-ESCs (Figs. 6H and EVIIE). Tead2 knockout had no effect on the levels of H3K4me1 and H3K27ac on these EP interactions at day 6 of SL-to-2iL transition (Fig. 6H). Additionally, there was no change in H3K27ac enrichment at the B4galt6 locus in either of the mutant clones (Fig. EVIIF). These results collectively demonstrate that TEAD2 contributes to EP interactions for 2i-specific genes.*” into the new manuscript at page 15, line 386.

Figure 6

Figure 6. Mediation of EP interactions by TEAD2 may involve Cohesin but not structural proteins such as YY1 and CTCF.

E. Strategy of generating 2iL-ESC lines with TEAD2 motif mutations at the *B4galt6* promoter region.

F and G. RT-qPCR analysis testing the expression of *Tead2* (F) and *B4galt6* (G) in wild-type 2iL-ESCs and two homozygous clones with base alterations in both TEAD2 binding motifs. Data are presented as the mean \pm SD. *p*-values were determined using the two-sided Student's *t*-test.

H. 4C tracks showing the interactions between the promoters and enhancers of *B4galt6* in wild-type and two mutant 2iL-ESCs, wild-type and *Tead2*-knockout cells at day 6 of the transition. The anchor region from QHR-4C is indicated.

Figure EV11

Figure EV11. Supplementary data of the QHC-4C experiments.

B. Restriction enzyme digestion strategy for identifying mutant clones.

C. Genomic PCR and enzyme digestion to verify corrected clones.

D. Sanger sequencing testing the region containing two TEAD2 motifs in wild-type and two homozygous mutant clones of 2iL-ESCs.

E. Barplot showing the normalized interaction frequency between the promoters and enhancers of *B4galt6* in wild-type and two mutant 2iL-ESCs, wild-type and *Tead2*-knockout cells at day 6 of the transition.

F. Genomic views of enrichment for H3K27ac at the *B4galt6* gene in wild-type and two homozygous mutant clones of 2iL-ESCs.

14. Discussion, lanes 342-343 "The transition between 2i-ESCs and serum-ESCs can simulate the changes of pluripotency in early embryonic development in vitro". I am not sure that this is the case. Do ESC+serum refer to a specific developmental stage relative to ESC+2i? I think that the comparison between 2i-ESC and serum-ESC only serves to identify factors implicated in ground-state pluripotency.

Answer: We appreciate these helpful comments and suggestions. 2iL- and SL-ESCs are represent two states of “naïve” pluripotency and share functional properties (Hackett & Surani, 2014; Nichols & Smith, 2009). 2iL-ESCs display a more homogeneous phenotype that resembles the inner cell mass of the preimplantation epiblast and better reflect the “ground-state” pluripotency (Boroviak et al., 2014; Marks et al., 2012; Marks & Stunnenberg, 2014; Ying et al., 2008). We have changed the sentence from “*The transition between 2i-ESCs and serum-ESCs can simulate the changes of pluripotency in early embryonic development in vitro*” to “***Importantly, the transition between 2iL- and SL-ESCs, achieved by altering the culture medium, provides a valuable system for investigating factors involved in ground-state pluripotency and studying gene regulation mechanisms***” at page 2, line 50 in our new manuscript.

15. Lanes 377-380. Conclusion sentence. "Therefore, it is of great significance to use chemical small molecules with high selectivity targeting TEAD2 to modulate stem cell fate determination instead of using genes in the future". This sentence is very unclear. Maybe the authors should conclude their work with something more realistic and relevant to the results of their manuscript.

Answer: According to the reviewer’s suggestions, we have rewritten the last paragraph of the **Discussion** section: “*In summary, we have identified a novel ancillary factor, TEAD2, that initiates ground-state pluripotency by mediating chromatin looping. This study contributes to our understanding of the molecular mechanisms underlying stem cell fate determination and unveils a previously unrecognized molecular function of TEAD2 in higher-order chromatin structure.*” in our new manuscript at page 19, line 478.

16. The authors referred very often to Atlasi et al. (PMID:31036938). However, other recent works have started to find factors required for ground-state or 2i-ESCs but none of them has been considered.

Answer: In the revised manuscript, we have added more references related to the

regulation of ground state pluripotency as follows.

1. Atlasi Y, Jafarnejad SM, Gkogkas CG, Vermeulen M, Sonenberg N, Stunnenberg HG (2020) The translational landscape of ground state pluripotency. *Nat Commun* 11: 1617
2. Peng T, Zhai Y, Atlasi Y, Ter Huurne M, Marks H, Stunnenberg HG, Megchelenbrink W (2020) STARR-seq identifies active, chromatin-masked, and dormant enhancers in pluripotent mouse embryonic stem cells. *Genome Biol* 21: 243
3. Ter Huurne M, Chappell J, Dalton S, Stunnenberg HG (2017) Distinct cell-cycle control in two different states of mouse pluripotency. *Cell Stem Cell* 21: 449-455.e444
4. van Mierlo G, Dirks RAM, De Clerck L, Brinkman AB, Huth M, Kloet SL, Saksouk N, Kroeze LI, Willems S, Farlik M et al (2019) Integrative proteomic profiling reveals PRC2-dependent epigenetic crosstalk maintains ground-state pluripotency. *Cell Stem Cell* 24: 123-137.e128
5. Festuccia N, Halbritter F, Corsinotti A, Gagliardi A, Colby D, Tomlinson SR, Chambers I (2018a) Esrrb extinction triggers dismantling of naïve pluripotency and marks commitment to differentiation. *Embo J* 37: e95476
6. Festuccia N, Owens N, Chervova A, Dubois A, Navarro P (2021) The combined action of Esrrb and Nr5a2 is essential for murine naïve pluripotency. *Development* 148: dev199604
7. Festuccia N, Owens N, Navarro P (2018b) Esrrb, an estrogen-related receptor involved in early development, pluripotency, and reprogramming. *FEBS Lett* 592: 852-877
8. Zhang Y, Ding H, Wang X, Wang X, Wan S, Xu A, Gan R, Ye SD (2021) MK2 promotes Tfcp2l1 degradation via β -TrCP ubiquitin ligase to regulate mouse embryonic stem cell self-renewal. *Cell Rep* 37: 109949
9. Qiu D, Ye S, Ruiz B, Zhou X, Liu D, Zhang Q, Ying QL (2015) Klf2 and Tfcp2l1, Two Wnt/ β -catenin targets, act synergistically to induce and maintain naive pluripotency. *Stem Cell Rep* 5: 314-322

Minor points

1. Legend of Fig. 1 is missing. Instead, there is the legend of Fig. 6.

Answer: We apologize for this mistake. We have added the legend for Figure 1 in our new version. Detailed are shown below:

Figure 1. Dynamics of chromatin accessibility during the interconversion between SL-ESCs and 2iL-ESCs.

A and B. Chromatin loci arranged into groups according to time and status of being closed or opened, closed to open (CO) or open-to-closed (OC), or permanently open (PO) during the transition from SL-to-2iL (A) and 2iL-to-SL (B). Representative genes are noted for each subgroup on the right.

C. Number of peaks defined in CO/OC and PO for (A and B).

D. Venn diagrams of CO/OC and PO peaks during interconversion between SL-ESCs and 2iL-ESCs.

E. Statistics of the number of genes that were switched at different time points of interconversion between SL-ESCs and 2iL-ESCs on the loci of Region 1/4 and Region 2/3.

F. GO analysis of 481 genes in Region 1/4-CO1/OC1 and CO2/OC2, and 766 genes in Region 2/3-OC1/CO1 and OC2/CO2 in (E).

G and H. Representative loci of *Mmp2* and *B4galt6* within Region 1/4 (G) and Region 2/3 (H) defined by ATAC-seq during the transition between SL-ESCs and 2iL-ESCs, respectively (left). Expression values of *Mmp2* and *B4galt6* from RNA-seq data (right).

2. lane 41. "...ESC+serum....are postulated to represent the confused pluripotency". The term "confused" for pluripotency is very strange. Could the author find another term (advanced, primed, etc.)?

Answer: We thank the Reviewer for this suggestion. We have revised the sentence as "*Mouse ESCs cultured in serum and leukemia inhibitory factor (serum/LIF; SL) display a metastable state, expressing various lineage-specific genes and being prone to differentiation (Chambers et al., 2007; Evans & Kaufman, 1981; Hayashi et al., 2008).*" in the first paragraph of the **INTRODUCTION** in our new manuscript.

3. Lane 79. "ATAC-seq, BL-Hi-C, RNA-seq and ChIP-seq experiments followed by high throughput sequencing,..". "high throughput sequencing" is redundant since all

the listed methods imply this.

Answer: We have removed "high throughput sequencing" from our new manuscript.

4. Fig. EV3. In the images, there is no SD and P values described in the corresponding Figure legend.

Answer: The image displays the mean \pm SD, which are visible upon magnification. we have removed the description of P value in this figure legend. The details are as follows: "**B.** *RT-qPCR analyzing the expression levels of candidate genes from the TEA, CP2, and NR families. These are enriched in Region 2/3 during the transition between 2iL-ESCs and SL-ESCs. Data are presented as the mean \pm SD.*"

5. BL-Hi-C should be introduced with the full name (Bridge Linker-Hi-C)

Answer: We have added the full name for BL-Hi-C in the new manuscript as "we conducted **Bridge Linker-Hi-C (BL-Hi-C)** experiments ·····" at page 12, line 307.

References

- Aljazi MB, Gao Y, Wu Y, Mias GI, He J (2020) Cell signaling coordinates global PRC2 recruitment and developmental gene expression in murine embryonic stem cells. *iScience* 23: 101646
- Anbanandam A, Albarado DC, Nguyen CT, Halder G, Gao X, Veeraraghavan S (2006) Insights into transcription enhancer factor 1 (TEF-1) activity from the solution structure of the TEA domain. *Proc Natl Acad Sci U S A* 103: 17225-17230
- Atlasi Y, Megchelenbrink W, Peng T, Habibi E, Joshi O, Wang SY, Wang C, Logie C, Poser I, Marks H *et al* (2019) Epigenetic modulation of a hardwired 3D chromatin landscape in two naive states of pluripotency. *Nat Cell Biol* 21: 568-578
- Boroviak T, Loos R, Bertone P, Smith A, Nichols J (2014) The ability of inner-cell-mass cells to self-renew as embryonic stem cells is acquired following epiblast specification. *Nat Cell Biol* 16: 516-528
- Bürglin TR (1991) The TEA domain: a novel, highly conserved DNA-binding motif. *Cell* 66: 11-12
- Chambers I, Silva J, Colby D, Nichols J, Nijmeijer B, Robertson M, Vrana J, Jones K, Grotewold L, Smith A (2007) Nanog safeguards pluripotency and mediates germline development. *Nature* 450: 1230-1234
- Currey L, Thor S, Piper M (2021) TEAD family transcription factors in development and disease.

Development 148: dev196675

Evans MJ, Kaufman MH (1981) Establishment in culture of pluripotential cells from mouse embryos. *Nature* 292: 154-156

Fairhead M, Howarth M (2015) Site-specific biotinylation of purified proteins using BirA. *Methods Mol Biol* 1266: 171-184

Festuccia N, Halbritter F, Corsinotti A, Gagliardi A, Colby D, Tomlinson SR, Chambers I (2018a) Esrrb extinction triggers dismantling of naïve pluripotency and marks commitment to differentiation. *Embo J* 37: e95476

Festuccia N, Owens N, Chervova A, Dubois A, Navarro P (2021) The combined action of Esrrb and Nr5a2 is essential for murine naïve pluripotency. *Development* 148: dev199604

Festuccia N, Owens N, Navarro P (2018b) Esrrb, an estrogen-related receptor involved in early development, pluripotency, and reprogramming. *FEBS Lett* 592: 852-877

Hackett JA, Surani MA (2014) Regulatory principles of pluripotency: from the ground state up. *Cell Stem Cell* 15: 416-430

Hayashi K, de Sousa Lopes SMC, Tang F, Lao K, Surani MA (2008) Dynamic equilibrium and heterogeneity of mouse pluripotent stem cells with distinct functional and epigenetic states. *Cell Stem Cell* 3: 391-401

Home P, Saha B, Ray S, Dutta D, Gunewardena S, Yoo B, Pal A, Vivian JL, Larson M, Petroff M *et al* (2012) Altered subcellular localization of transcription factor TEAD4 regulates first mammalian cell lineage commitment. *Proc Natl Acad Sci U S A* 109: 7362-7367

Joshi O, Wang SY, Kuznetsova T, Atlasi Y, Peng T, Fabre PJ, Habibi E, Shaik J, Saeed S, Handoko L *et al* (2015) Dynamic reorganization of extremely long-range promoter-promoter interactions between two states of pluripotency. *Cell Stem Cell* 17: 748-757

Lee DS, Vornrhein C, Albarado D, Raman CS, Veeraraghavan S (2016) A potential structural switch for regulating dna-binding by TEAD transcription factors. *J Mol Biol* 428: 2557-2568

Lian I, Kim J, Okazawa H, Zhao J, Zhao B, Yu J, Chinnaiyan A, Israel MA, Goldstein LS, Abujarour R *et al* (2010) The role of YAP transcription coactivator in regulating stem cell self-renewal and differentiation. *Genes Dev* 24: 1106-1118

Marks H, Kalkan T, Menafra R, Denissov S, Jones K, Hofemeister H, Nichols J, Kranz A, Stewart AF, Smith A *et al* (2012) The transcriptional and epigenomic foundations of ground state pluripotency. *Cell* 149: 590-604

Marks H, Stunnenberg HG (2014) Transcription regulation and chromatin structure in the pluripotent ground state. *Biochim Biophys Acta* 1839: 129-137

Nichols J, Smith A (2009) Naive and primed pluripotent states. *Cell Stem Cell* 4: 487-492

Pocaterra A, Romani P, Dupont S (2020) YAP/TAZ functions and their regulation at a glance. *J Cell Sci* 133: jcs230425

Qiu D, Ye S, Ruiz B, Zhou X, Liu D, Zhang Q, Ying QL (2015) Klf2 and Tfc2l1, Two Wnt/ β -Catenin targets, act synergistically to induce and maintain naive pluripotency. *Stem Cell Rep* 5: 314-322

Ying QL, Wray J, Nichols J, Battle-Morera L, Doble B, Woodgett J, Cohen P, Smith A (2008) The ground state of embryonic stem cell self-renewal. *Nature* 453: 519-523

Zhang Y, Ding H, Wang X, Wang X, Wan S, Xu A, Gan R, Ye SD (2021) MK2 promotes Tfc2l1 degradation via β -TrCP ubiquitin ligase to regulate mouse embryonic stem cell self-renewal. *Cell Rep* 37: 109949

Dear Dr Hongjie Yao,

Thank you for submitting your revised manuscript (EMBOJ-2023-114807R) to The EMBO Journal. Your amended study was sent back to the three referees for their scientific re-evaluation, and we have received detailed comments from all of them, which I enclose below.

As you will see, the experts state that the work has been substantially improved by the revisions and they are now in favour of publication, pending minor revision.

Thus, we are pleased to inform you that your manuscript has been accepted in principle for publication in The EMBO Journal.

Please consider the remaining minor comment of referees #1 and #2 regarding sample annotation carefully and amend the manuscript figures and text accordingly.

Also, we now need you to take care of a number of issues related to formatting and data presentation as detailed below, which should be addressed at re-submission.

Please contact me at any time if you have additional questions related to below points.

Thank you for giving us the chance to consider your manuscript for The EMBO Journal. I look forward to your final revision.

Again, please contact me at any time if you need any help or have further questions.

Best regards,

Daniel Klimmeck

>> Please add up to five keywords for your study.

>> Author Contributions: Please remove the author contributions information from the manuscript text. Note that CRediT has replaced the traditional author contributions section as of now because it offers a systematic machine-readable author contributions format that allows for more effective research assessment. and use the free text boxes beneath each contributing author's name to add specific details on the author's contribution.

More information is available in our guide to authors.
<https://www.embopress.org/page/journal/14602075/authorguide>

>> Adjust the title of the Competing Interests' section to 'Disclosure and Competing Interests Statement'.

>> Figures: main figures should be uploaded as individual, high resolution figure files. Figure legends should be added to the manuscript text, after References. Up to five of the EV figures can also be uploaded as individual figures and their legends should be in the manuscript, after the main figure legends. All other figures should be compiled in an appendix: a PDF with a ToC, the figures with their legends, and the nomenclature should be changed to "Appendix Figure S1" etc. .

>> Please ad Panel G to Figure EV6.

>> Dataset EV Legends: Tables EV1-5 should be uploaded as individual files and renamed "Dataset EV1-5". Tables EV6-11 should be uploaded as individual files and renumbered Tables EV1-6.

>> Data Availability Section: please remove referee token and ensure privacy is released.

>> Consider additional changes and comments from our production team as indicated below:

- Data availability section:

1. Please note that the specific URLs for GSE226316 and CRA009963 datasets are not provided in the data availability statement.
2. Please note that reviewer access codes for GSE226316 and CRA009963 datasets are not provided in the data availability statement.

>>> URLs added, data now accessible AD 25.1.24

- Figure legends:

1. Please define the annotated p values */*** in the legend of figures 2e; 3a; 5b; 6f-g; EV 6a; EV 9h-i; as appropriate.
2. Please indicate the statistical test used for data analysis in the legends of figures 1f; 2a-b, e; 3e; 4c, k-l; 5b, d.
3. Please note that in figure 2c; EV 1b-c; EV 4a, c-d; EV 6f; EV 8a; EV 11a; EV 12b; EV 13b; there is a mismatch between the annotated p values in the figure legend and the annotated p values in the figure file that should be corrected.
4. Please note that for the figures 2a-b, p-values and statistical tests are indicated in the legends. However, comparison for the same, "" has not been represented in the figures. Please rectify this in the figures or legends as applicable.
5. Please note that the box plots need to be defined in terms of minima, maxima, centre, bounds of box and whiskers, and percentile in the legends of figures 4k; 5b, d; EV 9b; EV 10a-b.
6. Please note that information related to n is missing in the legends of figures 4k; 5b, d; EV 1b-c; EV 9b; EV 9h-i; EV 10a-b; EV 11a.
7. Please note that the scale bar need to be defined for figure EV 4b.

Please use the link below to submit your revision:

Referee #1:

The authors have made great efforts to address all of my comments, and I am pleased with the revised manuscript.

There are two very minor issue that could be added in the final version:

Comment-2: The authors replied: "Answer: We have followed the reviewer's suggestion and used total ATAC-seq peaks as the background. However, the results still contain a high enrichment of CTCF sites."

Maybe I missed it, but where is the new data supporting the motif enrichment analysis using total ATAC-seq as background? Fig R1 shows an average plot for CTCF binding but there is no enrichment analysis/ p-value. Perhaps it's worth adding this analysis as a supplementary figure or at least mention the enrichment p-value in the main text.

Comment-3: The authors have performed AP-staining to assess the effect of Tead2 siRNA in ESCs. However, Fig-2E shows the number of cells. Are these total number of cells or AP-positive colonies? The idea of AP-staining is to quantify the percentage of 'undifferentiated colonies'. Perhaps the authors can, at least, refer to the morphology of cells based on AP staining in the text.

Referee #2:

The authors have now performed experiments that strengthen significantly the manuscript, and support their conclusions. The function of Tead4 in regulating expression of genes specifically up/down-regulated in ESCs cultured in 2i/LIF is distinguished from any global effect in supporting pluripotency, the long terms effects of Tead2 loss of function is analysed, and well-controlled studies included to address the role of this transcription factor in controlling 3D genome organisation.

A few points remain to be addressed:

Despite morphological and gene-expression differences, Tead2 KO ESCs self-renew in 2i/LIF: consider eliminating from abstract and main text the wording "indispensable" and "essential" when referring to the role of Tead2 in supporting ground state

pluripotency.(Line 32 and 425).

Fig 3E: The authors claim that genes in cluster 4 remain unchanged. This is surprising, and possibly due to a mistake in figure labels? Genes in cluster 4 seem to be the most affected by the loss of Tead2 among those presented in the heatmap.

In several figures throughout the manuscript the authors picked a small panel of 2i/LIF or serum specific genes to exemplify the effects of Tead2 loss or gain of function. An explanation of the rationale behind their choice is scattered in different paragraphs, and remains somehow unclear in the current form of the manuscript. Why were these serum and 2i/LIF specific genes selected? More importantly, boxplots showing the expression changes of all serum and 2i/LIF specific genes should be consistently presented.

The authors suggest that Tead1 may bind to different regions compared to Tead2 in ESCs. In support, they claim that Tead1/3 motifs are not enriched at 2i-specific accessible regions. However, are the motifs recognised by these TFs sufficiently different to allow for such distinctions? Figure EV12C does not permit to appreciate the similarity between the respective motifs. The authors should reverse-complement the Tead2-4 motifs shown in the figure, specify how they are derived, and discuss in better details what could drive any difference in genome occupancy and function between Tead1 and Tead2.

Referee #3:

This revised manuscript is greatly improved, and the authors have reasonably addressed all my previous comments.

Point-by-point Response to Reviewers

Referee #1:

The authors have made great efforts to address all of my comments, and I am pleased with the revised manuscript.

There are two very minor issue that could be added in the final version:

Comment-2: The authors replied: "Answer: We have followed the reviewer's suggestion and used total ATAC-seq peaks as the background. However, the results still contain a high enrichment of CTCF sites."

Maybe I missed it, but where is the new data supporting the motif enrichment analysis using total ATAC-seq as background? Fig R1 shows an average plot for CTCF binding but there is no enrichment analysis/ p-value. Perhaps it's worth adding this analysis as a supplementary figure or at least mention the enrichment p-value in the main text.

Answer: We apologize for omitting this result. We have included these results into the manuscript as below: Using total ATAC-seq peaks as the background, we found that CTCF still exhibited a high enrichment at Regions 1-OC1, 3-OC5, 4-CO1, and PO during the transition (Fig. R1).

Fig. R1. CTCF motifs are significantly enriched in CO/OC/PO categories of ATAC-seq peaks during the transition from SL-to-2iL (left) and 2iL-to-SL (right). The motifs for CTCF are indicated on the right of the heatmap. * $p < 0.05$. P-value was calculated by hypergeometric enrichment calculations from Homer software.

We have added p -value for Figure R1 (response letter 1) to show that the differences of CTCF enrichment between 2iL- and SL-ESCs were significant across different ATAC peak regions.

Figure R1 (response letter 1). Tag density pileup for the CTCF ChIP-seq signals at PO, CO1, OC5 and OC1 sites during interconversion between SL-ESCs and 2iL-ESCs. *** $p < 0.001$. P -value were calculated by Mann-Whitney U test.

Meanwhile, we marked the enrichment of TF motifs with $p < 1e-30$ by the “*” symbol for Figure 2A and 2B in our revised manuscript (Fig. 2A, B).

Fig. 2A, B. TF motifs are significantly enriched in CO/OC/PO categories of ATAC-seq peaks during the transition from SL-to-2iL (A) and 2iL-to-SL (B). The motifs for TFs are indicated on the right of the heatmap. $*p < 1e-30$. *P*-value was calculated by hypergeometric enrichment calculations from Homer software.

Comment-3: The authors have performed AP-staining to assess the effect of Tead2 siRNA in ESCs. However, Fig-2E shows the number of cells. Are these total number of cells or AP-positive colonies? The idea of AP-staining is to quantify the percentage of 'undifferentiated colonies'. Perhaps the authors can, at least, refer to the morphology of cells based on AP staining in the text.

Answer: Fig-2E is a statistical analysis of the total cell number at D0 and D3 of SL-to-2iL transition after knocking each candidate factor down, demonstrating the impact of knockdown of TFs on cell self-renewal capacity. In response to the reviewer's comments, we calculated the percentage of clones with abnormal morphology shown by AP staining at D3 of conversion and incorporated the findings into the manuscript: *“Our findings revealed that knocking down either Tead2 (siTead2) or Nr5a2 (siNr5a2) impeded domed colony formation during SL-to-2iL transition, with Tead2 knockdown*

exhibiting a more pronounced effect than *Nr5a2* knockdown (Fig. 2D-F). Conversely, knockdown of the other three factors (*siTead4*, *siEsrrb*, and *siTcfcp2l1*) had minimal effects on cell morphology (Fig. 2D-F).” in line 174-179 in our new manuscript.

Figure 2

Fig. 2F. The percentage of clones with abnormal morphology on day 3 of SL-to-2iL transition in (E). Data are shown as mean \pm SD of three independent fields of view.

Referee #2:

The authors have now performed experiments that strengthen significantly the manuscript, and support their conclusions. The function of Tead4 in regulating expression of genes specifically up/down-regulated in ESCs cultured in 2i/LIF is distinguished from any global effect in supporting pluripotency, the long terms effects of Tead2 loss of function is analysed, and well-controlled studies included to address the role of this transcription factor in controlling 3D genome organisation.

A few points remain to be addressed:

Despite morphological and gene-expression differences, Tead2 KO ESCs self-renew in 2i/LIF: consider eliminating from abstract and main text the wording "indispensable" and "essential" when referring to the role of Tead2 in supporting ground state pluripotency. (Line 32 and 425).

Answer: We appreciate these suggestions. We have replaced “*indispensable*” with “*prominent*”, “*essential*” with “*important*” in lines 31 and 421 in our new manuscript, respectively.

Fig 3E: The authors claim that genes in cluster 4 remain unchanged. This is surprising, and possibly due to a mistake in figure labels? Genes in cluster 4 seem to be the most affected by the loss of Tead2 among those presented in the heatmap.

Answer: We apologize for this error, and we have revised the expression pattern of the C4 genes as follows: *“Gene expression levels of cluster 1 (C1, 961 genes) were unchanged; cluster 2 (C2, 662 genes) exhibited a slight downregulation in SL-ESCs but were not affected during the transition; and cluster 4 (C4, 1101 genes) was normally upregulated during the transition.”* in line 240-243 of our new manuscript.

In several figures throughout the manuscript the authors picked a small panel of 2i/LIF or serum specific genes to exemplify the effects of Tead2 loss or gain of function. An explanation of the rationale behind their choice is scattered in different paragraphs, and remains somehow unclear in the current form of the manuscript. Why were these serum and 2i/LIF specific genes selected? More importantly, boxplots showing the expression changes of all serum and 2i/LIF specific genes should be consistently presented.

Answer: We selected serum-specific genes *Mmp2* and *Ank* from cluster 3 and 2i-specific genes *B4galt6*, *Kit*, *Idh2*, and *Ldhb* from cluster 5 of GO terms in Fig. 3E, which are related to developmental pathways and metabolic processes, respectively. Following the reviewer’s suggestion, we analyzed the changes of gene expression patterns in cluster 3 and cluster 5 during the SL-to-2iL transition and added these results into our new manuscript as Appendix Fig. S1, showing that:

“Cluster 3 genes (C3, 472 genes), involved in muscle structure and utero embryonic development (such as Mmp2 and Ank), demonstrated high expression after Tead2 loss during SL-to-2iL transition (Figs. 3E, F and Appendix Fig. S1H; Dataset EV3). Concurrently, cluster 5 genes (C5, 1210 genes), associated with carbohydrate and lactate metabolic processes (such as B4galt6, Kit, Idh2, and Ldhb), experienced downregulation after Tead2 loss during SL-to-2iL transition (Figs. 3E, G and Appendix Fig. S1I; Dataset EV3).”

Appendix Figure S1

Appendix Fig. S1H, I. Boxplots showing expression level of genes in cluster 3 (H) and cluster 5 (I) between wild-type and *Tead2*-knockout cells at day 0, 3, and 6 of the transition and 2iL-ESCs. The centerline indicates the median value, while the box and whiskers represent the interquartile range (IQR) and $1.5 \times \text{IQR}$, respectively, $n = 472$ in (H), $n = 1210$ in (I). *** $p < 0.001$. P -value was calculated by Mann-Whitney U test.

The authors suggest that Tead1 may bind to different regions compared to Tead2 in ESCs. In support, they claim that Tead1/3 motifs are not enriched at 2i-specific accessible regions. However, are the motifs recognised by these TFs sufficiently different to allow for such distinctions? Figure EV12C does not permit to appreciate the similarity between the respective motifs. The authors should reverse-complement the Tead2-4 motifs shown in the figure, specify how they are derived, and discuss in better details what could drive any difference in genome occupancy and function between Tead1 and Tead2.

Answer: We are appreciated for these comments. We took advantage of Homer software to analyze TEAD1-4 motifs from Biotin-TEAD2 ChIP-seq data (Fig. R2), and observed that the core motif sequences of TEAD1/3 and TEAD2/4 were identical after reverse complementing, which were both GGAAT. Therefore, this indeed could not prove that TEAD1/3 and TEAD2/4 bind to distinct gene regions. So, we removed the statement and evidence for “TEAD1 may bind to different regions compared to TEAD2 in ESCs” from the **DISCUSSION** section (including Fig. EV12C in the previous manuscript). Instead, we mentioned that “*Previous study showed that*

knocking TEAD1/3/4 down in ESCs results in downregulation of both Oct4 and Sox2 and loss of pluripotency (Lian et al, 2010). In contrast, knockdown of Tead2 had little effect on the expression of core pluripotent factors in ESCs (Fig. EV4C, D). We postulate that TEAD1 and TEAD2 function differently by interacting with diverse co-activators to modulate the pluripotency of stem cells.” in line 431-436 of our new manuscript.

Figure R2

Fig. R2. Sequence LOGOs of TEAD1-4 motifs enriched in ESCs by using Homer.

References

Lian I, Kim J, Okazawa H, Zhao J, Zhao B, Yu J, Chinnaiyan A, Israel MA, Goldstein LS, Abujarour R *et al* (2010) The role of YAP transcription coactivator in regulating stem cell self-renewal and differentiation. *Genes Dev* 24: 1106-1118

Dear Dr Hongjie Yao,

Thank you for submitting the revised version of your manuscript. I have now evaluated your amended manuscript and concluded that the remaining minor concerns have been sufficiently addressed.

I am pleased to inform you that your manuscript has been accepted for publication in the EMBO Journal.

On a different note, I would like to alert you that EMBO Press offers a format for a video-synopsis of work published with us, which essentially is a short, author-generated film explaining the core findings in hand drawings, and, as we believe, can be very useful to increase visibility of the work. Please see the following link for representative examples and their integration into the article web page:

<https://www.embopress.org/doi/full/10.15252/emboj.2019103932>

Best regards,

Daniel Klimmeck

Daniel Klimmeck, PhD
Senior Editor
The EMBO Journal
EMBO
Postfach 1022-40
Meyerhofstrasse 1
D-69117 Heidelberg
contact@embojournal.org
Submit at: <http://emboj.msubmit.net>
